# Draw-In-Mind: Rebalancing Designer-Painter Roles in Unified Multimodal Models Benefits Image Editing

**Ziyun Zeng**[1]     **David Junhao Zhang**[1]     **Wei Li**[2]     **Mike Zheng Shou**[1✉]

[1]Show Lab, National University of Singapore     [2]TikTok

## Abstract

In recent years, integrating multimodal understanding and generation into a single unified model has emerged as a promising paradigm. While this approach achieves strong results in text-to-image (T2I) generation, it still struggles with precise image editing. We attribute this limitation to an imbalanced division of responsibilities. The understanding module primarily functions as a translator that encodes user instructions into semantic conditions, while the generation module must simultaneously act as designer and painter, inferring the original layout, identifying the target editing region, and rendering the new content. This imbalance is counterintuitive because the understanding module is typically trained with several times more data on complex reasoning tasks than the generation module. To address this issue, we introduce *Draw-In-Mind* (DIM), a dataset comprising two complementary subsets: (**i**) DIM-T2I, containing 14M long-context image-text pairs to enhance complex instruction comprehension; and (**ii**) DIM-Edit, consisting of 233K chain-of-thought imaginations generated by GPT-4o, serving as explicit design blueprints for image edits. We connect a frozen Qwen2.5-VL-3B (Bai et al., 2025) with a trainable SANA1.5-1.6B (Xie et al., 2025a) via a lightweight two-layer MLP, and train it on the proposed DIM dataset, resulting in DIM-4.6B-T2I/Edit. Despite its modest parameter scale, DIM-4.6B-Edit achieves SOTA or competitive performance on the ImgEdit and GEdit-Bench benchmarks, outperforming much larger models such as UniWorld-V1 (Lin et al., 2025) and Step1X-Edit (Liu et al., 2025). These findings demonstrate that explicitly assigning the design responsibility to the understanding module provides significant benefits for image editing. Our dataset and models are available at https://github.com/showlab/DIM.

## 1 Introduction

Over the past few years, considerable effort has been devoted to developing unified models capable of both multimodal understanding and generation. Many such trials, *e.g.,* Show-o (Xie et al., 2024) and MetaQuery (Pan et al., 2025), have achieved impressive results on T2I generation, yet this paradigm falters when extended to instruction-guided image editing. Even recent methods such as BAGEL (Deng et al., 2025), UniWorld-V1 (Lin et al., 2025), and Step1X-Edit (Liu et al., 2025) struggle, as evidenced by the substantial performance gap with proprietary models like GPT-4o-Image (OpenAI, 2025) on the ImgEdit and GEdit-Bench benchmarks. While much concurrent research focuses on scaling parameters and data or on architectural modifications, in this paper we identify a novel challenge underlying current image editing models: *a fundamental imbalance division of responsibilities between the understanding and generation modules.*

Specifically, we observe that current image editing models often translate user instructions into semantic conditions through a semantic encoder, typically a multimodal large language model, yet this process lacks intermediate reasoning or refinement. The resulting conditions are then forwarded to the generation module, which is responsible for completing the editing process. At this stage, the

---

✉Corresponding Author.

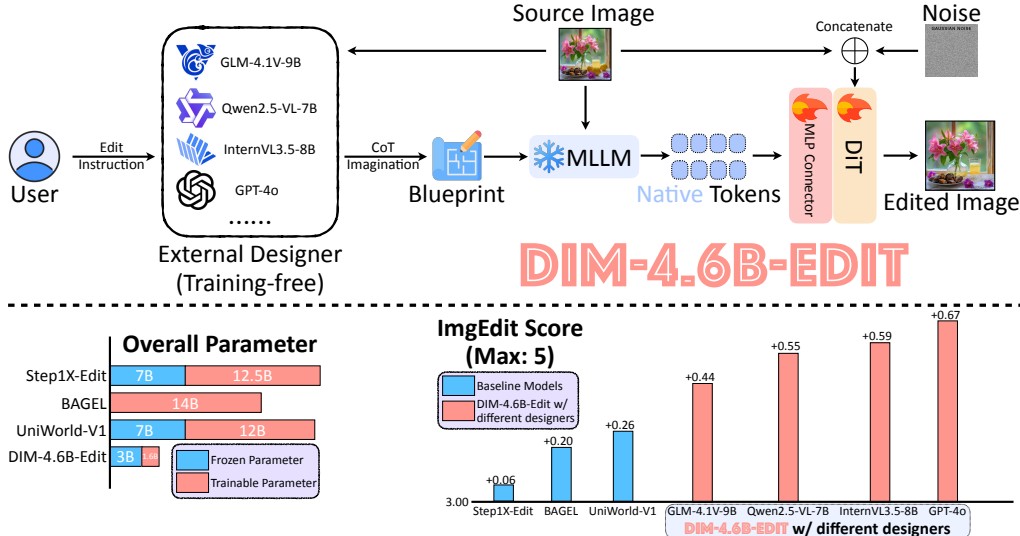

Figure 1: **Upper**: We employ a lightweight MLP connector to bridge a frozen MLLM, *i.e.*, Qwen2.5-VL-3B (Bai et al., 2025), with a trainable DiT, *i.e.*, SANA1.5-1.6B (Xie et al., 2025a), forming DIM-4.6B-Edit. In the editing process, we first leverage an external designer to produce a textual blueprint in a chain-of-thought style, which is then provided to DIM-4.6B-Edit to carry out precise image editing. **Lower**: DIM-4.6B-Edit establishes new state-of-the-art results on the challenging ImgEdit benchmark across diverse designers, while requiring 5× fewer parameters than existing frontier models. These results highlight both the effectiveness of the proposed DIM dataset and the generalizability of our approach.

generation module must simultaneously infer the original layout, determine the editing region, and render the new content. In this paradigm, the understanding module functions merely as a translator, while the generation module is burdened with the demanding tasks of both design and painting. This arrangement contrasts with natural human workflows, where planning and refinement typically precede the act of drawing. A more intuitive strategy is therefore to assign design-oriented reasoning to the understanding module while allowing the generation module to focus exclusively on painting.

Motivated by this observation, we introduce *Draw-In-Mind* (DIM), a dataset consisting of two complementary subsets: (**i**) DIM-T2I that contains 14M long-context image-text pairs annotated across 21 dimensions by in-house models to lay the groundwork for complex chain-of-thought comprehension; and (**ii**) DIM-Edit that comprises 233K high-quality chain-of-thought imagination generated by GPT-4o from existing image editing data, enabling the model to interpret explicit design plans from an external designer. We then establish a simple baseline by concatenating a frozen MLLM, *i.e.,* Qwen2.5-VL-3B, with a trainable DiT, *i.e.,* SANA1.5-1.6B, via a two-layer MLP and train it end-to-end on both public data and the proposed DIM dataset, resulting in DIM-4.6B-T2I/Edit. During edit inference, we employ an arbitrary external designer, feeding its chain-of-thought imagination into the model to guide precise image edits. The framework and performance overview are illustrated in Figure 1. Despite its simplicity, DIM-4.6B-Edit matches or outperforms 5× larger models such as Step1X-Edit (Liu et al., 2025) and UniWorld-V1 (Lin et al., 2025) on the ImgEdit benchmark. These results validate the effectiveness of the proposed DIM dataset and confirm the benefit of shifting the design responsibility from the generation module to the understanding module.

To summarize, we make the following contributions in this paper:

- We pinpoint a fundamental imbalanced division of responsibilities in current image editing models, which overburdens the generation module with both design and painting tasks.

- We introduce *Draw-In-Mind* (DIM), a unified dataset with two complementary subsets: DIM-T2I and DIM-Edit. This dataset explicitly frees the generation module from design responsibility and enables it to concentrate on painting, leading to substantial improvements in editing performance.

- We establish a simple baseline by connecting a frozen Qwen2.5-VL-3B with a trainable SANA1.5-1.6B via a two-layer MLP and train it on DIM. Despite its modest size and simple architecture, DIM-4.6B-Edit outperforms 5× larger competitors, validating the efficacy of DIM.

## 2 RELATED WORK

### 2.1 EXISTING IMAGE GENERATION DATASETS

**T2I Datasets**. Existing T2I datasets have provided many high-quality image-text pairs. They can be roughly grouped into three categories: (i) *purely AI-generated* data, *e.g.,* JourneyDB (Sun et al., 2023) and MidJourney-V6 (CortexLM, 2025), which collect images from the MidJourney API, and HQ-Edit (Hui et al., 2024), which generates images via DALL-E 3; (ii) *real-world* data, *e.g.,* COCO (Lin et al., 2014), which harvests images from Flickr and annotates them by human workers; and (iii) *mixed* data, *e.g.,* InstructP2P (Brooks et al., 2023), which sources real images from LAION-Aesthetics and produces edited variants via Prompt2Prompt (Hertz et al., 2022). Although these datasets deliver high perceptual quality, their prompts are typically short, limiting their utility for complex chain-of-thought reasoning in the editing stage. To ensure broad concept coverage, we opt for harvesting real-world images and annotate them with our in-house models from 21 dimensions, yielding 14M long-context image-text pairs, namely DIM-T2I, that form a robust foundation for complex CoT-guided editing.

**Image Editing Datasets**. Most large-scale image editing datasets either employ AI editors for end-to-end modification, *e.g.,* InstructP2P (Brooks et al., 2023) and HQ-Edit (Hui et al., 2024), or adopt a two-stage pipeline that first localizes the edit region via grounding models and then applies in-painting to alter the target objects, *e.g.,* UltraEdit (Zhao et al., 2024). A few efforts enlist human experts to annotate small-scale but high-quality edit pairs, *e.g.,* MagicBrush (Zhang et al., 2023) and SEED-Data-Edit-Part3 (Ge et al., 2024). However, their instructions are typically brief and occasionally misaligned with the corresponding image pairs. In contrast, our DIM-Edit comprises 233K deliberately designed chain-of-thought imaginations derived from these existing editing datasets. These rich and detailed CoT instructions act as explicit design blueprints, lighten the cognitive load on the generation module, and significantly improve editing performance.

### 2.2 UNIFIED MODELS FOR IMAGE GENERATION

**T2I Models**. In recent years, numerous successful attempts have been made to integrate understanding and generation modules into a unified architecture. These approaches can be broadly categorized into two technical routes: (**i**) *Integrative* approaches, *e.g.,* Show-o (Xie et al., 2024) and Janus (Wu et al., 2025a), which typically adopt an autoregressive generation paradigm to produce both image and text tokens; and (**ii**) *Connector-based* approaches, *e.g.,* MetaQuery (Pan et al., 2025), which use a connector to bridge an understanding module and a generation module. Since the understanding and generative capabilities are tightly coupled in the former architecture and sometimes lead to conflicts that degrade both, we adopt the connector-based design to preserve state-of-the-art cognitive ability by freezing the understanding module while enhancing generation performance.

**Image Editing Models**. When it comes to image editing, the challenge becomes significantly harder, as neither the latest integrative models (Lin et al., 2025) nor connector-based ones (Liu et al., 2025) achieve satisfactory performance on mainstream benchmarks such as ImgEdit and GEdit-Bench compared to proprietary models like GPT-4o-Image, even when employing large-scale understanding and generation models such as Qwen2.5-VL-7B (Bai et al., 2025) and FLUX.1-dev (Labs, 2024). This suggests that simply scaling model size is not an effective strategy for improving image editing capability. In this work, we take a different approach by addressing the problem from a perspective of *imbalanced division of responsibilities*. We propose DIM-4.6B-Edit, which leverages an external designer to create blueprints in a CoT manner in the textual space before editing. Despite having only 1.6B generative parameters, our model achieves SOTA editing performance, highlighting the effectiveness of shifting the design responsibility to the understanding module.

## 3 METHODOLOGY

### 3.1 THE DRAW-IN-MIND (DIM) DATASET

#### 3.1.1 DIM-T2I

There are typically two strategies to train an editing model, *i.e.,* (**i**) learning drawing first (T2I), followed by adaptation for editing, and (**ii**) directly learning editing. We observe that the vast majority of image editing models are built upon established T2I foundations (Brooks et al., 2023; Zhao et al., 2024; Liu et al., 2025). This aligns with the first strategy and represents a robust technical route

that benefits from curriculum learning. Consequently, we elected to achieve basic T2I ability and subsequently fine-tune the base model for the more challenging editing task.

However, we observed that despite the current T2I datasets performing well in terms of prompt-image alignment and image perceptual quality, the prompts in existing datasets are typically short and simple, as shown in Table 1. While these prompts accurately capture the semantics of the target image, they fall short in fostering long-context comprehension, which is an essential foundation for complex CoT-guided image editing. To bridge this gap, we collect 14M images with resolutions higher than $512 \times 512$ from the web. We believe that the aspects emphasized in widely recognized understanding datasets and benchmarks effectively capture the most frequent interactions between humans and objects in the real world. Therefore, we conduct a thorough literature review and an empirical analysis of existing understanding datasets and benchmarks, and finally derive 21 diverse dimensions and use internal models to generate long and detailed annotations, thoroughly covering all dimensions, resulting in DIM-T2I. As shown in Table 1, its average prompt length is at least twice that of existing corpora, effectively establishing a strong basis for complex CoT-guided image editing. The dimension-specific prompts and referred datasets/benchmarks are listed in Appendix F.

### 3.1.2 DIM-EDIT

As for image editing, the short-prompt issue is even more pronounced in current datasets. As shown in Table 1, prompts in mainstream datasets are generally overly simplistic, often consisting of only a few descriptive words. Such data is not conducive to effective image editing learning, as the prompts may fail to accurately reflect the actual changes between the source and target images. This phenomenon can be attributed to two main reasons: (**i**) *Inaccurate AI editing or human misoperation.* We observe that even SOTA proprietary models like GPT-

Table 1: The statistics of existing high-quality datasets and our proposed DIM dataset. APL is short for Average Prompt Length, counted by word numbers.

| Dataset Name | Size | Source | APL |
|---|---|---|---|
| *Text-to-Image* | | | |
| MidJourney-V6 (CortexLM, 2025) | 1.2M | AI Gen. | 9.59 |
| COCO (Lin et al., 2014) | 0.4M | Real | 10.46 |
| InstructP2P (Brooks et al., 2023) | 0.6M | Real & AI Gen. | 11.37 |
| JourneyDB (Sun et al., 2023) | 4.2M | AI Gen. | 29.27 |
| HQ-Edit (Hui et al., 2024) | 0.2M | AI Gen. | 38.08 |
| Dimba (Fei et al., 2024) | 0.3M | Real | 78.29 |
| DIM-T2I | 14M | Real | 146.76 |
| *Image Editing* | | | |
| MagicBrush (Zhang et al., 2023) | 8K | Real | 6.50 |
| SEED-Data-Edit-Part3 (Ge et al., 2024) | 82K | Real | 7.39 |
| UltraEdit (Zhao et al., 2024) | 4M | AI Gen. | 8.32 |
| ShareGPT-4o-Image (Chen et al., 2025b) | 46K | AI Gen. | 34.75 |
| DIM-Edit | 233K | Real & AI Gen. | 252.64 |

4o-Image frequently over-edit images, *e.g.,* removing objects not mentioned in the prompts. Such cases exist widely in AI-generated datasets like ShareGPT-4o-Image and UltraEdit. While in human-controlled datasets, operators may misunderstand or misapply the edits, resulting in unaligned data. (**ii**) *Ambiguous semantics.* Even if the prompt correctly describes the intended change, overly simple prompts can still result in multiple equally valid edits. For example, in SEED-Data-Edit-Part3, a common prompt is "change the background", yet the definition of "background" varies across images, while in practice the change almost always occurs in the sky, thereby reducing the effectiveness of the resulting edit data.

In addition, existing models typically use the understanding module merely as a translator, directly converting natural language instructions into semantic conditions. The generator must then rely on these conditions to simultaneously organize the layout of the edited image, recognize existing objects, localize the edit area, render new content, and preserve unchanged regions. In other words, the generator is forced to act as both designer and painter, which is a challenging and counterintuitive setup. By contrast, humans naturally prepare a mental blueprint before editing and then simply let their hands follow it to complete the changes.

Motivated by the above issues, we propose DIM-Edit, which first optimizes prompts and then imitates human thinking to complete the edits. The DIM-Edit creation pipeline is illustrated in Figure 2. We construct it from 233K high-quality image pairs collected from three sources: (**i**) 160K highly consistent edit pairs from UltraEdit, referred to as UltraEdit-160K-CoT, selected using a joint SSIM, DINOv2 similarity, and CLIP similarity-based filtering; (**ii**) 46K semantically rich samples from the editing subset of ShareGPT-4o-Image, referred to as ShareGPT-4o-Image-CoT; and (**iii**) 8K human-edited images from the MagicBrush training set and 19K human-edited images from SEED-Data-Edit-Part3, specifically targeting remove operations, referred to as HumanEdit-CoT. A detailed data collection pipeline can be found in Appendix D.

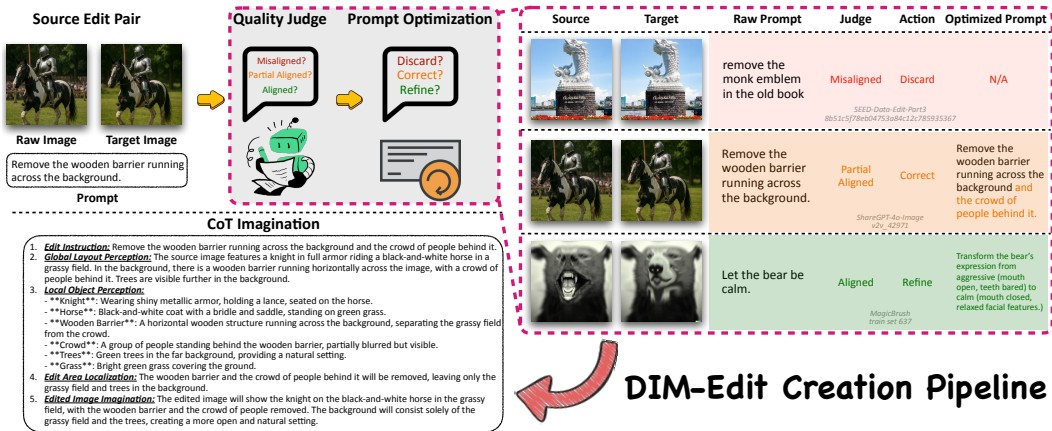

Figure 2: The creation pipeline of DIM-Edit begins with a quality assessment of existing image editing data, followed by prompt optimization using GPT-4o. Finally, the optimized prompts together with the corresponding image pairs are fed into GPT-4o, which generates a four-step chain-of-thought imagination in the textual space.

After collecting raw data, we first sent the raw edit pairs to GPT-4o for prompt quality evaluation, as shown in Figure 2. The results are categorized into three groups: (**i**) *Misaligned.* The prompt does not reflect the actual edit at all, possibly due to misannotation or misoperation. (**ii**) *Partially aligned.* The target image exhibits over-editing, *i.e.,* redundant objects are added to or removed from the source image. (**iii**) *Aligned.* The prompt fully corresponds to the edits.

Next, we take different actions to optimize the prompts based on the judgment: (**i**) For misaligned prompts, they are discarded outright. (**ii**) For partially aligned prompts, we ask GPT-4o to add details about unmentioned changes, *e.g.,* including objects that were incorrectly removed in the prompt. (**iii**) For aligned prompts, we instruct GPT-4o to remove ambiguity and refine the prompt, for example, by specifying the exact objects to be edited to avoid confusion with visually similar objects.

Finally, we provide the optimized prompts, along with the source image to GPT-4o and instruct it to produce a four-step CoT imagination that emulates human editing behavior. For the sake of accuracy, we also provide the target image to it for reference. The target of each CoT step is as follows: (**i**) *Global Layout Perception*: identify and describe all key objects and their positions in the source image. (**ii**) *Local Object Perception*: describe the appearance of each object or background element in the source image, including shape, color, texture, and state. (**iii**) *Edit Area Localization*: specify which objects or regions will be modified, based on the refined instruction. (**iv**) *Edited Image Imagination*: describe the expected appearance of the edited image, with an emphasis on the modified areas. As shown in Table 1 and Figure 2, the resulting CoT imagination is not only ultra-long but also highly accurate, effectively removing the design responsibility from the generation module and thereby significantly enhancing the efficiency of image editing learning. A quality assessment of the CoTs involving both MLLMs and human verification can be found in Appendix E.

## 3.2 DIM-4.6B-T2I/EDIT

Leveraging MLLMs to provide multimodal conditions for image generation has become a common practice recently. In this work, we first build a base T2I model and then adapt it to the editing task.

For the base T2I model, we start by establishing a simple baseline, similar to MetaQuery (Pan et al., 2025), to preserve state-of-the-art understanding capability. We select Qwen2.5-VL-3B (Bai et al., 2025) as the MLLM and SANA1.5-1.6B (Xie et al., 2025a) as the diffusion decoder for their modest size. Unlike MetaQuery, which employs a large 24-layer transformer with 1.6B parameters as a connector between the MLLM and the diffusion decoder, we adopt a much simpler design, *i.e.,* a two-layer MLP, to directly project multimodal tokens into the generation space. We refer to this model as DIM-4.6B-T2I, illustrated in Figure 1. We train DIM-4.6B-T2I on a mixture of the proposed DIM-T2I dataset and an additional 6.9M image-text pairs from MidJourney-V6 (CortexLM, 2025), COCO (Lin et al., 2014), InstructP2P (Brooks et al., 2023), JourneyDB (Sun et al., 2023), HQ-Edit (Hui et al., 2024), and Dimba (Fei et al., 2024). During training, Qwen2.5-VL-3B remains

Table 2: The text-to-image generation performance on **GenEval** and **MJHQ-30K**. ↑ and ↓ indicate that higher and lower values are better, respectively; † denotes using an LLM rewriter; ❄ and 🔥 denote frozen and trainable parameters, respectively.

| Model | Params | GenEval↑ | | | | | | | MJHQ-30K↓ |
|---|---|---|---|---|---|---|---|---|---|
| | | Single Obj. | Two Obj. | Counting | Colors | Position | Color Attr. | Overall | FID |
| *Gen. Only* | | | | | | | | | |
| PixArt-α (Chen et al., 2023) | 0.6B🔥 | 0.98 | 0.50 | 0.44 | 0.80 | 0.08 | 0.07 | 0.48 | 6.14 |
| SDXL (Podell et al., 2023) | 2.6B🔥 | 0.98 | 0.74 | 0.39 | 0.85 | 0.15 | 0.23 | 0.55 | 8.76 |
| DALL-E·3 (Betker et al., 2023) | - | 0.96 | 0.87 | 0.47 | 0.83 | 0.43 | 0.45 | 0.67 | - |
| SD3-Medium (Esser et al., 2024) | 2.0B🔥 | 0.99 | 0.94 | 0.72 | 0.89 | 0.33 | 0.60 | 0.74 | 11.92 |
| *Unified* | | | | | | | | | |
| Janus (Wu et al., 2025a) | 1.3B🔥 | 0.97 | 0.68 | 0.30 | 0.84 | 0.46 | 0.42 | 0.61 | 10.10 |
| Emu3-Gen† (Wang et al., 2024b) | 8.0B🔥 | 0.99 | 0.81 | 0.42 | 0.80 | 0.49 | 0.45 | 0.66 | - |
| Show-o (Xie et al., 2024) | 1.3B🔥 | 0.98 | 0.80 | 0.66 | 0.84 | 0.31 | 0.50 | 0.68 | 15.18 |
| Show-o2-7B (Xie et al., 2025b) | 7.0B🔥 | 1.00 | 0.87 | 0.58 | 0.92 | 0.52 | 0.62 | 0.76 | - |
| Janus-Pro-7B (Chen et al., 2025c) | 7.0B🔥 | 0.99 | 0.89 | 0.59 | 0.90 | 0.79 | 0.66 | 0.80 | 13.48 |
| BAGEL (Deng et al., 2025) | 14.0B🔥 | 0.99 | 0.94 | 0.81 | 0.88 | 0.64 | 0.63 | 0.82 | - |
| MetaQuery-L† (Pan et al., 2025) | 3.0B❄\|3.2B🔥 | - | - | - | - | - | - | 0.78 | 6.35 |
| DIM-4.6B-T2I† | 3.0B❄\|1.6B🔥 | 0.99 | 0.89 | 0.63 | 0.86 | 0.62 | 0.61 | 0.77 | 5.50 |

frozen, and we finetune only the parameters of the connector and SANA1.5-1.6B. Notably, distillation datasets like BLIP3-o-60K (Chen et al., 2025a) explicitly curate data to align with the structural patterns of benchmarks like GenEval, we exclude them to avoid any risk of data leakage (Wu et al., 2025b) or benchmark hacking in the evaluation to justify the contribution of our DIM data. We adopt vanilla flow matching as the sole objective, avoiding parameter-tuning tricks to highlight data effectiveness and maintain simplicity.

Thanks to the rich world knowledge and high-quality long-context prompts in DIM-T2I, the trained DIM-4.6B-T2I model provides a strong foundation for complex CoT comprehension. We then adopt a two-stage training strategy to adapt it for the editing task. In the first stage, we initialize the editing model from DIM-4.6B-T2I and fine-tune it on the UltraEdit (Zhao et al., 2024) dataset to develop basic editing capability. Following InstructP2P (Brooks et al., 2023), we concatenate the source image with noise along the channel dimension, as illustrated in Figure 1. In the second stage, we fine-tune the stage-one model exclusively on the proposed DIM-Edit dataset, resulting in DIM-4.6B-Edit. During inference, we employ an external designer to prepare a blueprint in the same format as DIM-Edit, except without access to the target image, ensuring alignment with real usage scenarios.

## 4 EXPERIMENTS

### 4.1 EXPERIMENTAL SETUP

During training, we use AdamW as the optimizer and keep most hyperparameters unchanged for simplicity. For DIM-4.6B-T2I, we first warm up by training only the connector for one epoch with a learning rate of $2 \times 10^{-5}$, then jointly train the connector and SANA1.5-1.6B for eight epochs with the same rate and a batch size of 256. For DIM-4.6B-Edit, we set the batch size to 32, training on UltraEdit for 10 epochs at a $1 \times 10^{-4}$ learning rate, then finetuning on DIM-Edit for 50 epochs at a $1 \times 10^{-5}$ learning rate. During inference, GPT-4o serves as the designer unless otherwise specified.

Although the primary focus of this paper is image editing, we evaluate DIM-4.6B-T2I on T2I benchmarks to verify the effectiveness of DIM-T2I. We report the GenEval (Ghosh et al., 2023) scores and MJHQ-30K (Li et al., 2024b) FID. Following MetaQuery (Pan et al., 2025) and Emu3 (Pan et al., 2025), we test LLM-rewritten prompts for GenEval evaluation. For image editing, we report scores on the recently proposed ImgEdit (Lin et al., 2025) and GEdit-Bench-EN (Liu et al., 2025) benchmarks, using GPT-4.1 for evaluation to ensure fair comparison with existing results. We also report results on MagicBrush (Zhang et al., 2023) to show the performance on automated metrics.

### 4.2 MAIN RESULTS

#### 4.2.1 TEXT-TO-IMAGE GENERATION

We first report T2I performance on GenEval and MJHQ-30K in Table 2. Our DIM-4.6B-T2I adopts a simple architecture with very few trainable parameters yet achieves SOTA or competitive performance, demonstrating the high data quality of DIM-T2I. For semantic alignment, DIM-4.6B-

Table 3: The image editing performance on **ImgEdit**. We use GPT-4.1 for evaluation to ensure consistency with the existing results reported in UniWorld-V1. ∗ indicates results evaluated by us using the official weights; ❄ and 🔥 denote frozen and trainable parameters, respectively.

| Model | Params | Add | Adjust | Extract | Replace | Remove | Background | Style | Hybrid | Action | Overall |
|-------|--------|-----|--------|---------|---------|--------|------------|-------|--------|--------|---------|
| MagicBrush (Zhang et al., 2023) | 0.9B🔥 | 2.84 | 1.58 | 1.51 | 1.97 | 1.58 | 1.75 | 2.38 | 1.62 | 1.22 | 1.83 |
| Instruct-P2P (Brooks et al., 2023) | 0.9B🔥 | 2.45 | 1.83 | 1.44 | 2.01 | 1.50 | 1.44 | 3.55 | 1.20 | 1.46 | 1.88 |
| AnyEdit (Yu et al., 2025) | 1.3B🔥 | 3.18 | 2.95 | 1.88 | 2.47 | 2.23 | 2.24 | 2.85 | 1.56 | 2.65 | 2.45 |
| UltraEdit (Zhao et al., 2024) | 2.0B🔥 | 3.44 | 2.81 | 2.13 | 2.96 | 1.45 | 2.83 | 3.76 | 1.91 | 2.98 | 2.70 |
| Step1X-Edit (Liu et al., 2025) | 7.0B❄\|12.5B🔥 | 3.88 | 3.14 | 1.76 | 3.40 | 2.41 | 3.16 | 4.63 | 2.64 | 2.52 | 3.06 |
| BAGEL (Deng et al., 2025) | 14.0B🔥 | 3.56 | 3.31 | 1.70 | 3.30 | 2.62 | 3.24 | 4.49 | 2.38 | 4.17 | 3.20 |
| UniWorld-V1 (Lin et al., 2025) | 7.0B❄\|12.0B🔥 | 3.82 | 3.64 | 2.27 | 3.47 | 3.24 | 2.99 | 4.21 | 2.96 | 2.74 | 3.26 |
| Janus-4o* (Chen et al., 2025b) | 7.0B🔥 | 3.35 | 3.35 | 2.25 | 3.01 | 2.18 | 3.32 | 4.71 | 2.49 | 4.04 | 3.19 |
| GPT-4o-Image (OpenAI, 2025) | - | 4.61 | 4.33 | 2.90 | 4.35 | 3.66 | 4.57 | 4.93 | 3.96 | 4.89 | 4.20 |
| DIM-4.6B-Edit | 3.0B❄\|1.6B🔥 | 4.09 | 3.47 | 2.30 | 4.00 | 3.43 | 3.87 | 4.92 | 2.85 | 4.08 | 3.67 |

Table 4: The overall task-wise performance on **GEdit-Bench-EN** Full set. ∗ indicates results evaluated by us. Task abbreviations: Background Change (BC), Color Alter (CA), Material Alter (MA), Motion Change (MC), PS Human (PH), Style Change (SC), Subject-Add (SA), Subject-Remove (SRM), Subject-Replace (SRP), Text Change (TC), Tone Transfer (TT), and Average (AVG).

| Model | BC | CA | MA | MC | PH | SC | SA | SRM | SRP | TC | TT | AVG | AVG w/o TC |
|-------|----|----|----|----|----|----|----|-----|-----|----|----|-----|-----------|
| UniWorld-V1 (Lin et al., 2025) | 4.92 | 6.37 | 4.79 | 1.85 | 4.03 | 5.64 | 7.23 | 6.17 | 5.70 | 1.15 | 5.54 | 4.85 | 5.22 |
| Janus-4o* (Chen et al., 2025b) | 4.31 | 5.02 | 4.41 | 2.71 | 4.07 | 5.80 | 4.07 | 1.69 | 3.69 | 2.35 | 3.96 | 3.83 | 3.97 |
| Step1X-Edit (Liu et al., 2025) | 7.03 | 6.26 | 6.46 | 3.66 | 5.23 | 7.24 | 7.17 | 6.42 | 7.39 | 7.40 | 6.62 | 6.44 | 6.35 |
| DIM-4.6B-Edit | 7.02 | 6.81 | 6.00 | 4.67 | 5.88 | 7.16 | 7.48 | 6.67 | 6.76 | 2.99 | 6.55 | 6.18 | 6.50 |

T2I shows only a small gap compared to much larger models like BAGEL (Deng et al., 2025) on GenEval. Compared with MetaQuery (Pan et al., 2025), which employs a large 1.6B-parameter connector for query learning, our model achieves nearly the same performance using only a two-layer MLP connector and naive multimodal tokens. In addition, it attains optimal perceptual quality, as evidenced by the lowest FID on the aesthetics-oriented MJHQ-30K benchmark. These results indicate that *even without complex aesthetic filtering, carefully crafted long-context prompts enable robust text-to-image generation*, offering a practical approach for rapid large-scale dataset creation by directly harvesting images from the web.

### 4.2.2 IMAGE EDITING

The image editing performance on ImgEdit is reported in Table 3. Our DIM-4.6B-Edit shows a significant improvement over previously available open source models. In comparison with other connector-based architectures such as Step1X-Edit and UniWorld-V1, which rely on a 12B FLUX backend for generation together with a 7B multimodal large language model for condition translation, DIM-4.6B-Edit achieves superior results while maintaining both a much smaller total parameter count and a very limited number of trainable parameters.

Since DIM-Edit includes high-quality images from ShareGPT-4o-Image (Chen et al., 2025b), we also evaluate Janus-4o, which is trained on the same dataset, for reference. Janus-4o achieves only suboptimal results, indicating that the improvement comes from DIM-Edit itself, whose natural and precise edit blueprints substantially enhance editing performance. These encouraging results validate our assumption that imbalanced division of responsibilities degrades image editing, confirm the soundness of our data creation pipeline, and highlight the effectiveness of the Draw-In-Mind paradigm: assigning the design responsibility to the understanding module while allowing the generation module to focus on actual editing exclusively.

We further demonstrate the capability of DIM-4.6B-Edit through intuitive visual comparisons of editing results on four AI-generated out-of-domain images in Figure 3. Janus-4o exhibits severe distortions despite being trained on GPT-4o-generated edit pairs, while Step1X-Edit produces less natural edits (rows 2-4) and fails in complex scenarios such as row 1, which involves manipulating multiple objects. In contrast, DIM-4.6B-Edit successfully follows the instructions to produce natural and consistent edited images. Please refer to Appendix B for more visualizations.

We also include overall task-wise performance on GEdit-Bench-EN in Table 4. The results reveal a similar pattern as reported in UniWorld-V1 (Lin et al., 2025): Step1X-Edit achieves notable gains in the Text Change task, whereas other models, including ours, perform less effectively due to the

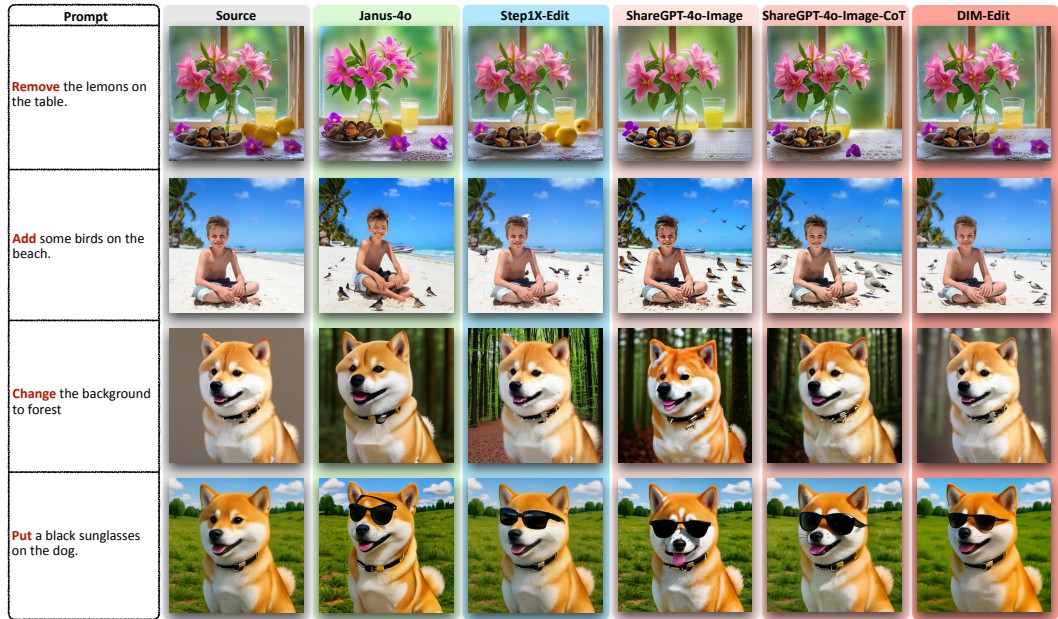

Figure 3: **Green** and **Blue** : the edits of Janus-4o and Step1X-Edit; **Red** : the edits of our models trained on different data corpora. All variants are tuned from the base checkpoint ✿ in Table 8.

Table 5: The **MagicBrush** test set performance. Metrics are calculated between human-edited groundtruth and AI-generated edits. 🥇 and 🥈 denote the 1st and 2nd best model, respectively.

| Method | Gen Params | L1↓ | CLIP-I↑ | DINO↑ |
|---|---|---|---|---|
| InstructP2P (Brooks et al., 2023) | 0.9B | 0.114 | 0.851 | 0.744 |
| MagicBrus (Zhang et al., 2023) | 0.9B | 0.074 | 0.908 | 0.847 |
| UltraEdit (Zhao et al., 2024) | 2.0B | 0.066 | 0.904 | 0.852 |
| FluxEdit (Paul, 2025) | 12.0B | 0.114 | 0.779 | 0.663 |
| FLUX.1 Fill (Labs, 2024) | 12.0B | 0.192 | 0.795 | 0.669 |
| RF-Solver Edit (Wang et al., 2024a) | 12.0B | 0.112 | 0.766 | 0.675 |
| ACE++ (Mao et al., 2025) | 12.0B | 0.195 | 0.741 | 0.591 |
| ICEdit (Zhang et al., 2025) | 12.0B | 0.060🥇 | 0.928🥇 | 0.853🥈 |
| DIM-4.6B-Edit | 1.6B | 0.065🥈 | 0.928🥇 | 0.882🥇 |

absence of such data in DIM-Edit. Excluding the Text Change task, DIM-4.6B-Edit beats Step1X-Edit while maintaining a compact size, underscoring the high efficacy of our CoT data. Please refer to Appendix B for full GEdit-Bench-EN results.

We further conduct evaluation on the MagicBrush to test automated pixel-to-pixel metrics computed between human-edits and AI-edits. The results are presented in Table 5. DIM-4.6B-Edit achieves SOTA performance. Notably, ICEdit employs a 12B FLUX.1 Fill backbone, with MagicBrush samples constituting approximately 20% of its total training set. In contrast, DIM-4.6B-Edit utilizes a compact 1.6B generation backbone, where MagicBrush data accounts for less than 3% of our DIM-Edit dataset. These comparable results validate the effectiveness of the Draw-In-Mind paradigm and the generalizability of our DIM-Edit CoT. Despite our training distribution being significantly less driven by MagicBrush data, our model matches the performance of 5× larger competitors.

### 4.3 ABLATION STUDY

**Generalizability to External Designers.** Although our proposed DIM-Edit is annotated with GPT-4o, we show that the resulting DIM-4.6B-Edit is compatible with various external designers, as reported in Table 6. In the first row, we remove the designer and directly use the raw prompt from ImgEdit. Even under this setting, DIM-4.6B-Edit achieves performance comparable to frontier models such as BAGEL, demonstrating that high-quality CoT annotations help strengthen basic editing by mitigating prompt–edit misalignment. We then replace GPT-4o with four mainstream MLLMs as external designers, *i.e.*, Qwen2.5-VL-7B (Bai et al., 2025), MiMo-VL-7B (Xiaomi,

Table 6: The **ImgEdit** performance *w.r.t.* different *external* designers.

| External Designer | Add | Adjust | Extract | Replace | Remove | Background | Style | Hybrid | Action | Overall |
|---|---|---|---|---|---|---|---|---|---|---|
| - | 3.53 | 3.23 | 2.01 | 3.49 | 1.47 | 3.42 | 4.79 | 2.35 | 3.64 | 3.10 |
| Qwen2.5-VL-7B (Bai et al., 2025) | 3.95 | 3.35 | 2.25 | 3.85 | 3.31 | 3.57 | 4.88 | 2.81 | 4.02 | 3.55 |
| MiMo-VL-7B (Xiaomi, 2025) | 3.95 | 3.32 | 2.20 | 3.75 | 2.46 | 3.82 | 4.88 | 2.52 | 3.93 | 3.43 |
| InternVL3.5-8B (Wang et al., 2025) | 3.98 | 3.40 | 2.05 | 4.14 | 3.30 | 3.84 | 4.94 | 2.77 | 3.89 | 3.59 |
| GLM-4.1V-9B (Hong et al., 2025) | 3.95 | 3.27 | 2.23 | 3.90 | 2.64 | 3.81 | 4.92 | 2.23 | 4.02 | 3.44 |
| GPT-4o (Hurst et al., 2024) | 4.09 | 3.47 | 2.30 | 4.00 | 3.43 | 3.87 | 4.92 | 2.85 | 4.08 | 3.67 |

Table 7: The **ImgEdit** performance *w.r.t.* the *internal* Qwen2.5-VL-3B designer.

| Internal Designer | Add | Adjust | Extract | Replace | Remove | Background | Style | Hybrid | Action | Overall |
|---|---|---|---|---|---|---|---|---|---|---|
| - | 3.53 | 3.23 | 2.01 | 3.49 | 1.47 | 3.42 | 4.79 | 2.35 | 3.64 | 3.10 |
| Qwen2.5-VL-3B ❄ | 3.80 | 3.24 | 2.03 | 3.89 | 3.21 | 3.52 | 4.92 | 2.71 | 4.05 | 3.49 |
| Qwen2.5-VL-3B 🔥 | 3.96 | 3.36 | 2.25 | 3.98 | 3.31 | 3.81 | 4.95 | 2.83 | 4.02 | 3.61 |
| GPT-4o | 4.09 | 3.47 | 2.30 | 4.00 | 3.43 | 3.87 | 4.92 | 2.85 | 4.08 | 3.67 |

2025), InternVL3.5-8B (Bai et al., 2025), and GLM-4.1V-9B (Hong et al., 2025). All of them deliver strong results compared to previous state-of-the-art models in Table 3, highlighting the robustness of DIM-4.6B-Edit and the generalizability of our DIM framework. Furthermore, models equipped with external designers significantly outperform the raw-prompt setting, confirming that CoT imagination effectively reduces the burden on the generation modules and enhances overall editing quality.

**Integrated End-to-End Evaluation.** To exclude potential influence from external designers, we establish a "self-play" configuration. In this setup, CoT embeddings generated by the internal MLLM (Qwen2.5-VL-3B) are directly fed into the painter to execute edits, effectively eliminating the need for the external inference round. The result (Table 7 2nd row) shows that this "self-play" model achieves SOTA performance, validating the effectiveness of the Draw-In-Mind paradigm and the high quality of the DIM-Edit data. We further investigate whether the CoT blueprints in DIM-Edit can serve as a corpus to bridge the gap between open-source and closed-source designers. To this end, we perform lightweight fine-tuning on Qwen2.5-VL-3B and subsequently feed its blueprints into DIM-4.6B-Edit. The results (Table 7 3rd row) demonstrate that fine-tuning from DIM-Edit's CoTs can effectively mitigate the performance disparity with the proprietary models like GPT-4o.

**Data Composition.** In Table 8, we present a rigorous data composition analysis for the editing task to identify the sources of performance improvements. In the first stage, we observe that training solely on ShareGPT-4o-Image already yields a satisfactory ImgEdit score, indicating strong semantic alignment, which is consistent with the behavior of Janus-4o. However, models trained exclusively on GPT-4o-generated data tend to alter the overall layout noticeably, which is undesirable. In contrast, training on UltraEdit produces slightly lower scores but preserves better consistency between the source and target images. When combining the two datasets, performance improves significantly, as the model benefits from the semantic richness while retaining the edit consistency.

In the second stage, we finetune the checkpoint trained solely on UltraEdit. The effectiveness of our CoT data is demonstrated by comparing row 4 with row 3 in Table 8, where using the CoT version of ShareGPT-4o-Image yields a significant improvement in overall scores compared with its non-CoT counterpart. We also observe that using UltraEdit-160K-CoT alone provides only marginal gains, while the HumanEdit-CoT portion has a more notable impact due to its high edit quality, though still less pronounced than the semantically rich ShareGPT-4o-Image-CoT. When combining all three CoT components, *i.e.,* the proposed DIM-Edit, performance improves substantially once again, indicating that UltraEdit-160K-CoT and HumanEdit-CoT are crucial for maintaining edit consistency, which is consistent with the pattern of row 3.

The visualization of three variants finetuned from the base checkpoint 🌼 in Table 8 is shown in Figure 3 for intuitive analysis. The variant tuned on ShareGPT-4o-Image significantly alters the layout despite following the edit prompt, while its counterpart tuned on ShareGPT-4o-Image-CoT preserves more details, indicating that CoT imagination helps maintain editing consistency. However, using ShareGPT-4o-Image-CoT alone still produces unstable edits. In contrast, the model tuned on the full DIM-Edit dataset, *i.e.,* DIM-4.6B-Edit, achieves the best results in both semantic alignment and edit consistency, demonstrating the effectiveness of all three data components in DIM-Edit.

Table 8: Impact of data compositions during the two training stages of DIM-4.6B-Edit on **ImgEdit**. Stage 2 models are tuned from checkpoint ❀.

| Data Composition | Add | Adjust | Extract | Replace | Remove | Background | Style | Hybrid | Action | Overall |
|---|---|---|---|---|---|---|---|---|---|---|
| *Stage1 Non-CoT Data* | | | | | | | | | | |
| ShareGPT-4o-Image | 3.35 | 2.74 | 1.93 | 3.05 | 1.95 | 3.16 | 4.91 | 2.00 | 3.70 | 2.98 |
| UltraEdit-4M ❀ | 3.41 | 3.03 | 1.91 | 2.94 | 1.07 | 3.09 | 3.77 | 2.64 | 2.97 | 2.76 |
| + ShareGPT-4o-Image | 3.85 | 3.09 | 1.84 | 3.71 | 2.26 | 3.51 | 4.88 | 2.17 | 3.67 | 3.22 |
| *Stage2 CoT Data* | | | | | | | | | | |
| ❀ + ShareGPT-4o-Image-CoT | 4.01 | 3.19 | 2.19 | 3.74 | 2.53 | 3.57 | 4.93 | 2.25 | 3.66 | 3.34 |
| ❀ + UltraEdit-160K-CoT | 3.69 | 3.21 | 1.90 | 2.50 | 1.22 | 3.20 | 3.53 | 2.71 | 3.14 | 2.79 |
| + HumanEdit-CoT | 3.63 | 2.99 | 2.01 | 3.01 | 2.64 | 3.11 | 3.73 | 3.03 | 3.01 | 3.02 |
| ❀ + DIM-Edit | 4.09 | 3.47 | 2.30 | 4.00 | 3.43 | 3.87 | 4.92 | 2.85 | 4.08 | 3.67 |

Table 9: Impact of CoT compositions on **ImgEdit**. GLP refers to Global Layout Perception, LOP to Local Object Perception, EAL to Edit Area Localization, and EII to Edited Image Imagination.

| CoT Composition | Add | Adjust | Extract | Replace | Remove | Background | Style | Hybrid | Action | Overall |
|---|---|---|---|---|---|---|---|---|---|---|
| DIM-Edit | 4.09 | 3.47 | 2.30 | 4.00 | 3.43 | 3.87 | 4.92 | 2.85 | 4.08 | 3.67 |
| w/o GLP | 3.85 | 3.29 | 2.06 | 3.91 | 3.24 | 3.55 | 4.80 | 2.79 | 3.92 | 3.49 |
| w/o LOP | 3.80 | 3.15 | 1.92 | 3.83 | 3.07 | 3.60 | 4.79 | 2.44 | 3.92 | 3.39 |
| w/o EAL | 3.79 | 3.25 | 1.96 | 3.73 | 2.96 | 3.65 | 4.81 | 2.82 | 3.82 | 3.42 |
| w/o EII | 3.77 | 3.22 | 1.82 | 3.88 | 2.96 | 3.61 | 4.78 | 2.55 | 3.58 | 3.35 |

**CoT Composition.** We also analyze the effect of each CoT component by individually removing it, as shown in Table 9. All components contribute positively to the performance, though their importance varies. The GLP has only a minor impact, likely because it is an easy task for the generator. In contrast, the other three CoT components, *i.e.,* LOP, EAL, and EII, have a significant effect. LOP and EAL require the model to focus on specific regions, while EII demands complex reasoning; none of these are trivial for the generator. These findings further validate the Draw-In-Mind paradigm, which reduces the cognitive burden on the generator and thereby improves performance.

Table 10: The **ImgEdit** performance of models initialized from scratch/DIM-4.6B-T2I.

| Initialization | Add | Adjust | Extract | Replace | Remove | Background | Style | Hybrid | Action | Overall |
|---|---|---|---|---|---|---|---|---|---|---|
| Scratch | 2.70 | 2.56 | 1.93 | 2.23 | 2.47 | 2.82 | 4.68 | 2.38 | 2.15 | 2.66 |
| DIM-4.6B-T2I | 4.09 | 3.47 | 2.30 | 4.00 | 3.43 | 3.87 | 4.92 | 2.85 | 4.08 | 3.67 |

**Necessity of DIM-T2I.** Our CoT-guided editing requires robust comprehension capabilities. We posit that T2I generation is simpler than editing and is better suited for fostering this capability. Rather than simultaneously tackling two challenging objectives, *i.e.,* complex instruction comprehension and image editing, we chose to establish strong instruction comprehension first in the T2I stage. To justify our assumption, we trained a model exclusively on DIM-Edit to test the feasibility of simultaneously achieving modality alignment, complex instruction comprehension, and editing capabilities in a single stage. As evident from the Table 10, the model trained from scratch significantly underperforms the version initialized with DIM-4.6B-T2I. This performance gap empirically validates the necessity of DIM-T2I as a foundational cornerstone for the Draw-In-Mind paradigm.

## 5 CONCLUSION

In this paper, we identify a crucial issue in existing image editing models, *i.e., imbalanced division of responsibilities*, where the generator is burdened with complex reasoning, leading to reduced performance. To address this, we propose the *Draw-In-Mind* (DIM) dataset, consisting of two parts: (**i**) DIM-T2I, 14M web-crawled image-text pairs with carefully crafted long-context prompts that provide a foundation for complex CoT comprehension in editing; and (**ii**) DIM-Edit, 233K high-quality image editing pairs with detailed and precise CoT imagination. By training on the DIM dataset and incorporating an external designer during editing, we present DIM-4.6B-Edit, which achieves SOTA or competitive performance on ImgEdit and GEdit-Bench-EN while maintaining a tiny overall and trainable parameter size. These results validate our motivation to shift the design responsibility from the generation module to the understanding module, as well as the high efficiency of our proposed CoT-guided DIM dataset.

ETHICS STATEMENT

All authors of this paper strictly adhere to the ICLR Code of Ethics. The proposed image-text pairs in DIM-T2I have undergone a rigorous safety check to filter harmful content, *e.g.,* NSFW images. The image pairs in DIM-Edit are collected from publicly available datasets, *i.e.,* UltraEdit (Zhao et al., 2024), MagicBrush (Zhang et al., 2023), SEED-Data-Edit-Part3 (Ge et al., 2024), and ShareGPT-4o-Image (Chen et al., 2025b), without introducing new content that may raise ethical concerns. The CoTs generated by GPT-4o were subjected to both OpenAI's internal safety mechanisms and an additional safety check by the authors, confirming that no harmful content is present. Therefore, the training process and the trained models do not introduce ethical issues.

REPRODUCIBILITY STATEMENT

The authors take full responsibility for the reproducibility of this work. For the proposed DIM dataset, we provide a detailed data creation pipeline in Section 3.1 and Appendix D, including data sourcing and processing. The prompts used for image annotation are presented in Appendix F. For the DIM-4.6B-T2I/Edit models, we describe their architectures in detail in Section 3.2. In addition, we specify our training configurations and evaluation setup in Section 4.1. To facilitate reproducibility, we have released the DIM dataset, the DIM-4.6B-T2I/Edit models, and the related code to the public at `https://github.com/showlab/DIM`.

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

## A    MORE DISCUSSION ABOUT PRIOR WORKS

Beyond the latest works mentioned in Section 2.2 (Lin et al., 2025; Liu et al., 2025), there have been several attempts to leverage MLLMs to guide image editing from a *modeling perspective*. For example, MGIE (Fu et al., 2023) tunes the Text Embeddings/Image Tokenizer (Adapter)/LM head to produce concise instructions (*e.g.,* 22.7 tokens) and leverages N learnable [IMG] tokens to query latent visual imaginations from the instruction. These [IMG] tokens are subsequently transferred to L learnable queries through a 4-layer encoder-decoder transformer (connector) to act as editing conditions; Kosmos-G (Pan et al., 2023) adopts a resampler to transfer a single image into several query tokens. They first align these query tokens with the inherent MLLM space by fine-tuning the entire MLLM. Later, a deeper alignment is conducted through a tailored AlignerNet (connector) to transfer multimodal instructions into the generation space; MIGE (Tian et al., 2025) adopts a Q-Former to translate an image into 32 queries. These queries later attend to VAE features from the same image to retain low-level details. The final multimodal sequence is fed to an LLM for fine-tuning before acting as conditions for generation.

Our work serves as a distinct complement to these approaches. While they focus on architectural modifications, we approach the problem from the *data perspective*, yielding the following unique characteristics: (**i**) *Native Arbitrary-Length Multimodal Condition.* All the aforementioned works leverage a finite number of learnable queries to transfer the visual modality into the generation condition space (*e.g.,* MGIE's [IMG] tokens and transferred queries, Kosmos-G's resampler, and MIGE's Q-Former). This inevitably leads to information loss due to the limited representation space of learnable queries. In contrast, our DIM-4.6B-Edit retains full, native MLLM tokens, where images are deliberately tokenized by advanced MLLM tokenizers with minimal information loss. (**ii**) *Minimal Condition Transfer Requirement.* From a data perspective, we demonstrate that by carefully curating high-quality, long, and complicated instructions (DIM-T2I and DIM-Edit), we can achieve high editing performance using only a simple two-layer MLP and frozen MLLM native tokens. This eliminates the need for sophisticated connector designs and avoids the risks associated with unstable and costly MLLM/connector training. (**iii**) *Generalizability to Various MLLMs.* We demonstrate that our DIM dataset provides high generalizability regarding the core MLLM, evidenced by stable editing performance across different open- and closed-source designers. This offers the flexibility to seamlessly upgrade to more advanced MLLMs to achieve better performance in real-world deployments. Conversely, the works mentioned above require training the LLM (and other bottom components) to some extent, making them less flexible regarding model upgrades in practical applications.

We believe the referenced works are highly insightful from the modeling side; however, they are constrained by the lack of high-quality, complex editing blueprints that align with actual editing behaviors. Consequently, they must make compromises, such as reducing condition length, compressing images via learnable queries, designing tailored connectors, and requiring MLLM tuning, to facilitate training. We have made an early attempt to fully unleash the power of native UMM tokens, supporting ultra-long and detailed CoT blueprints. This effectively relieves the painter from the designer role, thereby achieving better edits. Our work offers a robust solution from the data side, effectively complementing the current research field.

## B    ADDITIONAL EXPERIMENTS

**Detailed Performance on GEdit-Bench-EN.** Table 11 and 12 summarize overall and detailed task-wise performance of different models on GEdit-Bench-EN, respectively. Our DIM-4.6B-Edit ranks just behind the in-domain tester, *i.e.*, Step1X-Edit, while surpassing all other out-of-domain competitors. Moreover, among out-of-domain testers, DIM-4.6B-Edit is the only model that consistently preserves both semantic consistency and perceptual quality. This demonstrates the effectiveness of DIM-Edit, where edits with high perceptual fidelity are precisely aligned with CoT-style imagination, thereby ensuring semantic correctness.

**Inference Efficiency.** Beyond precise image editing, our DIM-4.6B-Edit also maintains highly efficient inference inherited from the SANA architecture. To verify this, we compare the average editing time over 100 samples between Step1X-Edit and DIM-4.6B-Edit, as reported in Table 13. Specifically, Step1X-Edit is provided with short raw prompts, while DIM-4.6B-Edit is evaluated with longer CoT prompts. Even under this more demanding setting, our model achieves a $4.5\times$ speedup while preserving high editing quality, highlighting the effectiveness of the proposed DIM dataset and the Draw-In-Mind paradigm.

Table 11: The overall image editing performance on **GEdit-Bench-EN**. We use GPT-4.1 for evaluation to ensure consistency with the existing results reported in Step1X-Edit. ∗ indicates results evaluated by us. SC and PQ denote Semantic Consistency and Perceptual Quality, respectively.

| Model | Intersection subset | | | Full set | | |
|---|---|---|---|---|---|---|
| | SC | PQ | Overall | SC | PQ | Overall |
| *Proprietary Models* | | | | | | |
| Gemini (Comanici et al., 2025) | 6.82 | 7.41 | 6.48 | 6.87 | 7.44 | 6.51 |
| GPT-4o (OpenAI, 2025) | 7.40 | 7.90 | 7.14 | 7.22 | 7.89 | 6.98 |
| Doubao (Gong et al., 2025) | 7.87 | 8.10 | 7.59 | 7.74 | 8.13 | 7.49 |
| *Open-Source Models* | | | | | | |
| Instruct-P2P (Brooks et al., 2023) | 3.34 | 6.21 | 3.23 | 3.30 | 6.19 | 3.22 |
| MagicBrush (Zhang et al., 2023) | 4.56 | 6.34 | 4.24 | 4.52 | 6.37 | 4.19 |
| AnyEdit (Yu et al., 2025) | 3.12 | 5.87 | 2.92 | 3.05 | 5.88 | 2.85 |
| OmniGen (Xiao et al., 2025) | 6.04 | 5.86 | 5.15 | 5.88 | 5.87 | 5.01 |
| UniWorld-V1 (Lin et al., 2025) | - | - | - | 4.93 | 7.43 | 4.85 |
| Janus-4o* (Chen et al., 2025b) | 4.69 | 4.68 | 3.91 | 4.64 | 4.57 | 3.83 |
| Step1X-Edit (Liu et al., 2025) | 7.29 | 6.96 | 6.62 | 7.13 | 7.00 | 6.44 |
| DIM-4.6B-Edit | 6.91 | 6.90 | 6.46 | 6.65 | 6.71 | 6.18 |

Table 12: The detailed task-wise performance on **GEdit-Bench-EN** Full set. ∗ indicates results evaluated by us. Task abbreviations: Background Change (BC), Color Alter (CA), Material Alter (MA), Motion Change (MC), PS Human (PH), Style Change (SC), Subject-Add (SA), Subject-Remove (SRM), Subject-Replace (SRP), Text Change (TC), Tone Transfer (TT), and Average (AVG).

| Model | BC | CA | MA | MC | PH | SC | SA | SRM | SRP | TC | TT | AVG | AVG w/o TC |
|---|---|---|---|---|---|---|---|---|---|---|---|---|---|
| *Semantic Consistency* | | | | | | | | | | | | | |
| UniWorld-V1 | 5.17 | 7.21 | 4.71 | 1.14 | 3.49 | 5.98 | 7.42 | 6.50 | 6.04 | 1.07 | 5.52 | 4.93 | 5.32 |
| Janus-4o* | 5.48 | 6.68 | 6.00 | 2.75 | 4.04 | 8.03 | 5.10 | 1.74 | 4.27 | 2.11 | 4.88 | 4.64 | 4.90 |
| Step1X-Edit | 8.40 | 7.68 | 7.95 | 3.40 | 5.06 | 8.13 | 7.92 | 6.88 | 8.27 | 7.72 | 7.05 | 7.13 | 7.07 |
| DIM-4.6B-Edit | 7.68 | 7.65 | 7.48 | 4.78 | 5.64 | 8.22 | 8.10 | 7.05 | 7.45 | 2.34 | 6.73 | 6.65 | 7.08 |
| *Perceptual Quality* | | | | | | | | | | | | | |
| UniWorld-V1 | 7.59 | 6.82 | 6.86 | 8.68 | 8.61 | 6.58 | 7.61 | 7.28 | 6.78 | 7.44 | 7.48 | 7.43 | 7.43 |
| Janus-4o* | 4.00 | 4.20 | 4.08 | 5.73 | 6.07 | 4.40 | 4.77 | 4.07 | 4.72 | 4.44 | 3.78 | 4.57 | 4.58 |
| Step1X-Edit | 6.40 | 6.10 | 5.60 | 7.63 | 8.31 | 6.75 | 7.27 | 7.49 | 6.85 | 7.86 | 6.73 | 7.00 | 6.91 |
| DIM-4.6B-Edit | 6.73 | 6.55 | 5.13 | 7.15 | 7.43 | 6.53 | 7.28 | 6.83 | 6.65 | 6.61 | 6.88 | 6.71 | 6.71 |
| *Overall* | | | | | | | | | | | | | |
| UniWorld-V1 | 4.92 | 6.37 | 4.79 | 1.85 | 4.03 | 5.64 | 7.23 | 6.17 | 5.70 | 1.15 | 5.54 | 4.85 | 5.22 |
| Janus-4o* | 4.31 | 5.02 | 4.41 | 2.71 | 4.09 | 5.80 | 4.07 | 1.69 | 3.69 | 2.35 | 3.96 | 3.83 | 3.97 |
| Step1X-Edit | 7.03 | 6.26 | 6.46 | 3.66 | 5.23 | 7.24 | 7.17 | 6.42 | 7.39 | 7.40 | 6.62 | 6.44 | 6.35 |
| DIM-4.6B-Edit | 7.02 | 6.81 | 6.00 | 4.67 | 5.88 | 7.16 | 7.48 | 6.67 | 6.76 | 2.99 | 6.55 | 6.18 | 6.50 |

**Impact of DIM CoT for Different Models.** To investigate the impact of DIM-style CoT on different models, we evaluated the performance of Janus-4o and Step1X-Edit when directly provided with the same CoT blueprints as input instructions on ImgEdit. The results are presented in Table 14. Based on these results, we have the following observations and analysis:

- DIM-4.6B-Edit is explicitly trained on complex CoT-style blueprints from the DIM-Edit dataset, it achieves superior CoT comprehension. Consequently, it demonstrates substantial performance gains when DIM-style CoTs are applied during inference.

- Janus-4o employs an end-to-end fine-tuning approach, which minimizes the gap between instruction understanding and generation. This makes it more robust to input distribution shifts. While it possesses mild CoT comprehension capabilities and benefits slightly from DIM-style CoTs, the performance gain is less pronounced compared to DIM-4.6B-Edit.

- Step1X-Edit adopts a training recipe similar to ours (using a frozen understanding core), this design makes it susceptible to input distribution shifts when facing unseen instruction formats. It struggles to process CoT inputs effectively, leading to performance degradation when DIM-style CoTs are applied.

Table 13: The generation configuration and inference speed of Step1X-Edit and DIM-4.6B-Edit.

| Model | Gen. Resolution | Gen. Steps | Und. Params | Gen. Params | VAE Rate | Prompt | Speed |
|---|---|---|---|---|---|---|---|
| Step1X-Edit | 1024×1024 | 30 | 7B | 12.5B | 8× | Raw | 28.19s |
| DIM-4.6B-Edit | | | 3B | 1.6B | 32× | CoT | 6.23s |

Table 14: The **ImgEdit** performance of different models with/without using DIM CoT as instruction.

| Model | Params | CoT | Add | Adjust | Extract | Replace | Remove | Background | Style | Hybrid | Action | Overall |
|---|---|---|---|---|---|---|---|---|---|---|---|---|
| DIM-4.6B-Edit | Und❄ \| Gen🔥 | ✗ | 3.53 | 3.23 | 2.01 | 3.49 | 1.47 | 3.42 | 4.79 | 2.35 | 3.64 | 3.10 |
| | | ✔ | 4.09 | 3.47 | 2.30 | 4.00 | 3.43 | 3.87 | 4.92 | 2.85 | 4.08 | 3.67 |
| Janus-4o | Und🔥 \| Gen🔥 | ✗ | 3.35 | 3.35 | 2.25 | 3.01 | 2.18 | 3.32 | 4.71 | 2.49 | 4.04 | 3.19 |
| | | ✔ | 3.95 | 2.74 | 2.49 | 3.59 | 2.28 | 3.31 | 4.72 | 2.62 | 4.02 | 3.30 |
| Step1X-Edit | Und❄ \| Gen🔥 | ✗ | 3.88 | 3.14 | 1.76 | 3.40 | 2.41 | 3.16 | 4.63 | 2.64 | 2.52 | 3.06 |
| | | ✔ | 3.56 | 2.47 | 1.81 | 3.13 | 2.02 | 2.84 | 4.18 | 1.80 | 2.46 | 2.70 |

Table 15: The performance of DIM-4.6B-T2I/Edit on understanding benchmarks.

| Model | Params | MME-P | MMB | SEED | MMMU | MM-Vet |
|---|---|---|---|---|---|---|
| Janus (Wu et al., 2025a) | 1.3B🔥 | 1338.0 | 69.4 | 63.7 | 30.5 | 34.3 |
| Emu3-Gen (Wang et al., 2024b) | 8.0B🔥 | - | 58.5 | 68.2 | 31.6 | 37.2 |
| Show-o (Xie et al., 2024) | 1.3B🔥 | 1097.2 | - | - | 26.7 | - |
| Show-o2-7B (Xie et al., 2025b) | 7.0B🔥 | 1620.5 | 79.3 | 69.8 | 48.9 | - |
| Janus-Pro-7B (Chen et al., 2025c) | 7.0B🔥 | 1567.1 | 79.2 | 72.1 | 41.0 | 50.0 |
| BAGEL (Deng et al., 2025) | 14.0B🔥 | 1687.0 | 85.0 | - | 55.3 | 67.2 |
| MetaQuery-L (Pan et al., 2025) | 3.0B❄ \| 3.2B🔥 | 1574.3 | 78.6 | 73.8 | 53.1 | 63.2 |
| DIM-4.6B-T2I/Edit | 3.0B❄ \| 1.6B🔥 | 1574.3 | 78.6 | 73.8 | 53.1 | 63.2 |

Based on these findings, we conclude that *superior CoT comprehension is pivotal for enhancing editing performance.* This finding validates our strategy of fostering CoT comprehension by constructing DIM-T2I and utilizing DIM-4.6B-T2I as the initialization for the editing task.

**Understanding Performance.** Since the MLLM component is frozen during DIM training, its understanding performance remains unaffected and is identical to the results reported in the original paper. To ensure clarity regarding the model's capabilities, we report the corresponding understanding performance in Table 15. Our experiments demonstrate that DIM-4.6B-Edit achieves superior editing performance even when utilizing a relatively small MLLM under a frozen setting. *This finding highlights the flexibility of our approach: users can seamlessly upgrade to advanced MLLMs to unlock even greater understanding and editing performance. Such integration is straightforward, as our streamlined architecture and training recipe avoid the need for intricate parameter tuning.*

# C  ADDITIONAL VISUALIZATIONS

## C.1  VISUALIZATION OF DIFFERENT EDITING OPERATIONS.

Beyond Figure 3 in the manuscript, we further visualize the edits of Janus-4o, Step1X-Edit, and our DIM-4.6B-Edit under the operations of *add*, *change*, *remove*, *replace*, and *style transfer* in Figure 4, 5, 6, 7, and 8, respectively. As shown, DIM-4.6B-Edit consistently preserves the overall layout while performing natural edits. For instance, in Figure 4, Janus-4o fails to generate details of the wooden cabin, while Step1X-Edit places the chimney on the river, which is counterfactual. In contrast, our DIM-4.6B-Edit carefully adds the wooden cabin while ensuring naturalness. In Figure 5, Janus-4o fails to follow the color change instruction. Step1X-Edit changes the singer's shirt to blue but also alters fine details such as the collar shape. By comparison, our DIM-4.6B-Edit changes the shirt to red while preserving all details, including the shadow cast by the hand. In Figure 6, both DIM-4.6B-Edit and Step1X-Edit perform successful removals, whereas Janus-4o fails to remove the seaplane. In Figure 7, only DIM-4.6B-Edit captures the semantics of "majestically" and generates a roaring lion. Finally, in Figure 8, although all three models succeed in style transfer, only DIM-4.6B-Edit captures subtle visual cues, such as the green grass in the last row, and repaints them faithfully in the edits.

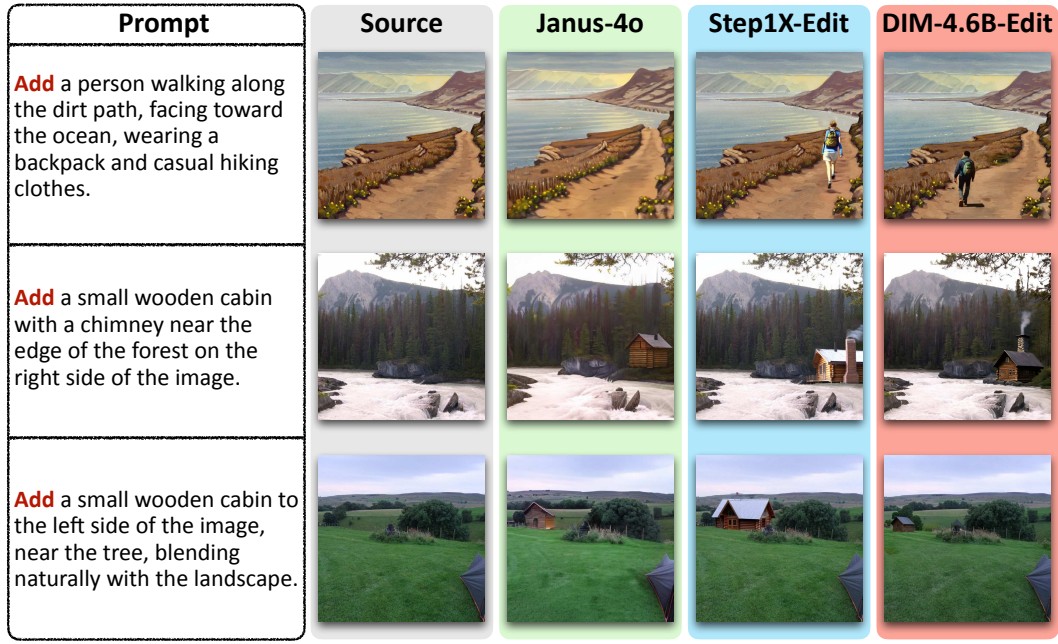

Figure 4: The edits of **Janus-4o** , **Step1X-Edit** , and **DIM-4.6B-Edit** for the *add* operation.

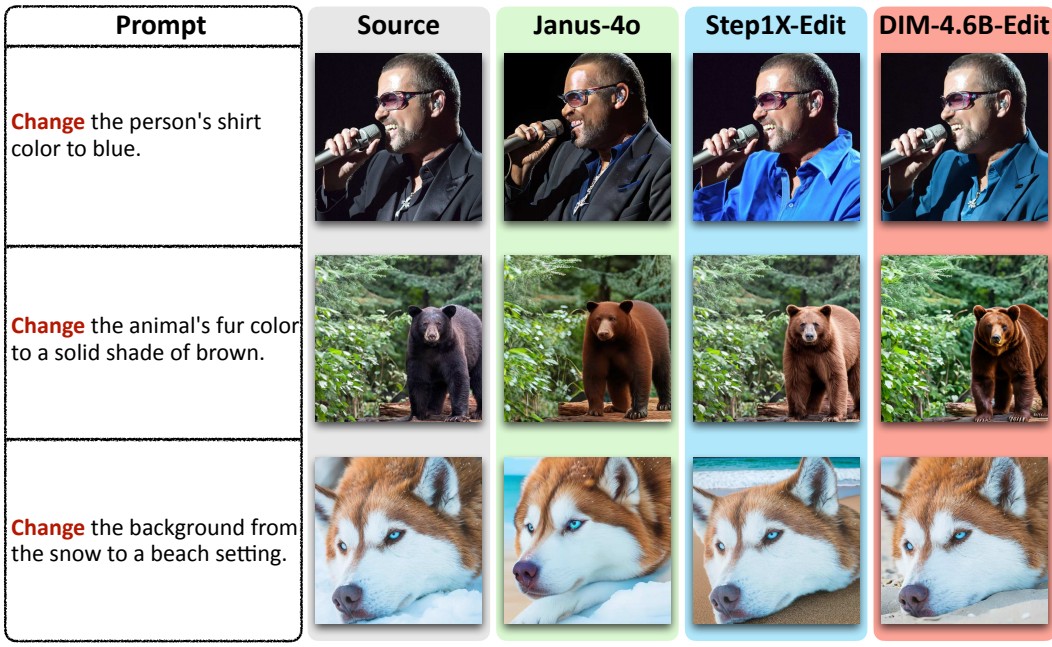

Figure 5: The edits of **Janus-4o** , **Step1X-Edit** , and **DIM-4.6B-Edit** for the *change* operation.

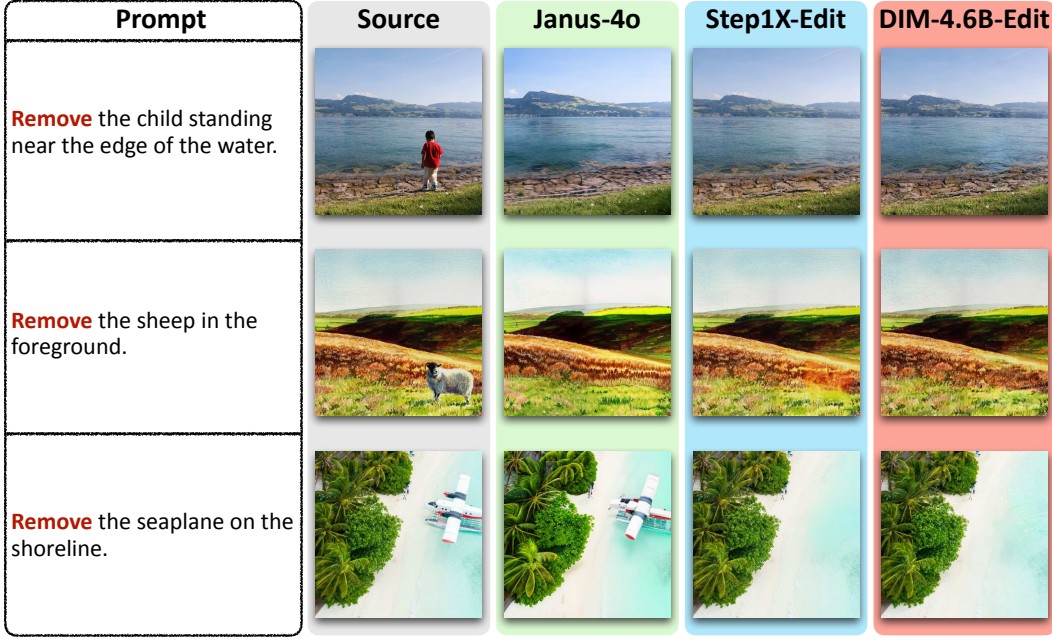

Figure 6: The edits of Janus-4o , Step1X-Edit , and DIM-4.6B-Edit for the *remove* operation.

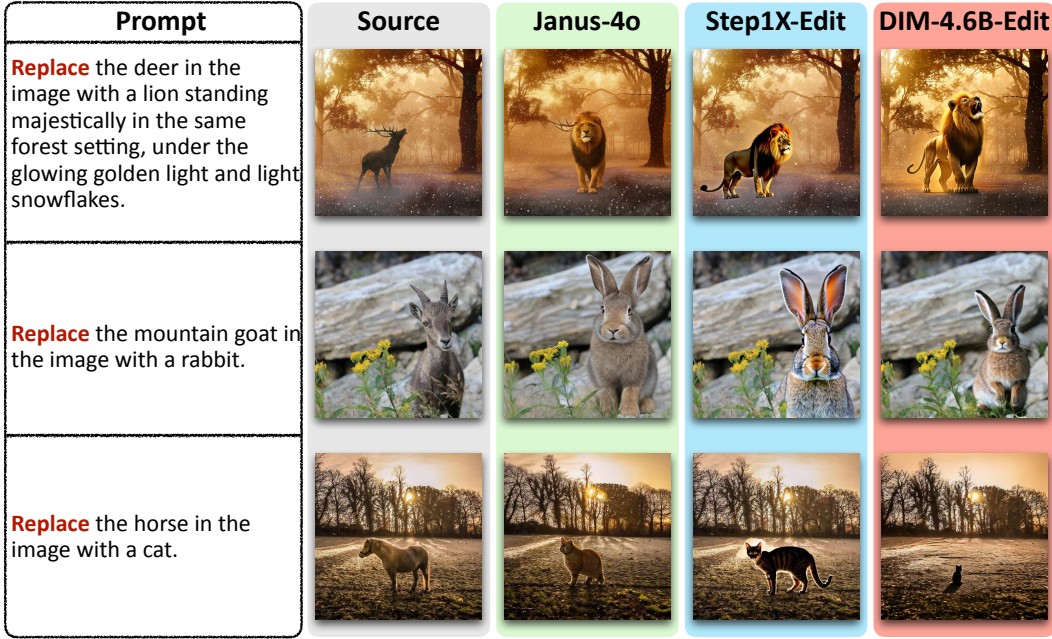

Figure 7: The edits of Janus-4o , Step1X-Edit , and DIM-4.6B-Edit for the *replace* operation.

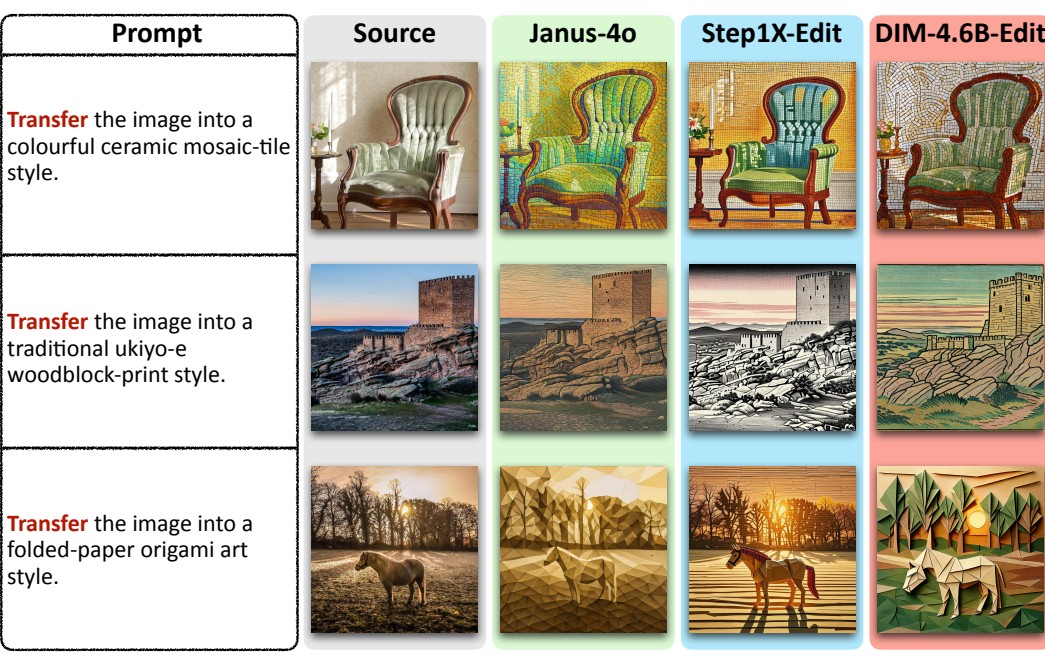

| Prompt | Source | Janus-4o | Step1X-Edit | DIM-4.6B-Edit |
|---|---|---|---|---|
| **Transfer** the image into a colourful ceramic mosaic-tile style. | | | | |
| **Transfer** the image into a traditional ukiyo-e woodblock-print style. | | | | |
| **Transfer** the image into a folded-paper origami art style. | | | | |

Figure 8: The edits of **Janus-4o** , **Step1X-Edit** , and **DIM-4.6B-Edit** for *style transfer*.

### C.2 VISUALIZATION OF THE DRAW-IN-MIND WORKFLOW'S IMPACT ON IMAGE EDITING.

Relying solely on numerical metrics may not intuitively convey the practical impact of the Draw-In-Mind workflow on image generation. To address this, we present Figure 9, 10, 11, 12, and 13 to showcase several advanced usage scenarios. These examples demonstrate complex cases that are successfully handled by DIM-Edit-4.6B, highlighting capabilities that remain beyond the reach of current baseline methods.

**Instruction Disambiguation.** In Figure 9, the user instruction presents an inherent ambiguity due to the presence of three lemons on the table. This task necessitates precise multi-object localization and removal, which is a challenge that proves difficult without the Draw-In-Mind paradigm, as standard models often struggle with the required multi-object reasoning. Consequently, both the 7B Janus-4o and 12B Step1X-Edit fail to execute the edit correctly. Similarly, when CoT is disabled, our DIM-4.6B-Edit also fails to remove all targets. However, with DIM CoT enabled, the generated design blueprints effectively disambiguate the instruction. They accurately localize the three lemons to the right of the vase and ensure their complete removal, while perfectly preserving the integrity of the unedited regions.

**Edit Navigation and Structural Planning.** In Figure 10, the user instruction presents two distinct challenges: (**i**) determining the optimal placement for a wooden cabin, and (**ii**) identifying the appropriate structural integration for a chimney. These dual requirements impose a significant burden on the generation model. Consequently, in Janus-4o's output, the chimney is nearly invisible, while Step1X-Edit places the cabin counterintuitively close to the river. Similarly, DIM without CoT fails to simultaneously resolve the cabin placement and chimney addition. In contrast, DIM powered by CoT effectively navigates these challenges. It observes that "the trees thin out on the right side" (GLP) and selects this area as the optimal location (EAL). It then explicitly envisions the cabin's appearance, including a chimney emitting smoke (EII), ultimately yielding the most plausible and high-quality edit among all competitors.

**Commonsense-guarded Editing.** In Figure 11, the editing task presents a subtle complexity: it requires commonsense reasoning regarding scale. From the same viewpoint, a cat should appear significantly smaller than a horse. All baseline models, including our own DIM w/o CoT, overlook this physical constraint, simply replacing the horse with a cat of identical dimensions. In contrast, DIM with CoT successfully leverages commonsense reasoning. It recognizes the size discrepancy and executes a "commonsense-guarded" edit, placing a naturally scaled cat at the target location, thereby preserving scene realism.

**Advanced Causal Editing.** In Figure 12, we present an advanced causal editing scenario where the instruction implies the target quantity (referencing "the second prime number") rather than stating it explicitly. Unsurprisingly, all baseline models fail to resolve this implicit requirement. In contrast, DIM with CoT swiftly infers the correct number of cherries and executes a successful edit, demonstrating its ability to handle knowledge-intensive instructions.

**Advanced Temporal Editing.** Figure 13 illustrates the most complex temporal editing scenario, which necessitates a deep understanding of chemical reaction dynamics. Similar to the previous example, none of the baseline models succeed in this task. In contrast, DIM with CoT accurately characterizes the reaction process and executes physically plausible edits, demonstrating its capability to handle sophisticated temporal reasoning.

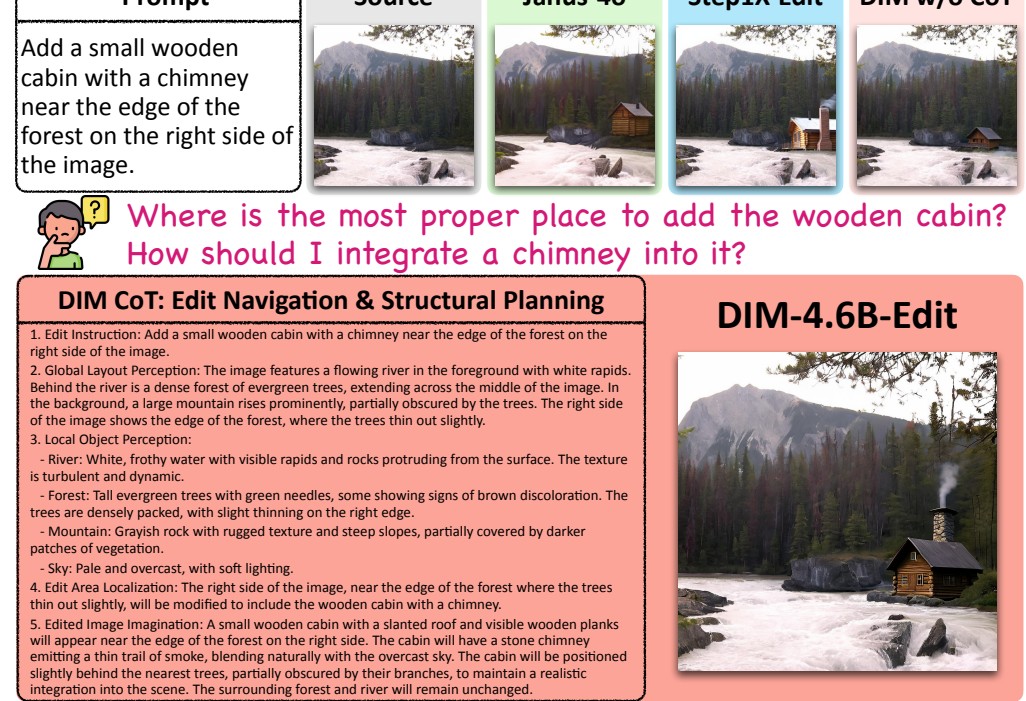

Figure 9: The edits of **Janus-4o** , **Step1X-Edit** , **DIM w/o CoT** , and **DIM-4.6B-Edit** when the user instruction is ambiguous. DIM CoT is capable of *instruction disambiguation* under this case.

Figure 10: The edits of **Janus-4o** , **Step1X-Edit** , **DIM w/o CoT** , and **DIM-4.6B-Edit** when the user instruction requires localization and involves fine-grained structure modification. DIM CoT is capable of *edit navigation and structural planning* under this case.

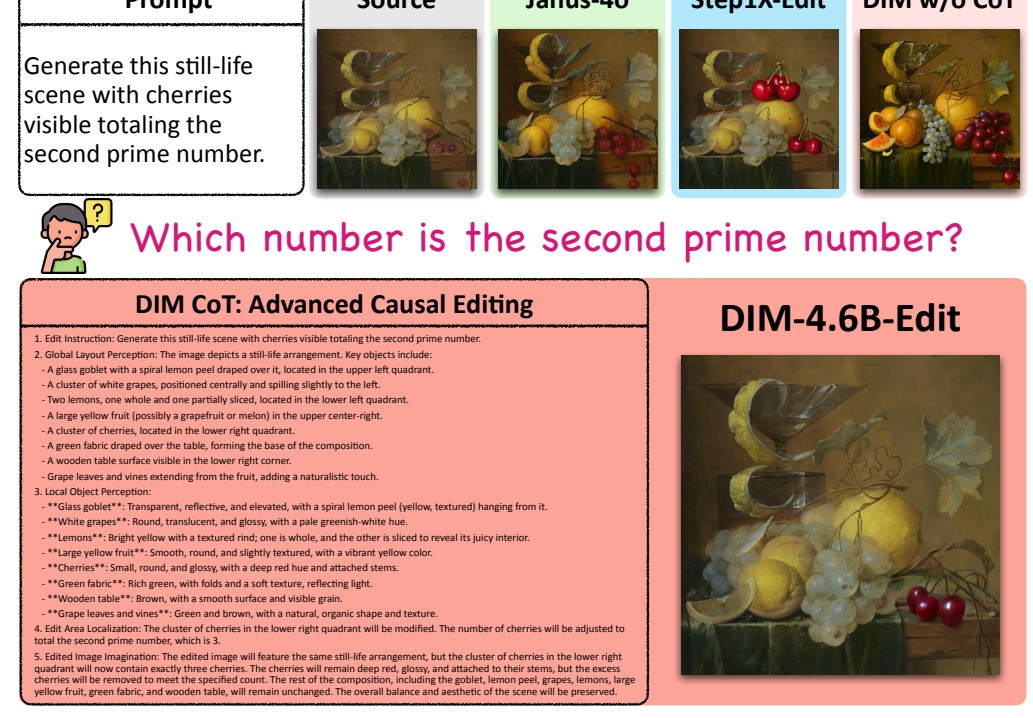

Figure 11: The edits of Janus-4o , Step1X-Edit , DIM w/o CoT , and DIM-4.6B-Edit when the user instruction involves implicit commonsense constraint. DIM CoT is capable of *commonsense-guarded editing* under this case.

Figure 12: The edits of Janus-4o , Step1X-Edit , DIM w/o CoT , and DIM-4.6B-Edit when the user instruction requires causal reasoning. DIM CoT is capable of *advanced causal editing* under this case.

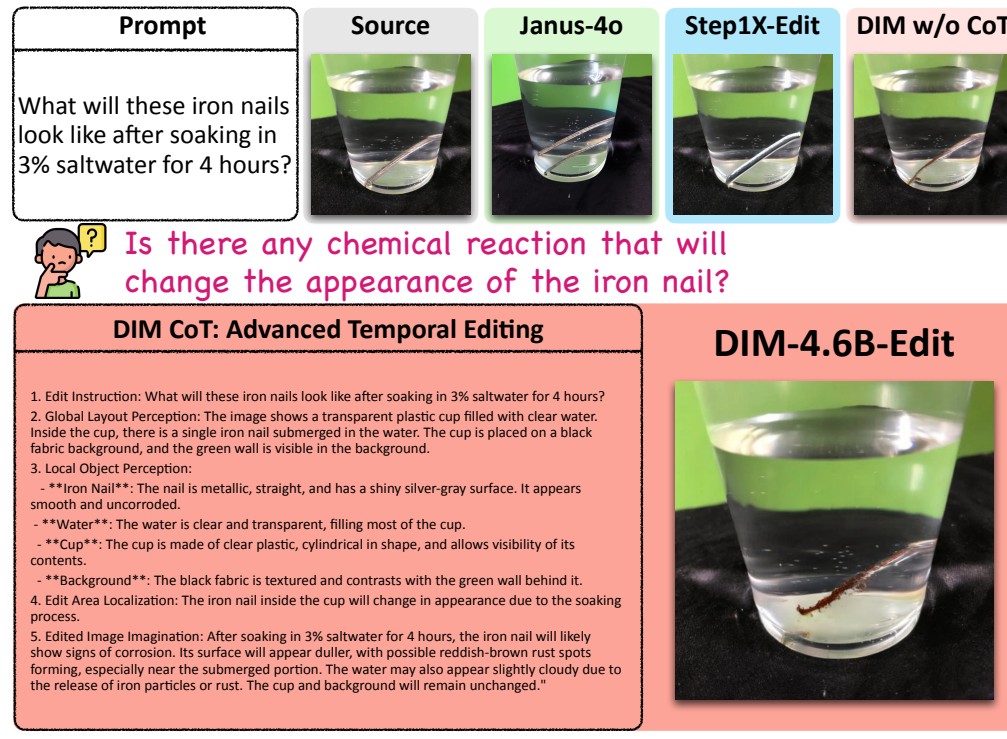

Figure 13: The edits of **Janus-4o** , **Step1X-Edit** , **DIM w/o CoT** , and **DIM-4.6B-Edit** when the user instruction requires temporal reasoning. DIM CoT is capable of *advanced temporal editing* under this case.

### C.3 VISUALIZATION OF FAILURE CASES

We are also open to discuss the limitations of our work, and provide three failure types with six specific cases in Figure 14, 15, and 16 to intuitively show the boundaries of our DIM-4.6B-Edit.

**Large-scale All-in-One Editing.** In Figure 14, the instructions involve simultaneous multi-step edits, a task that remains essentially challenging for almost all editing models, and one where DIM-Edit also encounters difficulties.

- For the first case, Janus-4o and Step1X-Edit completely fail to follow the physical laws dictating that the wooden tower should collapse. Our DIM-4.6B-Edit successfully imitates a scene of imminent collapse; however, it fails to preserve the exact appearance of the individual wooden blocks, as too many objects are involved in the manipulation.
- For the second case, Janus-4o and Step1X-Edit fail to change the view at all. While our DIM-4.6B-Edit completes the primary editing task, some fine-grained details are distorted (e.g., the window of the shoreside house is missing).

**Text and Logic Editing.** In Figure 15, where instructions involve complex text rendering and logical editing, DIM-4.6B-Edit struggles due to a combination of data scarcity and inherent VAE compression issues.

- For the first case, the use of SANA1.5's VAE with a 32x downsampling rate makes complex text rendering particularly challenging, a difficulty exacerbated by the lack of targeted training data. In contrast, Step1X-Edit employs an 8x downsampling VAE and is trained on proprietary, text-specific in-house data, allowing it to perform relatively well. We regard this as a necessary trade-off between efficiency and rendering quality: as shown in Table 10, DIM-4.6B-Edit requires only 6 seconds to complete an edit with a 200+ word CoT, whereas Step1X-Edit takes 28 seconds with a short raw prompt.
- For the second case, all editing models fail. This is fundamentally because none of the models, including DIM-4.6B-Edit, are specifically trained on geometric data. The underlying painter struggles to even draw these shapes, let alone edit them. We believe crafting such datasets remains a valuable and under-explored topic for future research.

**Reference-free Editing (in Pixel Space).** In Figure 16, the reference image does not provide a strong pixel constraint for the target image. Consequently, this task resembles multimodal generation rather than strict editing. All models fail here because existing editing architectures typically enforce strong pixel alignment with the source image.

- For the first case, which requests a view of the Golden Gate Bridge, Janus-4o and Step1X-Edit are completely ineffective. DIM-4.6B-Edit struggles to break free from the structural constraints of the reference image, resulting in a "scratchy" and distorted view that fails to meet the objective.
- For the second case, where the task involves a re-imagination of the source scene, Janus-4o produces a black-and-white edit, and Step1X-Edit fails completely. DIM-4.6B-Edit generates the most plausible result, successfully covering the scene with white snow. However, because the transformation fundamentally alters the source structure, specific details such as the castle are inevitably distorted.

In summary, the majority of failure cases arise when the task necessitates either generating an image that diverges drastically from the source or rendering complex text and geometric shapes. Even in these challenging scenarios, DIM-4.6B-Edit demonstrates superior instruction-following capabilities compared to baseline models. These limitations highlight persistent challenges within the current landscape of open-source editing models. We suggest that future research directions, such as intelligent routing that dynamically selects between T2I generation and editing pipelines based on instruction intent, offer promising avenues for resolving these issues, though significant progress is still required in the field.

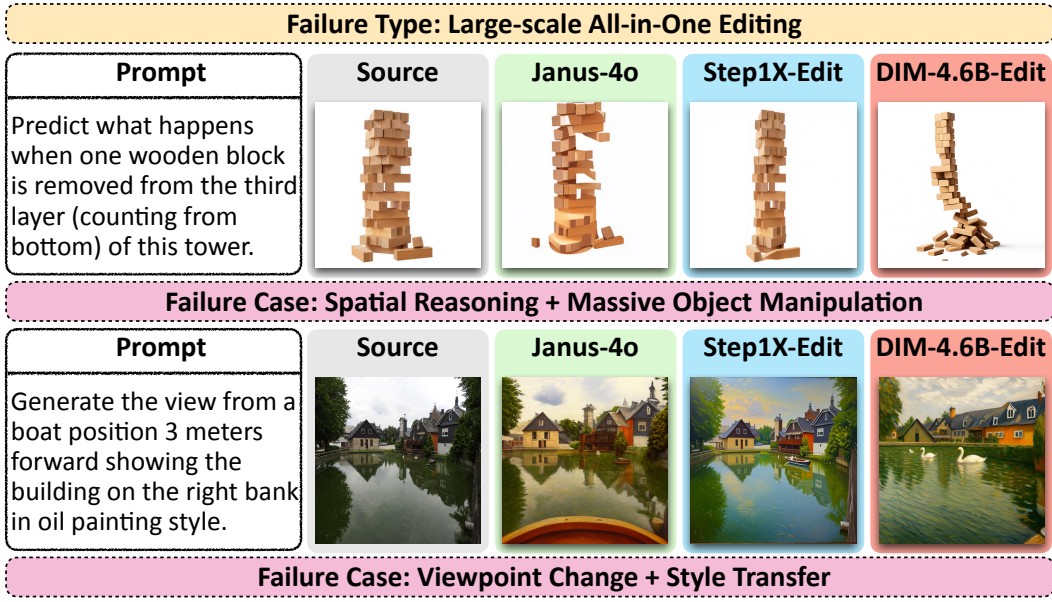

Figure 14: The edits of Janus-4o , Step1X-Edit , and DIM-4.6B-Edit for the failure type *large-scale all-in-one editing*.

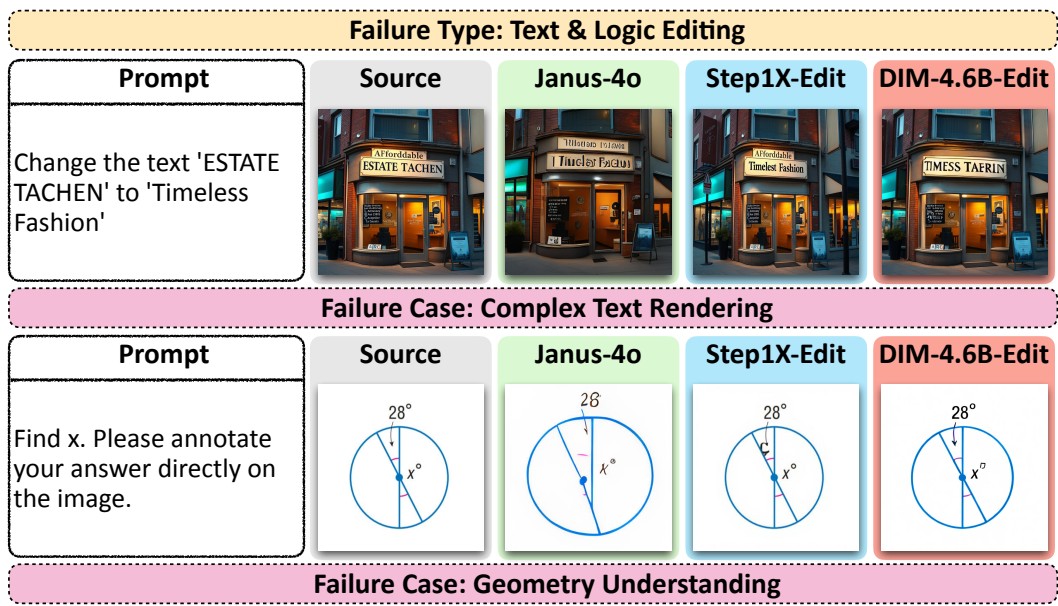

Figure 15: The edits of Janus-4o , Step1X-Edit , and DIM-4.6B-Edit for the failure type *text and logic editing*.

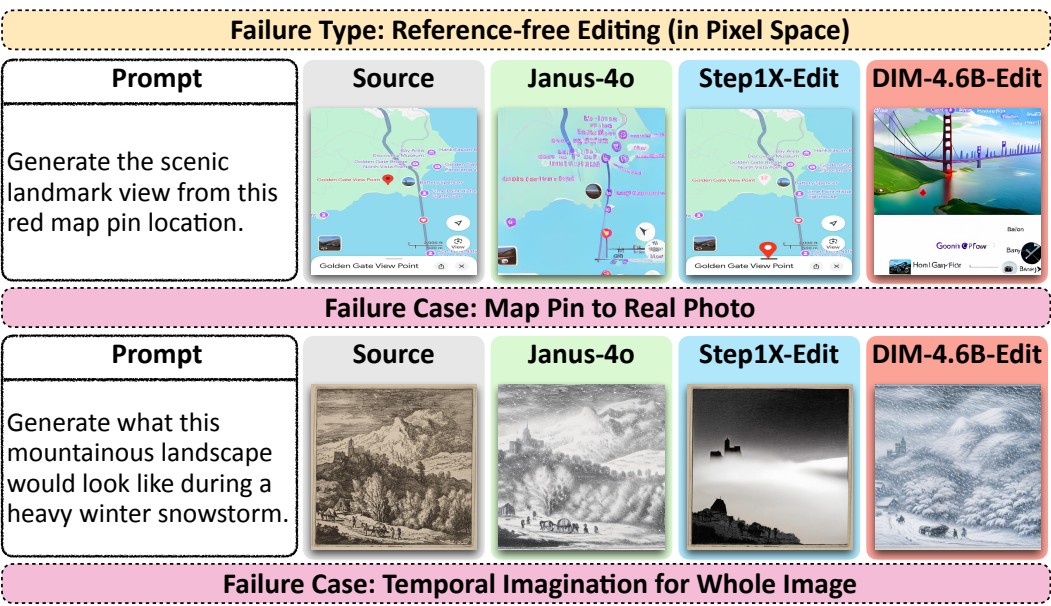

Figure 16: The edits of **Janus-4o**, **Step1X-Edit**, and **DIM-4.6B-Edit** for the failure type *reference-free editing (in pixel space)*.

## D  DIM-EDIT DATA COLLECTION PIPELINE

As stated in Section 3.1.2, we collect raw edit data from four publicly available datasets:

- **UltraEdit** (Zhao et al., 2024). In addition to the prompt quality evaluation and optimization in the DIM-Edit creation pipeline (Figure 2), which aligns textual prompts with actual editing behaviors, we employ three *image-to-image* metrics on the UltraEdit dataset to improve visual consistency and stabilize training: (**i**) CLIP image similarity, (**ii**) DINOv2 similarity, and (**iii**) SSIM. These metrics are used to select edit pairs that maintain consistent visual appearances. We retain only those edit pairs that satisfy the following conditions: (**i**) the CLIP similarity between the source and edited images is greater than 0.9; (**ii**) the DINOv2 similarity is greater than 0.9; (**iii**) the SSIM score is greater than 0.8; and (**iv**) the prompt does not contain "rainbow", since many edit pairs meeting (**i**)–(**iii**) are associated with low-quality "rainbow" edits. After filtering, we obtain roughly 160K edit pairs.
- **MagicBrush** (Zhang et al., 2023). We include only 8K images from the training set to avoid potential information leakage during evaluation.
- **SEED-Data-Edit-Part3** (Ge et al., 2024). Since the "remove" operation is absent in UltraEdit, we additionally select 19K edit pairs from SEED-Data-Edit-Part3 by filtering prompts that explicitly contain "remove."
- **ShareGPT-4o-Image** (Chen et al., 2025b). We include only its 46K image-to-image subset.

By combining these collected datasets, we obtain a total of 233K raw edit pairs for the proposed DIM-Edit.

## E  DIM-EDIT QUALITY ASSESSMENT

We further assess the quality of the CoTs in DIM-Edit through MLLM-powered validation. Specifically, due to API quota limitations, we randomly sample 30K edit pairs from DIM-Edit and use GPT-4.1 to evaluate the quality of the GPT-4o-annotated CoTs, categorizing them into four levels:

- **Low**: The optimized edit instruction does not reflect the change between the source and edited images at all.
- **Medium**: The optimized edit instruction captures the major change between the source and edited images, but the chain-of-thought contains some factual errors.
- **High**: The optimized edit instruction captures the major change between the source and edited images, and the chain-of-thought contains only minor factual errors.
- **Ultra-High**: The optimized edit instruction accurately captures all changes between the source and edited images, and the chain-of-thought contains no factual errors.

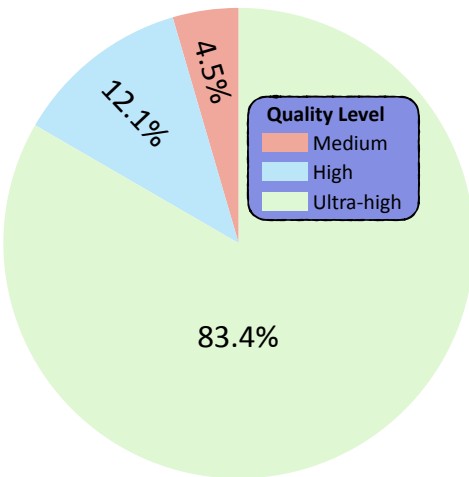

Figure 17: The percentage distribution of each quality level in DIM-Edit judged by GPT-4.1.

The percentage distribution of each quality level is shown in Figure 17. Notably, no data is categorized as "Low", while the majority falls under the "Ultra-High" level, demonstrating the strong overall quality of DIM-Edit.

We further conducted a human verification study. Specifically, we randomly sampled 25 instances from each of the data sources listed in Appendix D, resulting in a comprehensive evaluation set of 100 samples. Three human annotators were then recruited to assess the quality of the CoTs from two distinct perspectives:

## DIM-Edit CoT User Study

Figure 18: (**a**) The win rate of the optimized DIM-Edit instruction and the raw instruction. (**b**) The quality of the Edited Image Imagination (EII).

**Evaluation of Optimized Instructions (Start of CoT).** We presented annotators with both the raw instructions and the optimized instructions from DIM-Edit, alongside the corresponding source-edit image pairs. Annotators were tasked with selecting the instruction that best reflected the actual editing operations. A "Tied" option was included for cases where neither instruction was sufficiently accurate. The metric reported is the average win rate for each instruction type.

**Evaluation of Edited Image Imagination (End of CoT).** We asked annotators to assess the alignment between the Edited Image Imagination (EII) and the actual edited image. The quality was categorized into three levels: (**i**) Perfect Match, (**ii**) Minor Errors, and (**iii**) Major Errors. The metric reported is the percentage distribution across these error levels.

This efficient evaluation protocol enables a rapid yet robust assessment of the overall CoT quality within DIM-Edit. The results for both the instruction optimization (Win Rate) and the Edited Image Imagination (Error Distribution) are summarized in Figure 18, in which we have the following analysis:

- *Consistency with MLLM Assessment.* These results align closely with the MLLM-based quality assessment presented in Appendix D, where over 80% of DIM-Edit CoTs were judged clearer than the raw instructions, with no factual errors detected. Even in "Tied" cases where the optimization was not deemed strictly superior, the semantics of the raw instruction were fully preserved, ensuring that the optimization process introduces no regression.

- *Analysis of Minor Errors.* We observed that minor errors typically relate to subtle environmental inconsistencies, such as slight shifts in brightness (e.g., "the image should be a bit lighter"). These artifacts usually stem from the VAE's inability to perfectly reconstruct raw images in AI-generated pairs (e.g., from UltraEdit), leading to a slight loss of high-frequency features. As these discrepancies are barely perceptible to the human eye, they have a negligible impact on overall training efficiency.

- *Analysis of Major Errors.* Instances classified as having major errors generally correspond to extremely challenging scenarios where the edits are minute (e.g., the removed object occupies less than 2% of the pixels). These cases are difficult even for human annotators and advanced MLLMs like GPT-4o. Given their extreme rarity, these outliers do not adversely affect the stability of the training procedure.

Overall, the CoTs produced by our DIM-Edit pipeline maintain high quality and serve as effective design blueprints. This high data quality directly translates to better editing capabilities, as evidenced by the superior performance of the DIM-4.6B-Edit model trained on this dataset.

## F  DIM-T2I Analysis Dimensions

Figure 19 and 20 illustrate the 21 analysis dimensions and their corresponding prompts used in the DIM-T2I annotation process. The 21 dimensions were derived from a thorough literature review and an empirical analysis of existing understanding datasets and benchmarks. They are listed as follows:

MME (Fu et al., 2025), MMMU (Yue et al., 2024), MMMU-Pro (Yue et al., 2025), MMLU (Hendrycks et al., 2020), MMStar (Chen et al., 2024b), MMT-Bench (Ying et al., 2024), MM-Vet (Yu et al., 2023), MM-Vet V2 (Yu et al., 2024), LLaVA-Bench-Wild (Liu et al., 2023a), LLaVA-Bench-Wilder (Li et al., 2024a), WildVision (Lu et al., 2024), COCO (Lin et al., 2014), VQAv2 (Goyal et al., 2017), OK-VQA (Marino et al., 2019), TextCaps (Sidorov et al., 2020), TextVQA (Singh et al., 2019), AI2D (Kembhavi et al., 2016), ChartQA (Masry et al., 2022), DocVQA (Mathew et al., 2021), MathVista (Lu et al., 2023), MIA-Bench (Qian et al., 2024), MegaBench (Chen et al., 2024a), RWQA (xAI, 2024), OCRBench (Liu et al., 2023b), GSM8K (Cobbe et al., 2021), GPQA (Rein et al., 2024), IFEval (Zhou et al., 2023).

We believe that the aspects emphasized in widely recognized understanding datasets and benchmarks effectively capture the most frequent interactions between humans and objects in the real world. This makes them an ideal foundation for learning text-to-image generation tasks involving long and complex instructions. By constructing prompts that span these diverse fields, DIM-4.6B-T2I not only masters long-form instruction processing but also acquires the broad world knowledge necessary to facilitate sophisticated CoT comprehension and precise editing, thereby achieving high GenEval scores and low FID on MJHQ-30K.

## G  The Use of Large Language Models

This paper uses OpenAI ChatGPT solely for polishing the writing. The authors provided raw text to ChatGPT to correct grammatical errors and refine the statements into a more formal academic style. All polished text was manually reviewed and verified by the authors, who affirm that the paper contains no fabricated content. No statistical data were provided to ChatGPT. All numerical values in tables and figures were originally written by the authors.

| Dimension | Prompt |
|---|---|
| System Message | Please describe images in details, including but not limited to the user pre-defined dimensions. Please make sure your description is visually grounded for user provided image, namely the user can find visual cues in image for your generated image caption. The user pre-defined dimensions are: [DIMENSIONS] Please don't generate your response for each dimension, e.g., **something**, just give an overall image caption including all the dimensions. |
| Character Name | If a character is shown in the image, you must describe his/her names. The character include but not limited to the characters shown in Movies, TV Shows, Anime, Comics, Literature, Games, Virtual Idols/Characters. |
| Scene Description | Provide an overview of the image, identifying key objects, people, and any interactions. Clearly classify and describe each object (e.g., people, animals, buildings, plants). Specify their attributes, such as size, color, material, and texture. |
| Actions and Interactions | Describe any actions taking place in the image. Who is performing them, and how are they interacting with other objects or people? If there are dynamic elements (e.g., movement), detail their state (e.g., running, jumping, flying, waving). |
| Context and Environment | Describe the setting of the image, including the location (indoor or outdoor), time of day, weather, and any background elements (e.g., sky, buildings, roads). How does the environment contribute to the overall scene? Does the setting enhance the mood or theme? |
| Emotion and Sentiment | If people are present, describe their emotional states based on body language, facial expressions, and other visual cues. What mood or tone does the image convey (e.g., happiness, sadness, tension, peace)? How do these emotions connect to the scene? |
| Relationships and Spatial Arrangement | Explain how objects, people, and other elements are positioned in relation to one another (e.g., "next to," "above," "to the right of"). Consider foreground, background, and overall spatial composition. How does the positioning influence the overall visual balance or narrative? |
| Color and Texture | Describe the color palette of the image (e.g., colors of objects, background), and note any texture details (e.g., smooth, rough, soft). How do these color and texture choices contribute to the atmosphere or style of the image? |
| Symbolism or Abstract Interpretation | If relevant, interpret any symbolic or abstract elements within the image. What deeper meanings or metaphors can be inferred from the visual elements? How do these symbols tie into the image's broader themes or message? |
| Lighting and Shadows | Observe the lighting conditions in the image (e.g., sunlight, artificial light) and how shadows or reflections influence the objects' appearance. Note the intensity of the light and any patterns created by it. How do these lighting effects contribute to the mood or focal points of the image? |
| Details and Fine Elements | Focus on smaller, intricate details in the image (e.g., wrinkles in clothing, textures on surfaces, distinct features). These elements may carry significant meaning or help provide a more vivid, precise description. |

Figure 19: The 21 analysis dimensions and corresponding prompts for DIM-T2I.

| Dimension | Prompt |
|---|---|
| Perspective and Composition | Describe the viewpoint of the image (e.g., aerial, eye-level, side view) and its composition (e.g., symmetry, balance, focal point). How does the choice of perspective and composition affect the viewer's perception or interpretation of the scene? |
| Time and Season | If possible, infer the time of day or season based on visual cues (e.g., light quality, weather, clothing style). For example, a winter snow scene, a summer beach setting, or an autumn forest could suggest the specific season. |
| Target Audience | Consider if there's a specific target audience for the analysis. For instance, an analysis for an art historian might use more technical terms, while one for a general audience may keep the description simpler. Does the complexity of the image suggest it's meant for a particular demographic or purpose? |
| OCR | If text appears in the image, you must describe the text in its original language and provide an English translation in parentheses. For example: 书本 (book). Additionally, explain the meaning of the text within its context. |
| Person Description | If there are people in the image, describe their physical features (e.g. age, gender, hairstyle, clothing, etc.), their movements and expressions, and their relationship to the surrounding environment. If there is a single person, use 'he' or 'she' for reference instead of 'they'. |
| Mathematics | Analyze the image and describe the mathematical concepts it represents. Include specific details like geometric shapes, equations, numeric values, or relationships between elements. If the image includes a graph, describe its axes, scales, and key points. If applicable, explain how mathematical operations are visualized. |
| Information Extraction | Examine the image and extract textual and contextual information. If the image contains a document, transcribe its content accurately. For GUI or structured data, describe its layout, labels, and functionality. Summarize the core message or purpose of the content. |
| Planning | Identify the sequence or logical arrangement in the image. If it depicts a process, explain the steps and their correct order. For puzzles or games, provide the rules and possible solutions. |
| Science | Explain the scientific content or phenomenon depicted in the image. Provide details on experiments, natural phenomena, or theoretical concepts, including relevant terminology. |
| Perception | Provide a detailed perception-based description of the image. Identify objects, their attributes (color, shape, size), and spatial relationships. For specific tasks like facial analysis or pose estimation, include characteristics like expressions, poses, or physical traits. |
| Metrics | Evaluate the image based on predefined metrics. Assess its quality, authenticity, and adherence to caption content. For paper review or comparative tasks, provide constructive feedback or preference reasoning. |

Figure 20: The 21 analysis dimensions and corresponding prompts for DIM-T2I. (Continue)

