# OpenReview forum: "Draw-In-Mind: Rebalancing Designer-Painter Roles in Unified Multimodal Models Benefits Image Editing"
_ICLR.cc/2026/Conference — ICLR 2026 Poster_

### Official Review · Reviewer_B3js · 2025-10-30

**Soundness:** 3
**Presentation:** 3
**Contribution:** 3
**Rating:** 6
**Confidence:** 3

**Summary:**

This paper identifies a key limitation in current image editing models, which it frames as a "responsibility imbalance" between understanding and generation modules. The authors propose a new paradigm where the understanding module acts as a "designer," generating a detailed Chain-of-Thought (CoT) blueprint, allowing the generation module to focus solely on "painting." To support this, they introduce the Draw-In-Mind (DIM) dataset, which contains high-quality CoT "design blueprints." Their model, DIM-4.6B, connects a frozen MLLM to a trainable DiT and achieves competitive or state-of-the-art results on benchmarks like ImgEdit and GEdit-Bench, validating the proposed approach.

**Strengths:**

**Significant Dataset Contribution**: The DIM dataset is a valuable contribution. The authors have developed a systematic pipeline to generate high-quality "design blueprints," creating a useful resource for the community to explore more fine-grained control in image generation.

**Effective Architecture and Strong Results**: The model's simple design effectively isolates and demonstrates the power of the core paradigm and data. The fact that a 4.6B parameter model performs competitively against models over twice its size provides strong validation for the approach.

**Thorough Experimental Validation**: The experiments are well-structured. The main results are clearly presented, the generalizability study (Table 5) effectively demonstrates adaptability to different "designer" models, and the ablation studies (Tables 6 & 7) offer valuable insights into the sources of performance gain.

**Weaknesses:**

A key detail in the data generation process is that the "designer" model (GPT-4o) is provided with the target image for reference when creating the CoT blueprints for the training set. This creates a potential distribution mismatch between training and inference:

**During Training Data Generation**: The CoT blueprints are "oracle-guided," created with perfect knowledge of the desired outcome. They are essentially 100% accurate descriptions of the transformation.

**During Inference**: The "designer" model must generate the CoT blueprint without seeing the target image, relying solely on prediction. These blueprints are therefore susceptible to errors, omissions, or misinterpretations.

This discrepancy raises a question about how well the model, trained on "perfect" instructions, can handle the inevitably noisier and sometimes incorrect instructions it will receive in a real-world setting. While the strong results in Table 5 (showing generalizability across different designers) are encouraging and suggest a degree of robustness, a direct discussion of this gap would strengthen the paper.

**Questions:**

The current framework relies on an explicit, two-stage pipeline at inference time: first, calling a "Designer" (which can be an external model like GPT-4o or a separate open-source model) to generate the CoT blueprint, and second, feeding this blueprint into the "Painter" module. While effective, this explicitly modular design increases pipeline complexity.

It is worth discussing whether this "designing" capability could be internalized into the main understanding module itself, thereby streamlining the inference process. For instance, could the MLLM component of DIM-4.6B be trained to autoregressively generate the CoT blueprint before generating the control signals for the DiT, effectively merging the two stages into a unified, single-model call (perhaps via a multi-turn dialogue format or purely end-to-end training with CoT data)?

Did you explore standard unified training approaches where the model is taught to implicitly "reason" (generate the blueprint internally) before "acting" (generating the image)?

What are the key trade-offs that led to favoring the current explicit two-stage design over a more integrated, end-to-end approach? Is it primarily for the flexibility of swapping "designers," or did you find that integrated models inherently struggle to balance these two responsibilities even with high-quality CoT training data?

In summary, this is a good paper that presents an inspiring and effective solution to a key challenge in image editing. The methodology is sound and the experiments are thorough. The questions raised are not intended as criticisms but rather to encourage a deeper discussion on the important trade-offs inherent in this design.

---

> ### Author Response · Authors · 2025-11-20
> **Response to Reviewer B3js-Part 1/4**
>
> W1: The discrepancy between training and inference instruction distributions raises a question about how well the model, trained on "perfect" instructions, can handle the inevitably noisier and sometimes incorrect instructions it will receive in a real-world setting. While the strong results in Table 6 (showing generalizability across different designers) are encouraging and suggest a degree of robustness, a direct discussion of this gap would strengthen the paper.
>
> A: Thanks for raising this constructive point! Following your suggestion, we have conducted a comprehensive investigation of numerous cases in the revised manuscript. Our goal is to intuitively demonstrate how the Draw-In-Mind workflow enhances editing performance, particularly when instructions are ambiguous or highly challenging. The results are encouraging. Furthermore, in the interest of transparency, we also share specific failure cases to openly discuss the potential limitations of the proposed DIM-4.6B-Edit.
>
> Firstly, we explored cases where user instructions are ambiguous or require advanced reasoning in Figures 9-13. These examples demonstrate complex scenarios that are successfully handled by DIM-Edit-4.6B, highlighting capabilities that remain beyond the reach of current baseline methods:
>
> 1. **[Instruction Disambiguation]** In Figure 9, the user instruction presents an inherent ambiguity due to the presence of three lemons on the table. This task necessitates precise multi-object localization and removal, which is a challenge that proves difficult without the Draw-In-Mind paradigm, as standard models often struggle with the required multi-object reasoning. Consequently, both the 7B Janus-4o and 12B Step1X-Edit fail to execute the edit correctly. Similarly, when CoT is disabled, our DIM-4.6B-Edit also fails to remove all targets. However, with DIM CoT enabled, the generated design blueprints effectively disambiguate the instruction. They accurately localize the three lemons to the right of the vase and ensure their complete removal, while perfectly preserving the integrity of the unedited regions.
> 2. **[Edit Navigation & Structural Planning]** In Figure 10, the user instruction presents two distinct challenges: (i) determining the optimal placement for a wooden cabin, and (ii) identifying the appropriate structural integration for a chimney. These dual requirements impose a significant burden on the generation model. Consequently, in Janus-4o's output, the chimney is nearly invisible, while Step1X-Edit places the cabin counterintuitively close to the river. Similarly, DIM without CoT fails to simultaneously resolve the cabin placement and chimney addition. In contrast, DIM powered by CoT effectively navigates these challenges. It observes that "the trees thin out on the right side" (GLP) and selects this area as the optimal location (EAL). It then explicitly envisions the cabin's appearance, including a chimney emitting smoke (EII), ultimately yielding the most plausible and high-quality edit among all competitors.
> 3. **[Commonsense-guarded Editing]** In Figure 11, the editing task presents a subtle complexity: it requires commonsense reasoning regarding scale. From the same viewpoint, a cat should appear significantly smaller than a horse. All baseline models, including our own DIM w/o CoT, overlook this physical constraint, simply replacing the horse with a cat of identical dimensions. In contrast, DIM with CoT successfully leverages commonsense reasoning. It recognizes the size discrepancy and executes a "commonsense-guarded" edit, placing a naturally scaled cat at the target location, thereby preserving scene realism.
> 4. **[Advanced Causal Editing]** In Figure 12, we present an advanced causal editing scenario where the instruction implies the target quantity (referencing "the second prime number") rather than stating it explicitly. Unsurprisingly, all baseline models fail to resolve this implicit requirement. In contrast, DIM with CoT swiftly infers the correct number of cherries and executes a successful edit, demonstrating its ability to handle knowledge-intensive instructions.
> 5. **[Advanced Temporal Editing]** Figure 13 illustrates the most complex temporal editing scenario, which necessitates a deep understanding of chemical reaction dynamics. Similar to the previous example, none of the baseline models succeed in this task. In contrast, DIM with CoT accurately characterizes the reaction process and executes physically plausible edits, demonstrating its capability to handle sophisticated temporal reasoning.
>
> We believe these visualizations provide an intuitive demonstration of how the Draw-In-Mind paradigm significantly enhances image editing capabilities.

---

> ### Author Response · Authors · 2025-11-20
> **Response to Reviewer B3js-Part 2/4**
>
> (W1 Continue)
>
> Next, we provide three failure types with six specific cases in Figures 14-16 to intuitively show the boundaries of our DIM-4.6B-Edit:
>
> 1. **[Large-scale All-in-One Editing]** In Figure 14, the instructions involve simultaneous multi-step edits, a task that remains essentially challenging for almost all editing models, and one where DIM-Edit also encounters difficulties.
>     + For the first case, Janus-4o and Step1X-Edit completely fail to follow the physical laws dictating that the wooden tower should collapse. Our DIM-4.6B-Edit successfully imitates a scene of imminent collapse; however, it fails to preserve the exact appearance of the individual wooden blocks, as too many objects are involved in the manipulation.
>     + For the second case, Janus-4o and Step1X-Edit fail to change the view at all. While our DIM-4.6B-Edit completes the primary editing task, some fine-grained details are distorted (e.g., the window of the shoreside house is missing).
> 2. **[Text & Logic Editing]** In Figure 15, where instructions involve complex text rendering and logical editing, DIM-4.6B-Edit struggles due to a combination of data scarcity and inherent VAE compression issues.
>     + For the first case, the use of SANA1.5’s VAE with a 32x downsampling rate makes complex text rendering particularly challenging, a difficulty exacerbated by the lack of targeted training data. In contrast, Step1X-Edit employs an 8x downsampling VAE and is trained on proprietary, text-specific in-house data, allowing it to perform relatively well. We regard this as a necessary trade-off between efficiency and rendering quality: as shown in Table 13, DIM-4.6B-Edit requires only 6 seconds to complete an edit with a 200+ word CoT, whereas Step1X-Edit takes 28 seconds with a short raw prompt.
>     + For the second case, all editing models fail. This is fundamentally because none of the models, including DIM-4.6B-Edit, are specifically trained on geometric data. The underlying painter struggles to even draw these shapes, let alone edit them. We believe crafting such datasets remains a valuable and under-explored topic for future research.
> 3. **[Reference-free Editing (in Pixel Space)]** In Figure 16, the reference image does not provide a strong pixel constraint for the target image. Consequently, this task resembles multimodal generation rather than strict editing. All models fail here because existing editing architectures typically enforce strong pixel alignment with the source image.
>     + For the first case, which requests a view of the Golden Gate Bridge, Janus-4o and Step1X-Edit are completely ineffective. DIM-4.6B-Edit struggles to break free from the structural constraints of the reference image, resulting in a "scratchy" and distorted view that fails to meet the objective.
>     + For the second case, where the task involves a re-imagination of the source scene, Janus-4o produces a black-and-white edit, and Step1X-Edit fails completely. DIM-4.6B-Edit generates the most plausible result, successfully covering the scene with white snow. However, because the transformation fundamentally alters the source structure, specific details such as the castle are inevitably distorted.
>
> In summary, the majority of failure cases arise when the task necessitates either generating an image that diverges drastically from the source or rendering complex text and geometric shapes. Even in these challenging scenarios, DIM-4.6B-Edit demonstrates superior instruction-following capabilities compared to baseline models. These limitations highlight persistent challenges within the current landscape of open-source editing models. We suggest that future research directions, such as intelligent routing that dynamically selects between T2I generation and editing pipelines based on instruction intent, offer promising avenues for resolving these issues, though significant progress is still required in the field.
>
> **Revisions**
>
> 1. L1082-L1124 and Figures 9-13: Visualizations of how the Draw-In-Mind workflow impacts image editing.
> 2. L1298-L1348 and Figures 14-16: Visualizations and analysis of failure cases.

---

> ### Author Response · Authors · 2025-11-20
> **Response to Reviewer B3js-Part 3/4**
>
> Q1: It is worth discussing whether this "designing" capability could be internalized into the main understanding module itself, thereby streamlining the inference process. For instance, could the MLLM component of DIM-4.6B be trained to autoregressively generate the CoT blueprint before generating the control signals for the DiT, effectively merging the two stages into a unified, single-model call (perhaps via a multi-turn dialogue format or purely end-to-end training with CoT data)? Did you explore standard unified training approaches where the model is taught to implicitly "reason" (generate the blueprint internally) before "acting" (generating the image)?
>
> A: We appreciate this insightful question. We would like to clarify that by configuring the designer as Qwen2.5-VL-3B (the inherent MLLM of DIM-4.6B-Edit), we realize a **one-step designer-painter system** that eliminates the need for an additional round of external designer calls. In this setup, the design blueprints are directly generated via the internal MLLM, and the embeddings are fluently fed into the painter to complete the editing. Notably, this one-step model, without any external parameters, already reaches SOTA editing performance across open-source models. This indicates the effectiveness of the Draw-In-Mind paradigm as well as the high data quality of DIM-Edit. During the rebuttal, we follow your advice to conducted an additional experiment to demonstrate that fine-tuning this internal MLLM on the DIM-Edit CoT data in an autoregressive manner leads to improved performance. The results are attached below:
>
> | Designer | Add | Adjust | Extract | Replace | Remove | Background | Style | Hybrid | Action | Overall |
> | :--- | :---: | :---: | :---: | :---: | :---: | :---: | :---: | :---: | :---: | :---: |
> | - | 3.53 | 3.23 | 2.01 | 3.49 | 1.47 | 3.42 | 4.79 | 2.35 | 3.64 | 3.10 |
> | Qwen2.5-VL-3B❄️ | 3.80 | 3.24 | 2.03 | 3.89 | 3.21 | 3.52 | 4.92 | 2.71 | 4.05 | 3.49 |
> | Qwen2.5-VL-3B🔥 | 3.96 | 3.36 | 2.25 | 3.98 | 3.31 | 3.81 | 4.95 | 2.83 | 4.02 | 3.61 |
> | GPT-4o | 4.09 | 3.47 | 2.30 | 4.00 | 3.43 | 3.87 | 4.92 | 2.85 | 4.08 | 3.67 |
>
> **The second row corresponds to a "self-play" configuration, where the CoT embeddings generated by the internal MLLM are directly fed into the painter to execute edits, effectively bypassing the need for an additional external designer inference round.** To maintain simplicity and ensure consistency with our primary narrative regarding external designer integration, we chose not to emphasize this specific mode in the main text in our manuscript.
>
> Thanks again for this insightful suggestion. Based on the discussion, we conclude that our work offers dual advantages in terms of both data and modeling:
>
> 1. **Parameter-Constrained Setting:** Our DIM data provides a highly efficient mechanism to bridge the performance gap between open-source and closed-source designers through lightweight CoT fine-tuning.
> 2. **Data-Constrained Setting:** Our DIM model enables immediate performance boosts by seamlessly integrating advanced external designers without additional training. This is possible because our DIM-Edit CoT maintains both high quality and strong generalizability across different models.
>
> We have incorporated this discussion into the revised manuscript. We believe these findings strengthen the soundness and broader impact of our work.
>
> **Revisions**
>
> 1. L454-L462 and Table 7: Integrated end-to-end evaluation.

---

> ### Author Response · Authors · 2025-11-20
> **Response to Reviewer B3js-Part 4/4**
>
> Q2: What are the key trade-offs that led to favoring the current explicit two-stage design over a more integrated, end-to-end approach? Is it primarily for the flexibility of swapping "designers," or did you find that integrated models inherently struggle to balance these two responsibilities even with high-quality CoT training data?
>
> A: Thanks for raising this constructive discussion! **Yes, our design choice primarily prioritizes the flexibility of swapping "designers".** As discussed in Q2, the integrated end-to-end approach already yields SOTA editing performance. Since utilizing blueprints from a DIM-Edit-finetuned MLLM further enhances results, we infer that fine-tuning the entire system in an end-to-end manner could theoretically maximize performance and would not struggle with responsibility balance if trained on our DIM dataset. Anyway, such an undertaking falls outside the primary scope of this paper. Our objective is not to train a massive system solely to chase benchmark high scores. Instead, we provide a minimal, efficient training recipe designed to rigorously verify the effectiveness of the DIM-Edit data and the Draw-In-Mind paradigm.
>
> Furthermore, we believe this flexibility significantly lowers the barrier to entry for other researchers. Resource-rich groups can leverage our DIM dataset to construct entirely new systems. Conversely, our DIM-4.6B-Edit offers a modular platform for those who wish to build upon existing advanced MLLMs (e.g., InternVL3.5). Its compact size ensures that training remains affordable for the broader research community. For inference-only users, the flexibility to integrate customized designers offers a convenient path to improved performance. Rather than following the industrial trend of simply scaling up model and data sizes—an avenue often restricted to large tech companies—our work proposes a novel solution focused on *data form*. We hope this perspective inspires wider engagement in the field.
>
> **If you have any further concerns, please feel free to leave us a message. We are willing to resolve them!**

---

### Official Review · Reviewer_PX1Q · 2025-11-01

**Soundness:** 3
**Presentation:** 4
**Contribution:** 3
**Rating:** 8
**Confidence:** 3

**Summary:**

This paper identifies a key issue in current image editing models: an imbalanced division of labor where the generator bears both design and painting responsibilities. To address this, the authors propose the Draw-In-Mind (DIM) dataset, which includes detailed, CoT blueprints for image edits. They also introduce a simple model, DIM-4.6B-Edit, which leverages an external designer to generate these blueprints. This approach shifts the design burden from the generator to the understanding module. Extensive experiments show that their method achieves state-of-the-art performance on multiple image editing benchmarks, outperforming much larger models.

**Strengths:**

*   **Well-Motivated Idea:** The paper is built on a clear and insightful observation about the fundamental limitations of current image editing models. The concept of rebalancing the "design" and "painting" responsibilities between modules is novel, well-justified, and intuitively appealing.
*   **High-Quality Dataset:** The introduction of the DIM dataset, particularly DIM-Edit with its detailed CoT annotations, is a significant contribution. The ablation studies effectively demonstrate the value of this data, and the dataset is likely to be a valuable asset to the research community.
*   **Comprehensive Evaluation:** The paper provides thorough experimental comparisons, convincingly demonstrating the advantages of the proposed DIM-4.6B-Edit model over other strong open-source baselines across multiple benchmarks and editing tasks.

**Weaknesses:**

*   **Potentially Unfair Parameter Comparison:** The parameter count comparisons in Figure 1 and the main tables do not include the parameters of the External Designer (e.g., GPT-4o), which is a critical component of the proposed pipeline during inference. A significant portion of the performance gain likely stems from the powerful text refinement and CoT generation capability of this external model. Therefore, comparing the 4.6B parameter core model against the end-to-end parameters of other models may present an incomplete picture of the total "cost" of the method.

**Questions:**

- The paper shows many successful editing examples. Could the authors provide some analysis or examples of cases where their method fails? Understanding the limitations and failure modes (e.g., types of instructions or image complexities that are challenging) would help the readers to better understand the boundaries of this approach.

---

> ### Author Response · Authors · 2025-11-20
> **Response to Reviewer PX1Q-Part 1/3**
>
> W1: **Potentially Unfair Parameter Comparison:** The parameter count comparisons in Figure 1 and the main tables do not include the parameters of the External Designer (e.g., GPT-4o), which is a critical component of the proposed pipeline during inference. A significant portion of the performance gain likely stems from the powerful text refinement and CoT generation capability of this external model. Therefore, comparing the 4.6B parameter core model against the end-to-end parameters of other models may present an incomplete picture of the total "cost" of the method.
>
> A: We appreciate this insightful question. We would like to clarify that by configuring the designer as Qwen2.5-VL-3B (the inherent MLLM of DIM-4.6B-Edit), we realize a **one-step designer-painter system** that eliminates the need for an additional round of external designer calls. In this setup, the design blueprints are directly generated via the internal MLLM, and the embeddings are fluently fed into the painter to complete the editing. Notably, this one-step model, without any external parameters, already reaches SOTA editing performance across open-source models. This indicates the effectiveness of the Draw-In-Mind paradigm as well as the high data quality of DIM-Edit. During the rebuttal, we conducted an additional experiment to demonstrate that fine-tuning this internal MLLM on the DIM-Edit CoT data leads to improved performance. The results are attached below:
>
> | Designer | Add | Adjust | Extract | Replace | Remove | Background | Style | Hybrid | Action | Overall |
> | :--- | :---: | :---: | :---: | :---: | :---: | :---: | :---: | :---: | :---: | :---: |
> | - | 3.53 | 3.23 | 2.01 | 3.49 | 1.47 | 3.42 | 4.79 | 2.35 | 3.64 | 3.10 |
> | Qwen2.5-VL-3B❄️ | 3.80 | 3.24 | 2.03 | 3.89 | 3.21 | 3.52 | 4.92 | 2.71 | 4.05 | 3.49 |
> | Qwen2.5-VL-3B🔥 | 3.96 | 3.36 | 2.25 | 3.98 | 3.31 | 3.81 | 4.95 | 2.83 | 4.02 | 3.61 |
> | GPT-4o | 4.09 | 3.47 | 2.30 | 4.00 | 3.43 | 3.87 | 4.92 | 2.85 | 4.08 | 3.67 |
>
> **The second row corresponds to a "self-play" configuration, where the CoT embeddings generated by the internal MLLM are directly fed into the painter to execute edits, effectively bypassing the need for an additional external designer inference round.** To maintain simplicity and ensure consistency with our primary narrative regarding external designer integration, we chose not to emphasize this specific mode in the main text in our manuscript.
>
> Thanks again for this insightful suggestion. Based on the discussion, we conclude that our work offers dual advantages in terms of both data and modeling:
>
> 1. **Parameter-Constrained Setting:** Our DIM data provides a highly efficient mechanism to bridge the performance gap between open-source and closed-source designers through lightweight CoT fine-tuning.
> 2. **Data-Constrained Setting:** Our DIM model enables immediate performance boosts by seamlessly integrating advanced external designers without additional training. This is possible because our DIM-Edit CoT maintains both high quality and strong generalizability across different models.
>
> We have incorporated this discussion into the revised manuscript. We believe these findings strengthen the soundness and broader impact of our work.
>
> **Revisions**
>
> 1. L454-L462 and Table 7: Integrated end-to-end evaluation.

---

> ### Author Response · Authors · 2025-11-20
> **Response to Reviewer PX1Q-Part 2/3**
>
> Q1: The paper shows many successful editing examples. Could the authors provide some analysis or examples of cases where their method fails? Understanding the limitations and failure modes (e.g., types of instructions or image complexities that are challenging) would help the readers to better understand the boundaries of this approach.
>
> A: Excellent question! We have included two additional visualization parts to intuitively show the boundaries of this approach as follows.
>
> **[Failure Case]** We provide three failure types with six specific cases in Figures 14-16 to show the boundaries of our DIM-4.6B-Edit:
>
> 1. **[Large-scale All-in-One Editing]** In Figure 14, the instructions involve simultaneous multi-step edits, a task that remains essentially challenging for almost all editing models, and one where DIM-Edit also encounters difficulties.
>     + For the first case, Janus-4o and Step1X-Edit completely fail to follow the physical laws dictating that the wooden tower should collapse. Our DIM-4.6B-Edit successfully imitates a scene of imminent collapse; however, it fails to preserve the exact appearance of the individual wooden blocks, as too many objects are involved in the manipulation.
>     + For the second case, Janus-4o and Step1X-Edit fail to change the view at all. While our DIM-4.6B-Edit completes the primary editing task, some fine-grained details are distorted (e.g., the window of the shoreside house is missing).
> 2. **[Text & Logic Editing]** In Figure 15, where instructions involve complex text rendering and logical editing, DIM-4.6B-Edit struggles due to a combination of data scarcity and inherent VAE compression issues.
>     + For the first case, the use of SANA1.5’s VAE with a 32x downsampling rate makes complex text rendering particularly challenging, a difficulty exacerbated by the lack of targeted training data. In contrast, Step1X-Edit employs an 8x downsampling VAE and is trained on proprietary, text-specific in-house data, allowing it to perform relatively well. We regard this as a necessary trade-off between efficiency and rendering quality: as shown in Table 13, DIM-4.6B-Edit requires only 6 seconds to complete an edit with a 200+ word CoT, whereas Step1X-Edit takes 28 seconds with a short raw prompt.
>     + For the second case, all editing models fail. This is fundamentally because none of the models, including DIM-4.6B-Edit, are specifically trained on geometric data. The underlying painter struggles to even draw these shapes, let alone edit them. We believe crafting such datasets remains a valuable and under-explored topic for future research.
> 3. **[Reference-free Editing (in Pixel Space)]** In Figure 16, the reference image does not provide a strong pixel constraint for the target image. Consequently, this task resembles multimodal generation rather than strict editing. All models fail here because existing editing architectures typically enforce strong pixel alignment with the source image.
>     + For the first case, which requests a view of the Golden Gate Bridge, Janus-4o and Step1X-Edit are completely ineffective. DIM-4.6B-Edit struggles to break free from the structural constraints of the reference image, resulting in a "scratchy" and distorted view that fails to meet the objective.
>     + For the second case, where the task involves a re-imagination of the source scene, Janus-4o produces a black-and-white edit, and Step1X-Edit fails completely. DIM-4.6B-Edit generates the most plausible result, successfully covering the scene with white snow. However, because the transformation fundamentally alters the source structure, specific details such as the castle are inevitably distorted.
>
> In summary, the majority of failure cases arise when the task necessitates either generating an image that diverges drastically from the source or rendering complex text and geometric shapes. Even in these challenging scenarios, DIM-4.6B-Edit demonstrates superior instruction-following capabilities compared to baseline models. These limitations highlight persistent challenges within the current landscape of open-source editing models. We suggest that future research directions, such as intelligent routing that dynamically selects between T2I generation and editing pipelines based on instruction intent, offer promising avenues for resolving these issues, though significant progress is still required in the field.

---

> ### Author Response · Authors · 2025-11-20
> **Response to Reviewer PX1Q-Part 3/3**
>
> (Q1 Continue)
>
> **[Advanced Editing]** We provide Figures 9-13 in the revised manuscript, which showcases several advanced usage scenarios. These examples demonstrate complex cases that are successfully handled by DIM-Edit-4.6B, highlighting capabilities that remain beyond the reach of current baseline methods:
>
> 1. **[Instruction Disambiguation]** In Figure 9, the user instruction presents an inherent ambiguity due to the presence of three lemons on the table. This task necessitates precise multi-object localization and removal, which is a challenge that proves difficult without the Draw-In-Mind paradigm, as standard models often struggle with the required multi-object reasoning. Consequently, both the 7B Janus-4o and 12B Step1X-Edit fail to execute the edit correctly. Similarly, when CoT is disabled, our DIM-4.6B-Edit also fails to remove all targets. However, with DIM CoT enabled, the generated design blueprints effectively disambiguate the instruction. They accurately localize the three lemons to the right of the vase and ensure their complete removal, while perfectly preserving the integrity of the unedited regions.
> 2. **[Edit Navigation & Structural Planning]** In Figure 10, the user instruction presents two distinct challenges: (i) determining the optimal placement for a wooden cabin, and (ii) identifying the appropriate structural integration for a chimney. These dual requirements impose a significant burden on the generation model. Consequently, in Janus-4o's output, the chimney is nearly invisible, while Step1X-Edit places the cabin counterintuitively close to the river. Similarly, DIM without CoT fails to simultaneously resolve the cabin placement and chimney addition. In contrast, DIM powered by CoT effectively navigates these challenges. It observes that "the trees thin out on the right side" (GLP) and selects this area as the optimal location (EAL). It then explicitly envisions the cabin's appearance, including a chimney emitting smoke (EII), ultimately yielding the most plausible and high-quality edit among all competitors.
> 3. **[Commonsense-guarded Editing]** In Figure 11, the editing task presents a subtle complexity: it requires commonsense reasoning regarding scale. From the same viewpoint, a cat should appear significantly smaller than a horse. All baseline models, including our own DIM w/o CoT, overlook this physical constraint, simply replacing the horse with a cat of identical dimensions. In contrast, DIM with CoT successfully leverages commonsense reasoning. It recognizes the size discrepancy and executes a "commonsense-guarded" edit, placing a naturally scaled cat at the target location, thereby preserving scene realism.
> 4. **[Advanced Causal Editing]** In Figure 12, we present an advanced causal editing scenario where the instruction implies the target quantity (referencing "the second prime number") rather than stating it explicitly. Unsurprisingly, all baseline models fail to resolve this implicit requirement. In contrast, DIM with CoT swiftly infers the correct number of cherries and executes a successful edit, demonstrating its ability to handle knowledge-intensive instructions.
> 5. **[Advanced Temporal Editing]** Figure 13 illustrates the most complex temporal editing scenario, which necessitates a deep understanding of chemical reaction dynamics. Similar to the previous example, none of the baseline models succeed in this task. In contrast, DIM with CoT accurately characterizes the reaction process and executes physically plausible edits, demonstrating its capability to handle sophisticated temporal reasoning.
>
> We believe these visualizations provide an intuitive demonstration of how the Draw-In-Mind paradigm significantly enhances image editing capabilities.
>
> **Revisions**
>
> 1. L1298-L1348 and Figures 14-16: Visualizations and analysis of failure cases.
> 2. L1082-L1124 and Figures 9-13: Visualizations of how the Draw-In-Mind workflow impacts image editing.
>
> **If you have any further concerns, please feel free to leave us a message. We are willing to resolve them!**

---

### Official Review · Reviewer_RiTv · 2025-11-01

**Soundness:** 3
**Presentation:** 3
**Contribution:** 2
**Rating:** 4
**Confidence:** 3

**Summary:**

The paper avoids complex architecture, using only a “two-layer MLP + small DiT”, thereby making it clear that the performance gains stem from data and the division of responsibilities, which keeps the causal narrative coherent. However, systematic evaluation on understanding tasks and on broader generative tasks remains insufficient, and the reliance on LLM-as-Judge and on GPT-4o as the data/designer limits the external validity of the conclusions.

**Strengths:**

1. Emphasizes the importance and contribution of data.
2. Demonstrates the effectiveness of "think before drawing" (plan-then-generate).

**Weaknesses:**

1. Evaluation relies on LLM-as-Judge; adding quantifiable, automated metrics is recommended.
2. Claims to be a unified understanding–generation model, but lacks evaluation on understanding tasks.
3. Insufficient exploration of how the new workflow impacts image generation.

**Questions:**

1. Is there no way for a unified model to unify both thinking/reasoning and generation? If not, what is the significance of unifying understanding and generation? (Just for discussion)
2. If this paper splits generation and understanding into two components, how does it differ from prior encoder-side work in models that use an MLLM to unify generation tasks? Is the main difference simply the addition of CoT?
3. Why claim a unified understanding–generation model without evaluating understanding tasks or other generation tasks?

---

> ### Author Response · Authors · 2025-11-20
> **Response to Reviewer RiTv-Part 1/8**
>
> Summary: "The reliance on GPT-4o as the designer limits the external validity of the conclusions."
>
> A: Thanks for raising this concern! We agree with you that relying solely on closed-source models like GPT-4o can limit external validity and hinder broad applicability. **This concern was our primary motivation for conducting the generalizability experiments presented in Table 6 (see below).**
>
> | Designer | Add | Adjust | Extract | Replace | Remove | Background | Style | Hybrid | Action | Overall |
> | :--- | :---: | :---: | :---: | :---: | :---: | :---: | :---: | :---: | :---: | :---: |
> | - | 3.53 | 3.23 | 2.01 | 3.49 | 1.47 | 3.42 | 4.79 | 2.35 | 3.64 | 3.10 |
> | Qwen2.5-VL-7B | 3.95 | 3.35 | 2.25 | 3.85 | 3.31 | 3.57 | 4.88 | 2.81 | 4.02 | 3.55 |
> | MiMo-VL-7B | 3.95 | 3.32 | 2.20 | 3.75 | 2.46 | 3.82 | 4.88 | 2.52 | 3.93 | 3.43 |
> | InternVL3.5-8B | 3.98 | 3.40 | 2.05 | 4.14 | 3.30 | 3.84 | 4.94 | 2.77 | 3.89 | 3.59 |
> | GLM-4.1V-9B | 3.95 | 3.27 | 2.23 | 3.90 | 2.64 | 3.81 | 4.92 | 2.23 | 4.02 | 3.44 |
> | GPT-4o | 4.09 | 3.47 | 2.30 | 4.00 | 3.43 | 3.87 | 4.92 | 2.85 | 4.08 | 3.67 |
>
> In this experiment, we utilized the same prompt to generate DIM-style CoTs using various **open-source** MLLMs and fed them into DIM-4.6B-Edit as design blueprints during inference. **The results are encouraging: all tested open-source designers yielded performance comparable to the closed-source GPT-4o. This indicates that our Draw-In-Mind paradigm is not only effective but also highly generalizable across different MLLMs.** These findings strongly strengthen the external validity of our work for the academic community, demonstrating that researchers can freely substitute the external designer with their own models and still achieve successful editing outcomes.
>
> We appreciate your insightful comment, which prompted us to rigorously verify external validity from the **training perspective**, complementing our initial inference-side analysis. While our primary experiments demonstrated generalizability by freezing the MLLM and tuning the DiT, we have now conducted a **complementary "reverse" experiment** to assess the value of our dataset for MLLM training. Specifically, we performed SFT on Qwen2.5-VL-3B using the CoT corpus from DIM-Edit. We then utilized the CoTs generated by this fine-tuned model to guide our DIM-4.6B-Edit. This experiment empirically validates whether DIM-Edit can effectively bridge the performance disparity between advanced closed-source designers (e.g., GPT-4o) and open-source alternatives. The results are presented below:
>
> | Designer | Add | Adjust | Extract | Replace | Remove | Background | Style | Hybrid | Action | Overall |
> | :--- | :---: | :---: | :---: | :---: | :---: | :---: | :---: | :---: | :---: | :---: |
> | - | 3.53 | 3.23 | 2.01 | 3.49 | 1.47 | 3.42 | 4.79 | 2.35 | 3.64 | 3.10 |
> | Qwen2.5-VL-3B❄️ | 3.80 | 3.24 | 2.03 | 3.89 | 3.21 | 3.52 | 4.92 | 2.71 | 4.05 | 3.49 |
> | Qwen2.5-VL-3B🔥 | 3.96 | 3.36 | 2.25 | 3.98 | 3.31 | 3.81 | 4.95 | 2.83 | 4.02 | 3.61 |
> | GPT-4o | 4.09 | 3.47 | 2.30 | 4.00 | 3.43 | 3.87 | 4.92 | 2.85 | 4.08 | 3.67 |
>
> As shown in the table above, utilizing CoTs from the fine-tuned Qwen2.5-VL-3B results in clear performance gains during evaluation. This demonstrates that our DIM-Edit dataset possesses dual utility: it is not only effective for training editing behaviors directly but also serves as a robust corpus for enhancing the planning of accurate design blueprints on the understanding side. Consequently, DIM-Edit stands as a valuable data asset for the research community.
>
> **Revisions**
>
> 1. L454-L462 and Table 6/7: The ImgEdit performance of different external designers and the performance of utilizing CoTs from the Qwen2.5-VL-3B fine-tuned on DIM-Edit's CoT.

---

> ### Author Response · Authors · 2025-11-20
> **Response to Reviewer RiTv-Part 2/8**
>
> W1: Evaluation relies on LLM-as-Judge; adding quantifiable, automated metrics is recommended.
>
> A: We appreciate the suggestion. Following your advice, we conducted a comprehensive review of benchmarks from recent years and identified **MagicBrush** as a unique benchmark where the test set is generated by humans. This characteristic allows for precise pixel-level evaluation between AI-generated edits and human-created ground truth. The evaluation metrics include L1 distance, CLIP-Image similarity, and DINO similarity. Consequently, we compared our DIM-4.6B-Edit against the current state-of-the-art model on this benchmark, i.e., **ICEdit**. The results are presented below:
>
> | Method | Gen Params | L1 $\downarrow$ | CLIP-I $\uparrow$ | DINO $\uparrow$ |
> | :--- | :---: | :---: | :---: | :---: |
> | InstructP2P [CVPR'23] | 0.9B | 0.114 | 0.851 | 0.744 |
> | MagicBrush [NeurIPS'23] | 0.9B | 0.074 | 0.908 | 0.847 |
> | UltraEdit [NeurIPS'24] | 2.0B | 0.066 | 0.904 | 0.852 |
> | FluxEdit [HuggingFace] | 12.0B | 0.114 | 0.779 | 0.663 |
> | FLUX.1 Fill [HuggingFace] | 12.0B | 0.192 | 0.795 | 0.669 |
> | RF-Solver Edit [ICML'25] | 12.0B | 0.112 | 0.766 | 0.675 |
> | ACE++ [arXiv'25] | 12.0B | 0.195 | 0.741 | 0.591 |
> | ICEdit [NeurIPS'25] | 12.0B | 0.060 **(1st)** | 0.928 **(1st)** | 0.853 **(2nd)** |
> | DIM-4.6B-Edit | 1.6B | 0.065 **(2nd)** | 0.928 **(1st)** | 0.882 **(1st)** |
>
> It can be seen that DIM-4.6B-Edit achieves state-of-the-art performance comparable to ICEdit. **Notably, ICEdit employs a 12B FLUX.1 Fill backbone, with MagicBrush samples constituting approximately 20% of its total training set. In contrast, DIM-4.6B-Edit utilizes a compact 1.6B generation backbone, where MagicBrush data accounts for less than 3% of our DIM-Edit dataset.** These comparable results validate the effectiveness of the Draw-In-Mind paradigm and the generalizability of our DIM-Edit CoT. Despite our training distribution being significantly less driven by MagicBrush data, our model matches the performance of a competitor that is 5x larger. We have included this comparison in the revised manuscript.
>
> **Revisions**
>
> 1. L417-L424 and Table 5: The performance of automated metrics on MagicBrush test set.
>
> |----------------------------------------|
>
> W2: Claims to be a unified understanding–generation model, but lacks evaluation on understanding tasks.
>
> A: Thanks for pointing out this negligence! Given that this submission targets the Dataset Track, our primary focus is to rigorously evaluate the effectiveness of the Draw-In-Mind paradigm and the proposed DIM dataset. Consequently, we intentionally maintained a streamlined architecture to isolate the data's contribution. As detailed in Section 3.2, we freeze the MLLM to preserve its state-of-the-art understanding capabilities and fine-tune only the connector and the small DiT. The fact that this constrained setting achieves SOTA editing performance provides strong evidence that the performance gains stem directly from the quality of the DIM dataset, rather than complex architectural engineering. We believe these findings offer valuable insights into the potential of the "plan-then-generate" paradigm for future editing research.
>
> **Since the MLLM component is frozen during training, its understanding performance remains unaffected and is identical to the results reported in the original paper.** Consequently, we initially omitted these metrics from the manuscript. We apologize for any confusion this may have caused. To ensure clarity regarding the model's capabilities, we report the corresponding understanding performance in the table below:
>
> | Model | Params | MME-P | MMB | SEED | MMMU | MM-Vet |
> | :--- | :---: | :---: | :---: | :---: | :---: | :---: |
> | Janus | 1.3B🔥 | 1338.0 | 69.4 | 63.7 | 30.5 | 34.3 |
> | Emu3-Gen | 8.0B🔥 | - | 58.5 | 68.2 | 31.6 | 37.2 |
> | Show-o | 1.3B🔥 | 1097.2 | - | - | 26.7 | - |
> | Show-o2-7B | 7.0B🔥 | 1620.5 | 79.3 | 69.8 | 48.9 | - |
> | Janus-Pro-7B | 7.0B🔥 | 1567.1 | 79.2 | 72.1 | 41.0 | 50.0 |
> | BAGEL | 14.0B🔥 | 1687.0 | 85.0 | - | 55.3 | 67.2 |
> | MetaQuery-L | 3.0B❄️ \| 3.2B🔥 | 1574.3 | 78.6 | 73.8 | 53.1 | 63.2 |
> | DIM-4.6B-T2I/Edit | 3.0B❄️ \| 1.6B🔥 | 1574.3 | 78.6 | 73.8 | 53.1 | 63.2 |
>
> Our experiments demonstrate that DIM-4.6B-Edit achieves superior editing performance even when utilizing a relatively small MLLM under a frozen setting. This finding highlights the flexibility of our approach: **users can seamlessly upgrade to more advanced MLLMs to unlock even greater understanding and editing performance. Such integration is straightforward, as our streamlined architecture and training recipe avoid the need for intricate parameter tuning.**
>
> **Revisions**
>
> 1. L914-L921 and Table 15: Understanding benchmarks' performances.

---

> ### Author Response · Authors · 2025-11-20
> **Response to Reviewer RiTv-Part 3/8**
>
> W3: Insufficient exploration of how the new workflow impacts image generation.
>
> A: We appreciate the valuable suggestion. To assess the individual impact of each CoT component, we conducted an ablation study by removing components one at a time. The results are reported in Table 9 in our manuscript (attached below):
>
> | CoT Composition | Add | Adjust | Extract | Replace | Remove | Background | Style | Hybrid | Action | Overall |
> | :--- | :---: | :---: | :---: | :---: | :---: | :---: | :---: | :---: | :---: | :---: |
> | DIM-Edit | 4.09 | 3.47 | 2.30 | 4.00 | 3.43 | 3.87 | 4.92 | 2.85 | 4.08 | 3.67 |
> | &nbsp;&nbsp;&nbsp; w/o GLP | 3.85 | 3.29 | 2.06 | 3.91 | 3.24 | 3.55 | 4.80 | 2.79 | 3.92 | 3.49 |
> | &nbsp;&nbsp;&nbsp; w/o LOP | 3.80 | 3.15 | 1.92 | 3.83 | 3.07 | 3.60 | 4.79 | 2.44 | 3.92 | 3.39 |
> | &nbsp;&nbsp;&nbsp; w/o EAL | 3.79 | 3.25 | 1.96 | 3.73 | 2.96 | 3.65 | 4.81 | 2.82 | 3.82 | 3.42 |
> | &nbsp;&nbsp;&nbsp; w/o EII | 3.77 | 3.22 | 1.82 | 3.88 | 2.96 | 3.61 | 4.78 | 2.55 | 3.58 | 3.35 |
>
> All components contribute positively to the performance, though their importance varies. The GLP has minimal impact, likely because it is an easy task for the generator. While the other three CoT components, i.e., LOP, EAL, and EII, have a significant effect. LOP and EAL require the model to focus on specific regions, while EII demands complex reasoning; none of these are trivial for the generator.

---

> ### Author Response · Authors · 2025-11-20
> **Response to Reviewer RiTv-Part 4/8**
>
> (W3 Continue)
>
> We have recognized that relying solely on numerical metrics may not intuitively convey the practical impact of the new workflow on image generation. To address this, we have included Figures 9-13 in the revised manuscript, which showcases several advanced usage scenarios. These examples demonstrate complex cases that are successfully handled by DIM-Edit-4.6B, highlighting capabilities that remain beyond the reach of current baseline methods:
>
> 1. **[Instruction Disambiguation]** In Figure 9, the user instruction presents an inherent ambiguity due to the presence of three lemons on the table. This task necessitates precise multi-object localization and removal, which is a challenge that proves difficult without the Draw-In-Mind paradigm, as standard models often struggle with the required multi-object reasoning. Consequently, both the 7B Janus-4o and 12B Step1X-Edit fail to execute the edit correctly. Similarly, when CoT is disabled, our DIM-4.6B-Edit also fails to remove all targets. However, with DIM CoT enabled, the generated design blueprints effectively disambiguate the instruction. They accurately localize the three lemons to the right of the vase and ensure their complete removal, while perfectly preserving the integrity of the unedited regions.
> 2. **[Edit Navigation & Structural Planning]** In Figure 10, the user instruction presents two distinct challenges: (i) determining the optimal placement for a wooden cabin, and (ii) identifying the appropriate structural integration for a chimney. These dual requirements impose a significant burden on the generation model. Consequently, in Janus-4o's output, the chimney is nearly invisible, while Step1X-Edit places the cabin counterintuitively close to the river. Similarly, DIM without CoT fails to simultaneously resolve the cabin placement and chimney addition. In contrast, DIM powered by CoT effectively navigates these challenges. It observes that "the trees thin out on the right side" (GLP) and selects this area as the optimal location (EAL). It then explicitly envisions the cabin's appearance, including a chimney emitting smoke (EII), ultimately yielding the most plausible and high-quality edit among all competitors.
> 3. **[Commonsense-guarded Editing]** In Figure 11, the editing task presents a subtle complexity: it requires commonsense reasoning regarding scale. From the same viewpoint, a cat should appear significantly smaller than a horse. All baseline models, including our own DIM w/o CoT, overlook this physical constraint, simply replacing the horse with a cat of identical dimensions. In contrast, DIM with CoT successfully leverages commonsense reasoning. It recognizes the size discrepancy and executes a "commonsense-guarded" edit, placing a naturally scaled cat at the target location, thereby preserving scene realism.
> 4. **[Advanced Causal Editing]** In Figure 12, we present an advanced causal editing scenario where the instruction implies the target quantity (referencing "the second prime number") rather than stating it explicitly. Unsurprisingly, all baseline models fail to resolve this implicit requirement. In contrast, DIM with CoT swiftly infers the correct number of cherries and executes a successful edit, demonstrating its ability to handle knowledge-intensive instructions.
> 5. **[Advanced Temporal Editing]** Figure 13 illustrates the most complex temporal editing scenario, which necessitates a deep understanding of chemical reaction dynamics. Similar to the previous example, none of the baseline models succeed in this task. In contrast, DIM with CoT accurately characterizes the reaction process and executes physically plausible edits, demonstrating its capability to handle sophisticated temporal reasoning.
>
> We believe these visualizations provide an intuitive demonstration of how the Draw-In-Mind paradigm significantly enhances image editing capabilities.

---

> ### Author Response · Authors · 2025-11-20
> **Response to Reviewer RiTv-Part 5/8**
>
> (W3 Continue)
>
> Besides, we are open to discuss the limitations of our work, and provide three failure types with six specific cases in Figures 14-16 to intuitively show the boundaries of our DIM-4.6B-Edit:
>
> 1. **[Large-scale All-in-One Editing]** In Figure 14, the instructions involve simultaneous multi-step edits, a task that remains essentially challenging for almost all editing models, and one where DIM-Edit also encounters difficulties.
>     + For the first case, Janus-4o and Step1X-Edit completely fail to follow the physical laws dictating that the wooden tower should collapse. Our DIM-4.6B-Edit successfully imitates a scene of imminent collapse; however, it fails to preserve the exact appearance of the individual wooden blocks, as too many objects are involved in the manipulation.
>     + For the second case, Janus-4o and Step1X-Edit fail to change the view at all. While our DIM-4.6B-Edit completes the primary editing task, some fine-grained details are distorted (e.g., the window of the shoreside house is missing).
> 2. **[Text & Logic Editing]** In Figure 15, where instructions involve complex text rendering and logical editing, DIM-4.6B-Edit struggles due to a combination of data scarcity and inherent VAE compression issues.
>     + For the first case, the use of SANA1.5’s VAE with a 32x downsampling rate makes complex text rendering particularly challenging, a difficulty exacerbated by the lack of targeted training data. In contrast, Step1X-Edit employs an 8x downsampling VAE and is trained on proprietary, text-specific in-house data, allowing it to perform relatively well. We regard this as a necessary trade-off between efficiency and rendering quality: as shown in Table 13, DIM-4.6B-Edit requires only 6 seconds to complete an edit with a 200+ word CoT, whereas Step1X-Edit takes 28 seconds with a short raw prompt.
>     + For the second case, all editing models fail. This is fundamentally because none of the models, including DIM-4.6B-Edit, are specifically trained on geometric data. The underlying painter struggles to even draw these shapes, let alone edit them. We believe crafting such datasets remains a valuable and under-explored topic for future research.
> 3. **[Reference-free Editing (in Pixel Space)]** In Figure 16, the reference image does not provide a strong pixel constraint for the target image. Consequently, this task resembles multimodal generation rather than strict editing. All models fail here because existing editing architectures typically enforce strong pixel alignment with the source image.
>     + For the first case, which requests a view of the Golden Gate Bridge, Janus-4o and Step1X-Edit are completely ineffective. DIM-4.6B-Edit struggles to break free from the structural constraints of the reference image, resulting in a "scratchy" and distorted view that fails to meet the objective.
>     + For the second case, where the task involves a re-imagination of the source scene, Janus-4o produces a black-and-white edit, and Step1X-Edit fails completely. DIM-4.6B-Edit generates the most plausible result, successfully covering the scene with white snow. However, because the transformation fundamentally alters the source structure, specific details such as the castle are inevitably distorted.
>
> In summary, the majority of failure cases arise when the task necessitates either generating an image that diverges drastically from the source or rendering complex text and geometric shapes. Even in these challenging scenarios, DIM-4.6B-Edit demonstrates superior instruction-following capabilities compared to baseline models. These limitations highlight persistent challenges within the current landscape of open-source editing models. We suggest that future research directions, such as intelligent routing that dynamically selects between T2I generation and editing pipelines based on instruction intent, offer promising avenues for resolving these issues, though significant progress is still required in the field.
>
> **Revisions**
>
> 1. Table 9: Impact of CoT compositions.
> 2. L1082-L1124 and Figures 9-13: Visualizations of how the Draw-In-Mind workflow impacts image editing.
> 3. L1298-L1348 and Figures 14-16: Visualizations and analysis of failure cases.

---

> ### Author Response · Authors · 2025-11-20
> **Response to Reviewer RiTv-Part 6/8**
>
> Q1: Is there no way for a unified model to unify both thinking/reasoning and generation? If not, what is the significance of unifying understanding and generation? (Just for discussion)
>
> A: We find this question intriguing. In fact, we have implemented a unified framework for reasoning and generation within DIM-4.6B-Edit by utilizing its inherent Qwen2.5-VL-3B model as the designer. Our experiments show that fine-tuning this internal module on the DIM-Edit CoT data leads to improved performance. Although a comprehensive exploration of this unified training paradigm extends beyond the scope of the current paper, we provide the results below to demonstrate its effectiveness:
>
> | Designer | Add | Adjust | Extract | Replace | Remove | Background | Style | Hybrid | Action | Overall |
> | :--- | :---: | :---: | :---: | :---: | :---: | :---: | :---: | :---: | :---: | :---: |
> | - | 3.53 | 3.23 | 2.01 | 3.49 | 1.47 | 3.42 | 4.79 | 2.35 | 3.64 | 3.10 |
> | Qwen2.5-VL-3B❄️ | 3.80 | 3.24 | 2.03 | 3.89 | 3.21 | 3.52 | 4.92 | 2.71 | 4.05 | 3.49 |
> | Qwen2.5-VL-3B🔥 | 3.96 | 3.36 | 2.25 | 3.98 | 3.31 | 3.81 | 4.95 | 2.83 | 4.02 | 3.61 |
> | GPT-4o | 4.09 | 3.47 | 2.30 | 4.00 | 3.43 | 3.87 | 4.92 | 2.85 | 4.08 | 3.67 |
>
> **The second row corresponds to a "self-play" configuration, where the CoT embeddings generated by the internal MLLM are directly fed into the painter to execute edits, effectively bypassing the need for an additional external designer inference round.** To maintain simplicity and ensure consistency with our primary narrative regarding external designer integration, we chose not to emphasize this specific mode in the main text in our manuscript. Now we have included this discussion in the revision.
>
> For a detailed discussion on the significance of **unifying understanding and editing**, please refer to our response to Q2.
>
> **Revisions**
>
> 1. L454-L462 and Table 7: Integrated end-to-end evaluation.

---

> ### Author Response · Authors · 2025-11-20
> **Response to Reviewer RiTv-Part 7/8**
>
> Q2: If this paper splits generation and understanding into two components, how does it differ from prior encoder-side work in models that use an MLLM to unify generation tasks? Is the main difference simply the addition of CoT?
>
> A: We apologize for the ambiguity in our original statement. We wish to clarify that **unifying image editing and understanding** is not equivalent to **unifying image generation and understanding**. Our work represents a pioneering contribution specifically in the realm of editing-understanding unification, advancing the field from two distinct perspectives:
>
> From an architectural view, while advanced unified models like Show-o2 excel in T2I and understanding, image editing presents unique challenges that differentiate it from standard generation tasks. We elaborate on this distinction and our pioneering role below:
>
> **1. Task Nature and Modality Integration:**
> T2I primarily involves modality transfer between text and visuals. In contrast, image editing is a complex multimodal-to-visual task that necessitates the simultaneous processing of reference images and instructions. Traditional approaches (e.g., InstructP2P, UltraEdit) decouple these modalities, treating the reference image merely as a pixel constraint and encoding instructions via pure text encoders like T5. This separation inevitably leads to semantic loss, as the instruction is not internally grounded in the reference image.
>
> **2. Pioneering UMM-Powered Architecture:**
> The success of unified generation and understanding has inspired a shift toward Unified Multimodal Model (UMM)-powered editing. However, excluding concurrent work period (post-June), research in this specific domain remains sparse. The primary representative, Step1X-Edit, links a frozen Qwen2.5-VL-7B with a 12B FLUX via a specialized connector but lacks sophisticated instruction handling mechanisms. Our work represents a pioneering attempt to fully leverage UMM capabilities, introducing CoT reasoning to realize advanced, instruction-faithful image editing. We believe the just-released Nano Banana Pro's equipment of CoTs also proves that our work is a timely contribution for developing advanced editing model.
>
> From a data perspective, which is paramount for this Dataset Track submission, we provide a valuable asset comprising high-quality CoT-paired editing data, a contribution widely recognized by the reviewers. We observe that in the current academic landscape, "open-source" typically refers only to model weights, while high-quality training data remains scarce. Advanced UMMs (e.g., BAGEL) are primarily developed by major technology companies using proprietary in-house data. Consequently, we identified a critical bottleneck: it is difficult to train a unified T2I model with robust long-context instruction handling capabilities due to this data shortage. Since this capability is a prerequisite for complex CoT-guided editing, we contribute DIM-T2I, a dataset consisting of 14 million long and complex instructions, to the open-source community. We trained a model exclusively on DIM-Edit to justify the necessity of DIM-T2I for image editing. The results are reported in the following table:
>
> | Initialization | Add | Adjust | Extract | Replace | Remove | Background | Style | Hybrid | Action | Overall |
> | :--- | :---: | :---: | :---: | :---: | :---: | :---: | :---: | :---: | :---: | :---: |
> | Scratch | 2.70 | 2.56 | 1.93 | 2.23 | 2.47 | 2.82 | 4.68 | 2.38 | 2.15 | 2.66 |
> | DIM-4.6B-T2I | 4.09 | 3.47 | 2.30 | 4.00 | 3.43 | 3.87 | 4.92 | 2.85 | 4.08 | 3.67 |
>
> As evident from the results, the model trained from scratch significantly underperforms the version initialized with DIM-4.6B-T2I. This performance gap empirically validates the necessity of DIM-T2I as a cornerstone for the Draw-In-Mind paradigm. Furthermore, the robust long-instruction comprehension capabilities established by this dataset extend beyond editing, serving as a foundation for various advanced generation tasks and establishing DIM-T2I as a valuable asset for the broader research community.

---

> ### Author Response · Authors · 2025-11-20
> **Response to Reviewer RiTv-Part 8/8**
>
> (Q2 Continue)
>
> In the domain of image editing, we pioneer the proposal of DIM-Edit, the first high-quality Chain-of-Thought dataset specifically tailored for UMM-powered image editing systems. Despite its modest size of 233K samples, DIM-4.6B-Edit outperforms competitors that are 5x larger, while utilizing significantly fewer parameters. This serves as strong evidence for the effectiveness and high quality of our DIM data. Notably, even our version trained from scratch matches the performance of UltraEdit (2.70), a model initialized from SD3 and post-trained on 4 million editing pairs, demonstrating a nearly **20x improvement in data efficiency**. Furthermore, the clear performance gains observed when utilizing CoTs from the fine-tuned Qwen2.5-VL-3B demonstrate the **dual utility** of our dataset: DIM-Edit is not only effective for directly training editing behaviors but also serves as a robust corpus for enhancing the planning of accurate "design blueprints" on the understanding side.
>
> In summary, we provide a **full-stack data infrastructure** ranging from foundational long-instruction comprehension (DIM-T2I) to advanced, complex CoT-guided editing (DIM-Edit). This infrastructure democratizes access and encourages broader engagement in advanced image editing research. We believe these contributions are highly meaningful for the academic community.
>
> **Revisions**
>
> 1. L519-L527 and Table 10: Necessity of DIM-T2I.
>
> |----------------------------------------|
>
> Q3: Why claim a unified understanding–generation model without evaluating understanding tasks or other generation tasks?
>
> A: Thank you for this comment. We wish to clarify that the primary objective of this work is to leverage a unified architecture to enhance **image editing** through the proposed Draw-In-Mind paradigm and the DIM dataset. Regarding the evaluation of understanding capabilities, please refer to our response to **W2**. As this work specifically targets image editing, general image generation tasks fall outside the current scope. Consequently, we have focused our evaluation on widely recognized image editing benchmarks, including the newly incorporated **MagicBrush**, as per your suggestion.
>
> **Revisions**
>
> 1. L417-L424 and Table 5: MagicBrush evaluation.
>
> **If you have any further concerns, please feel free to leave us a message. We are willing to resolve them!**

---

> ### Comment · Reviewer_RiTv · 2025-11-24
>
> I believe the authors have not sufficiently reviewed prior work on using MLLMs for understanding in generation/editing tasks, and I recommend adding discussion of the papers listed blow.
> [1] Guiding instruction-based image editing via multimodal large language models
> [2] Kosmos-G: Generating Images in Context with Multimodal Large Language Models
> [3] MIGE: Mutually Enhanced Multimodal Instruction-Based Image Generation and Editing
> However, the authors have included additional experiments that address some of my concerns. I am willing to raise my score.

---

> > ### Author Response · Authors · 2025-11-24
> > **Thanks for the timely response and acknowledgement**
> >
> > Thank you for your timely response and acknowledgement of our work. We appreciate your recommendation of the relevant papers we missed, and we agree that discussing these works will significantly strengthen our paper.
> >
> > As you correctly pointed out, there have been several attempts to leverage MLLMs to guide image editing from a **modeling perspective**. For example:
> >
> > 1.  **MGIE [1]** tunes the Text Embeddings/Image Tokenizer (Adapter)/LM head to produce concise instructions (e.g., 22.7 tokens) and leverages N learnable `[IMG]` tokens to query latent visual imaginations from the instruction. These `[IMG]` tokens are subsequently transferred to L learnable queries through a 4-layer encoder-decoder transformer (connector) to act as editing conditions.
> > 2.  **Kosmos-G [2]** adopts a resampler to transfer a single image into several query tokens. They first align these query tokens with the inherent MLLM space by fine-tuning the entire MLLM. Later, a deeper alignment is conducted through a tailored AlignerNet (connector) to transfer multimodal instructions into the generation space.
> > 3.  **MIGE [3]** adopts a Q-Former to translate an image into 32 queries. These queries later attend to VAE features from the same image to retain low-level details. The final multimodal sequence is fed to an LLM for fine-tuning before acting as conditions for generation.
> >
> > **Interestingly, we believe our work serves as a distinct complement to these approaches. While they focus on architectural modifications, we approach the problem from the data perspective, yielding the following unique characteristics:**
> >
> > 1.  **Native Arbitrary-Length Multimodal Condition:** All the aforementioned works leverage a finite number of learnable queries to transfer the visual modality into the generation condition space (e.g., MGIE’s `[IMG]` tokens and transferred queries, Kosmos-G’s resampler, and MIGE’s Q-Former). This inevitably leads to information loss due to the limited representation space of learnable queries. In contrast, our DIM-4.6B-Edit retains full, native MLLM tokens, where images are deliberately tokenized by advanced MLLM tokenizers with minimal information loss.
> > 2.  **Minimal Condition Transfer Requirement:** From a data perspective, we demonstrate that by carefully curating high-quality, long, and complicated instructions (DIM-T2I and DIM-Edit), we can achieve high editing performance using only a simple two-layer MLP and frozen MLLM native tokens. This eliminates the need for sophisticated connector designs and avoids the risks associated with unstable and costly MLLM/connector training.
> > 3.  **Generalizability to Various MLLMs:** We demonstrate that our DIM dataset provides high generalizability regarding the core MLLM, evidenced by stable editing performance across different open- and closed-source designers. This offers the flexibility to seamlessly upgrade to more advanced MLLMs to achieve better performance in real-world deployments. Conversely, the works mentioned above require training the LLM (and other bottom components) to some extent, making them less flexible regarding model upgrades in practical applications.
> >
> > We believe the referenced works are highly insightful from the modeling side; however, they are constrained by the lack of high-quality, complex editing blueprints that align with actual editing behaviors. Consequently, they must make compromises, such as reducing condition length, compressing images via learnable queries, designing tailored connectors, and requiring MLLM tuning, to facilitate training. We have made an early attempt to fully unleash the power of native UMM tokens, supporting ultra-long and detailed CoT blueprints. This effectively relieves the painter from the designer role, thereby achieving better edits. Our work offers a robust solution from the data side, effectively complementing the current research field.
> >
> > We promise to cite these papers and include the relevant discussion in the final revision. We have opted not to modify the text immediately to avoid altering the line numbers, which could be inconvenient for other reviewers referencing specific lines in their responses.
> >
> > Again, thanks for providing us with the opportunity to engage in a broader and constructive discussion in this field :)
> >
> > [1] Guiding instruction-based image editing via multimodal large language models
> >
> > [2] Kosmos-G: Generating Images in Context with Multimodal Large Language Models
> >
> > [3] MIGE: Mutually Enhanced Multimodal Instruction-Based Image Generation and Editing

---

### Official Review · Reviewer_ui71 · 2025-11-03

**Soundness:** 3
**Presentation:** 2
**Contribution:** 3
**Rating:** 6
**Confidence:** 3

**Summary:**

This paper introduces the Draw-In-Mind (DIM) dataset and corresponding framework, which rebalances the roles of designer and painter in text-guided image editing. The authors identify an important issue in current editing models, that is, an imbalanced division of responsibilities, where the generator is overloaded with reasoning and design tasks. To address this, they propose the DIM dataset, consisting of DIM-T2I (14M long-context image–text pairs with CoT blueprints) and DIM-Edit (233K image editing pairs with CoT blueprints). Based on DIM-Edit, the proposed DIM-4.6B-Edit model achieves SOTA performance on image editing benchmarks with a simple architecture.

The paper’s main strengths lie in its explicit introduction of CoT reasoning into image editing, its high-value datasets and clear annotation taxonomy. The main finding of this paper is that comprehension, rather than model size, is key to editing performance. The experiments are thorough and convincing.

However, the narrative focus could be more consistent, as T2I experiments slightly distract from the editing theme. The relationship between DIM-T2I and DIM-4.6B-Edit is unclear, and there are fairness concerns about using GPT-4o as an external reasoning module.

**Strengths:**

1. This paper explicitly introduces CoT reasoning in image editing and is the first to leverage an external designer to produce textual
blueprints in a chain-of-thought style, which enables precise and stable image editing.
2. This paper constructs two high value datasets for T2I and image editing, where DIM-Edit comprises 233K carefully designed CoT imaginations derived from existing editing datasets. If released publicly, this dataset would be extremely valuable for community-based MLLM image editing research.  Moreover, the paper also provides explicit annotation dimensions for both T2I and image editing,
which would be highly helpful references for future multimodal dataset construction.
3. The paper demonstrates that, for image editing, model size is not the dominant performance factor. Instead, improving the model’s instruction comprehension and text–image alignment capability is far more important. This offers an insightful and meaningful perspective for the field.
4. The proposed DIM-4.6B-Edit model adopts an extremely simple architecture, which is just a two-layer MLP connecting an MLLM and a diffusion decoder, but it achieves SOTA performance on image editing tasks. This indicates that the paper’s data annotation paradigm has great potential for more sophisticated architectures.
5. To ensure strict independence between training and evaluation data, the paper takes great care to avoid any data leakage. The experiments and analyses are sufficient, convincing, and logically coherent.

**Weaknesses:**

Seen in Questions

**Questions:**

1. The title "DRAW-IN-MIND: REBALANCING DESIGNER–PAINTER ROLES IN UNIFIED MULTIMODAL MODELS BENEFITS IMAGE EDITING" emphasizes image editing as the main goal. However, the main experiments also include extensive T2I results (Table 2), which slightly dilute the paper’s thematic focus. The paper appears to have initially targeted both T2I and image editing but was later revised toward editing, leaving traces of both directions throughout the text. While this does not affect the experimental validity, it affects narrative coherence.
Future revisions could streamline the storyline to emphasize editing as the main task, framing T2I results as auxiliary evidence that supports DIM-Edit rather than a co-primary objective.

2. Line 479-480 states

>  "DIM-T2I, 14M web-crawled image–text pairs with carefully crafted long-context prompts that
provide a foundation for complex CoT comprehension in editing."

However, it remains unclear whether DIM-4.6B-Edit is trained based on DIM-4.6B-T2I (i.e., fine-tuned from its checkpoint) or whether it is simply trained with the same architecture on different datasets. If the latter is true, mentioning T2I results in the main narrative feels somewhat off-topic. Clarifying this relationship would make the methodology more coherent, that is, whether DIM-4.6B-Edit is initialized from DIM-4.6B-T2I.

3. Line 297-299 states

> "During inference, we employ an external designer, e.g., GPT-4o, to prepare a blueprint in the same format as DIM-Edit, except without access to the target image, ensuring alignment with real usage scenarios."

Although Table 2 compares DIM-4.6B-T2I with unified base models such as Show-o and BAGEL, DIM adopts a modular design that includes an external reasoning interface (GPT-4o). Strictly speaking, this introduces an auxiliary capability that is unavailable to other baselines.
A more rigorous comparison would involve providing the same GPT-4o CoT blueprints to all models, to confirm whether the observed improvements stem from DIM’s superior CoT comprehension or from GPT-4o’s reasoning power itself.

4. In Figure 2 (third example on the right), the optimized prompt reads

> "Remove the lower sign and the sticker on the ‘No Parking’ sign."

However, the sticker mentioned in the text is not clearly visible in the original image. This inconsistency should be corrected or clarified.

5. The paper uses an LLM to automatically evaluate CoT quality with a fine-grained scoring scheme (Line 254). However, the paper would be stronger if a small manually verified subset were included to cross-validate the LLM’s automatic scoring results. Such a human validation experiment could be briefly mentioned in the main text, with full results moved to the appendix.

6. In Figure 1 (lower part), the meanings of the blue and red colors on the left and right sides are not explained. A short explanation clarifying what each color represents would improve readability.

7. Line119-122 states

> "To ensure broad concept coverage, we opt for harvesting real-world images and annotate them with our in-house models from 21 dimensions, yielding 14M long-context image–text pairs, namely DIM-T2I, that form a robust foundation for complex CoT-guided editing."

It is unclear whether these 21 dimensions were derived from empirical analysis of existing datasets, theoretical considerations, or prior taxonomies. Providing the rationale or references for these dimensions would enhance transparency and reproducibility.

8. The editing dataset, UltraEdit-160K-CoT, is filtered using SSIM, DINOv2, and CLIP similarity metrics before CoT annotation (Line 209).
However, later in the pipeline, the data are again classified into misaligned, partially aligned, and aligned categories. Is the filtering in the first stage necessary? Why? how does it differ from the later alignment classification?

9. Line 862-863

> "We include only 8K images from the training set of MagicBrush to avoid potential information leakage during evaluation. "

Line 269

> "we exclude distillation datasets such as BLIP3-o-60K (Chen et al., 2025a) to avoid any risk of data leakage"

It is not explicitly explained why such leakage might occur. A short clarification of the specific risk would make this precaution clearer and more accessible to less familiar readers.

---

> ### Author Response · Authors · 2025-11-20
> **Response to Reviewer ui71-Part 1/6**
>
> Q1: The title emphasizes image editing as the main goal. However, the main experiments also include T2I results, which slightly dilute the paper’s thematic focus. Future revisions could **streamline the storyline to emphasize editing as the main task, framing T2I results as auxiliary evidence that supports DIM-Edit rather than a co-primary objective.**
>
> A: Thanks for the suggestion! We would like to clarify the motivation for building DIM-T2I from two perspectives, along with supportive experiments, to demonstrate that **developing DIM-T2I and DIM-Edit are not co-primary objectives. Rather, they are complementary, where DIM-T2I serves as the cornerstone for the training success on DIM-Edit**:
>
> **[Motivation for T2I Training]** Upon deciding the model architecture, our first objective was to align the condition space between the MLLM (Qwen2.5-VL-3B) and the DiT (SANA1.5-1.6B). To achieve this, we considered two strategies:
>
> 1. Learning to generate first (T2I), followed by adaptation for editing.
> 2. Directly learning the editing task.
>
> We observe that the vast majority of image editing models are built upon established T2I foundations. This aligns with the first strategy and represents a robust technical route that benefits from *curriculum learning*. Consequently, we elected to achieve modality alignment via T2I training first, and subsequently fine-tune this model (DIM-4.6B-T2I) for the more challenging editing task.
>
> **[Motivation for Building DIM-T2I]** Our ultimate goal is to leverage long, complex design blueprints in a CoT style to alleviate the burden of detailed design on the painter. This requires the model to possess robust comprehension capabilities for complex instructions. We posit that T2I generation is a simpler task than editing and is better suited for fostering this capability. Rather than simultaneously tackling two challenging objectives, i.e., complex instruction comprehension and precise image editing, we chose to establish strong instruction comprehension during the T2I training stage.
>
> However, during data collection, we observed that mainstream publicly available datasets consist predominantly of simple, short prompts (as detailed in Table 1). While advanced open-source models capable of handling long instructions exist (e.g., BAGEL, Janus-Pro, Step1X-Edit), they are primarily developed by large technology companies using proprietary, in-house data. This discrepancy hinders the advancement of image editing research, as academic researchers are often restricted to building upon large, inflexible models where fine-tuning costs are prohibitive.
>
> To address this, we introduce DIM-T2I, a large-scale, publicly accessible dataset comprising 14 million images paired with ultra-long, high-quality prompts covering a wide range of world knowledge. While our primary motivation was to establish a foundation for CoT comprehension in subsequent editing tasks, DIM-T2I also empowers the research community to develop customized T2I models with advanced instruction support, which is a prerequisite for applications like CoT-guided editing. By providing this full-stack data infrastructure, we aim to democratize access and encourage broader engagement in advanced image editing research.
>
> **[Reason for Presenting T2I Results]** We present Table 2 to demonstrate that training on DIM-T2I enables open-source models to achieve generation capabilities and advanced instruction comprehension comparable to frontier models relying on proprietary in-house data (e.g., BAGEL). This evidence confirms that DIM-T2I provides the necessary foundation for the sophisticated CoT comprehension required for complex editing tasks.
>
> **[Additional Supportive Experiments]** Thank you for highlighting the ambiguity that may have led readers to view DIM-T2I and DIM-Edit as co-primary objectives. In response to your feedback, we trained a model exclusively on DIM-Edit to test the feasibility of simultaneously achieving modality alignment, complex instruction comprehension, and editing capabilities in a single stage. The results are reported in the following table:
>
> | Initialization | Add | Adjust | Extract | Replace | Remove | Background | Style | Hybrid | Action | Overall |
> | :--- | :---: | :---: | :---: | :---: | :---: | :---: | :---: | :---: | :---: | :---: |
> | Scratch | 2.70 | 2.56 | 1.93 | 2.23 | 2.47 | 2.82 | 4.68 | 2.38 | 2.15 | 2.66 |
> | DIM-4.6B-T2I | 4.09 | 3.47 | 2.30 | 4.00 | 3.43 | 3.87 | 4.92 | 2.85 | 4.08 | 3.67 |
>
> As evident from the results, the model trained from scratch significantly underperforms the version initialized with DIM-4.6B-T2I. This performance gap empirically validates the necessity of DIM-T2I as a foundational cornerstone for the Draw-In-Mind paradigm. We have included this ablation study in the revised manuscript and updated the relevant sections to reflect these findings.
>
> **Revisions**
>
> 1. L159-L176: Motivation for DIM-T2I as a foundation of DIM-Edit.
> 2. L520-L527 and Table 10: Necessity of DIM-T2I.

---

> ### Author Response · Authors · 2025-11-20
> **Response to Reviewer ui71-Part 2/6**
>
> Q2: It remains unclear whether DIM-4.6B-Edit is trained based on DIM-4.6B-T2I (i.e., fine-tuned from its checkpoint) or whether it is simply trained with the same architecture on different datasets. **Clarifying whether DIM-4.6B-Edit is initialized from DIM-4.6B-T2I would make the methodology more coherent.**
>
> A: We appreciate the opportunity to clarify this relationship. DIM-4.6B-Edit is fine-tuned from DIM-4.6B-T2I. The T2I training stage establishes fundamental long-instruction comprehension capabilities. DIM-4.6B-Edit inherits this foundation to achieve sophisticated CoT comprehension, thereby realizing superior editing performance. As discussed in Q1, training on DIM-Edit from scratch results in significantly degraded performance, empirically demonstrating the necessity of the DIM-4.6B-T2I initialization. We have added these clarifications to the revised manuscript.
>
> **Revisions**
>
> 1. L297-L298: Clarification of initialization.
>
> |----------------------------------------|
>
> Q3: Although Table 2 compares DIM-4.6B-T2I with unified base models such as Show-o and BAGEL, DIM adopts a modular design that includes an external reasoning interface (GPT-4o). This introduces an auxiliary capability that is unavailable to other baselines. **A more rigorous comparison would involve providing the same GPT-4o CoT blueprints to all models, to confirm whether the observed improvements stem from DIM’s superior CoT comprehension or from GPT-4o’s reasoning power itself.**
>
> A: We apologize for the ambiguity. To clarify, DIM-4.6B-T2I is trained on long, complex instructions rather than CoT formatted instructions. The calling of an external designer occurs only during the editing inference phase. For the T2I task, no external models are involved, ensuring that the comparisons presented in Table 2 are fair.
>
> Still, we found your suggestion to be highly constructive. To investigate the impact of DIM-style CoT on different models, we evaluated the performance of **Janus-4o** and **Step1X-Edit** when directly provided with the same CoT blueprints as input instructions on ImgEdit. The results are presented below:
>
> | Model | Params | CoT | Add | Adjust | Extract | Replace | Remove | Background | Style | Hybrid | Action | Overall |
> | :--- | :---: | :---: | :---: | :---: | :---: | :---: | :---: | :---: | :---: | :---: | :---: | :---: |
> | DIM-4.6B-Edit | Und❄️ \| Gen🔥 | ✗ | 3.53 | 3.23 | 2.01 | 3.49 | 1.47 | 3.42 | 4.79 | 2.35 | 3.64 | 3.10 |
> | | | ✓ | 4.09 | 3.47 | 2.30 | 4.00 | 3.43 | 3.87 | 4.92 | 2.85 | 4.08 | 3.67 |
> | Janus-4o | Und🔥 \| Gen🔥 | ✗ | 3.35 | 3.35 | 2.25 | 3.01 | 2.18 | 3.32 | 4.71 | 2.49 | 4.04 | 3.19 |
> | | | ✓ | 3.95 | 2.74 | 2.49 | 3.59 | 2.28 | 3.31 | 4.72 | 2.62 | 4.02 | 3.30 |
> | Step1X-Edit | Und❄️ \| Gen🔥 | ✗ | 3.88 | 3.14 | 1.76 | 3.40 | 2.41 | 3.16 | 4.63 | 2.64 | 2.52 | 3.06 |
> | | | ✓ | 3.56 | 2.47 | 1.81 | 3.13 | 2.02 | 2.84 | 4.18 | 1.80 | 2.46 | 2.70 |
>
> Based on these results, we have the following observations and analysis:
>
> + **DIM-4.6B-Edit** is explicitly trained on complex CoT-style blueprints from the DIM-Edit dataset, it achieves superior CoT comprehension. Consequently, it demonstrates substantial performance gains when DIM-style CoTs are applied during inference.
> + **Janus-4o** employs an end-to-end fine-tuning approach, which minimizes the gap between instruction understanding and generation. This makes it more robust to input distribution shifts. While it possesses mild CoT comprehension capabilities and benefits slightly from DIM-style CoTs, the performance gain is less pronounced compared to DIM-4.6B-Edit.
> + **Step1X-Edit** adopts a training recipe similar to ours (using a frozen understanding core), this design makes it susceptible to input distribution shifts when facing unseen instruction formats. It struggles to process CoT inputs effectively, leading to performance degradation when DIM-style CoTs are applied.
>
> Based on these findings, we conclude that **superior CoT comprehension is pivotal for enhancing editing performance**. This finding validates our strategy of fostering CoT comprehension by constructing DIM-T2I and utilizing DIM-4.6B-T2I as the initialization for the editing task.

---

> ### Author Response · Authors · 2025-11-20
> **Response to Reviewer ui71-Part 3/6**
>
> (Q3 Continue)
>
> **Furthermore, the high quality of our DIM-CoT data is corroborated by the fact that external models, such as Janus-4o, also derive performance benefits from it.** This phenomenon motivated us to conduct a further validation experiment by fine-tuning Qwen2.5-VL-3B on the CoT blueprints from DIM-Edit. The results are presented below:
>
> | Designer | Add | Adjust | Extract | Replace | Remove | Background | Style | Hybrid | Action | Overall |
> | :--- | :---: | :---: | :---: | :---: | :---: | :---: | :---: | :---: | :---: | :---: |
> | - | 3.53 | 3.23 | 2.01 | 3.49 | 1.47 | 3.42 | 4.79 | 2.35 | 3.64 | 3.10 |
> | Qwen2.5-VL-3B❄️ | 3.80 | 3.24 | 2.03 | 3.89 | 3.21 | 3.52 | 4.92 | 2.71 | 4.05 | 3.49 |
> | Qwen2.5-VL-3B🔥 | 3.96 | 3.36 | 2.25 | 3.98 | 3.31 | 3.81 | 4.95 | 2.83 | 4.02 | 3.61 |
> | GPT-4o | 4.09 | 3.47 | 2.30 | 4.00 | 3.43 | 3.87 | 4.92 | 2.85 | 4.08 | 3.67 |
>
> The results in the table demonstrate that using CoTs from the fine-tuned Qwen2.5-VL-3B leads to clear performance gains during evaluation. This serves as further validation of the high quality of our proposed DIM-Edit dataset. Thank you for this valuable suggestion! We have updated the manuscript accordingly.
>
> **Revisions**
>
> 1. L895-L913 and Table 14: Impact of DIM CoT for different models.
> 2. L454-L462 and Table 7: Performance using CoTs from DIM-Edit CoT-finetuned Qwen2.5-VL-3B.
>
> |----------------------------------------|
>
> Q4: In Figure 2 (third example on the right), the optimized prompt reads "Remove the lower sign and the sticker on the ‘No Parking’ sign.", **However, the sticker mentioned in the text is not clearly visible in the original image.** This inconsistency should be corrected or clarified.
>
> A: Sorry for the ambiguous figure. Accurately speaking, the sticker is on the letter "A" at the bottom. The GPT-4o (and also other MLLMs) may treat the whole pole as a "No Parking" sign during refinement. To verify our assumption, we provide the same refined prompt to the most advanced editing model, i.e., Nano Banana, and the behavior is consistent with our model, where the 3-WAY board and the sticker on it are removed. This sample is indeed a bit ambiguous, and we have replaced it with another clearer one. Thanks for the detailed observation!
>
> **Revisions**
>
> 1. Figure 2: Revised sample for prompt refinement.

---

> ### Author Response · Authors · 2025-11-20
> **Response to Reviewer ui71-Part 4/6**
>
> Q5: The paper uses an LLM to automatically evaluate CoT quality with a fine-grained scoring scheme (Line 254). However, **the paper would be stronger if a small manually verified subset were included to cross-validate the LLM’s automatic scoring results.** Such a human validation experiment could be briefly mentioned in the main text, with full results moved to the appendix.
>
> A: We appreciate this valuable suggestion. In response, we conducted a human verification study to ensure data quality. Specifically, we randomly sampled 25 instances from each of the data sources listed in Appendix C, resulting in a comprehensive evaluation set of 100 samples. Three human annotators were then recruited to assess the quality of the CoTs from two distinct perspectives:
>
> 1. **Evaluation of Optimized Instructions (Start of CoT):** We presented annotators with both the raw instructions and the optimized instructions from DIM-Edit, alongside the corresponding source-edit image pairs. Annotators were tasked with selecting the instruction that best reflected the actual editing operations. A "Tied" option was included for cases where neither instruction was sufficiently accurate. The metric reported is the average win rate for each instruction type.
> 2. **Evaluation of Edited Image Imagination (End of CoT):** We asked annotators to assess the alignment between the Edited Image Imagination (EII) and the actual edited image. The quality was categorized into three levels: (i) *Perfect Match*, (ii) *Minor Errors*, and (iii) *Major Errors*. The metric reported is the percentage distribution across these error levels.
>
> This efficient evaluation protocol enables a rapid yet robust assessment of the overall CoT quality within DIM-Edit. The results for both the instruction optimization (Win Rate) and the Edited Image Imagination (Error Distribution) are summarized below:
>
> | Instruction Type | DIM-Edit | Raw | Tied |
> | :--- | :---: | :---: | :---: |
> | **Win Rate** | 89% | 5% | 6% |
>
> | EII Quality | Perfect Match | Minor Error | Major Error |
> | :--- | :---: | :---: | :---: |
> | **Percentage** | 81% | 15% | 4% |
>
> Based on these observations, we have the following analysis:
>
> 1. **Consistency with MLLM Assessment:** These results align closely with the MLLM-based quality assessment presented in Appendix D, where over 80% of DIM-Edit CoTs were judged clearer than the raw instructions, with no factual errors detected. Even in "Tied" cases where the optimization was not deemed strictly superior, the semantics of the raw instruction were fully preserved, ensuring that the optimization process introduces no regression.
> 2. **Analysis of Minor Errors:** We observed that minor errors typically relate to subtle environmental inconsistencies, such as slight shifts in brightness (e.g., "the image should be a bit lighter"). These artifacts usually stem from the VAE's inability to perfectly reconstruct raw images in AI-generated pairs (e.g., from UltraEdit), leading to a slight loss of high-frequency features. As these discrepancies are barely perceptible to the human eye, they have a negligible impact on overall training efficiency.
> 3. **Analysis of Major Errors:** Instances classified as having major errors generally correspond to extremely challenging scenarios where the edits are minute (e.g., the removed object occupies less than 2% of the pixels). These cases are difficult even for human annotators and advanced MLLMs like GPT-4o. Given their extreme rarity, these outliers do not adversely affect the stability of the training procedure.
>
> Overall, the CoTs produced by our DIM-Edit pipeline maintain high quality and serve as effective design blueprints. This high data quality directly translates to better editing capabilities, as evidenced by the superior performance of the DIM-4.6B-Edit model trained on this dataset. We have included related discussion in the revision.
>
> **Revisions**
>
> 1. L256 & L1509-L1558 and Figure 18: User study of DIM-Edit's CoT.
>
> |----------------------------------------|
>
> Q6: **In Figure 1 (lower part), the meanings of the blue and red colors on the left and right sides are not explained.** A short explanation clarifying what each color represents would improve readability.
>
> A: We apologize for the ambiguity in Figure 1. To clarify, the blue and red blocks on the left represent frozen and trainable parameters, respectively. On the right, the blue columns indicate the performance of baseline models, while the red columns denote the performance of DIM-4.6B-Edit utilizing different designers. We have added the corresponding legends to the revised figure to ensure clarity.
>
> **Revisions**
>
> 1. Figure 1: Adding legends for a clearer comprehension.

---

> ### Author Response · Authors · 2025-11-20
> **Response to Reviewer ui71-Part 5/6**
>
> Q7: It is unclear whether the 21 dimensions of DIM-T2I were derived from empirical analysis of existing datasets, theoretical considerations, or prior taxonomies. **Providing the rationale or references for these dimensions** would enhance transparency and reproducibility.
>
> A: Very good question. **The 21 dimensions were derived from a thorough literature review and an empirical analysis of existing understanding datasets and benchmarks.** They are listed as follows:
>
> MME (Fu et al., 2025), MMMU (Yue et al., 2024), MMMU-Pro (Yue et al., 2025), MMLU (Hendrycks et al., 2020), MMStar (Chen et al., 2024b), MMT-Bench (Ying et al., 2024), MM-Vet (Yu et al., 2023), MM-Vet V2 (Yu et al., 2024), LLaVA-Bench-Wild (Liu et al., 2023a), LLaVA-Bench-Wilder (Li et al., 2024a), WildVision (Lu et al., 2024), COCO (Lin et al., 2014), VQAv2 (Goyal et al., 2017), OK-VQA (Marino et al., 2019), TextCaps (Sidorov et al., 2020), TextVQA (Singh et al., 2019), AI2D (Kembhavi et al., 2016), ChartQA (Masry et al., 2022), DocVQA (Mathew et al., 2021), MathVista (Lu et al., 2023), MIA-Bench (Qian et al., 2024), MegaBench (Chen et al., 2024a), RWQA (xAI, 2024), OCRBench (Liu et al., 2023b), GSM8K (Cobbe et al., 2021), GPQA (Rein et al., 2024), IFEval (Zhou et al., 2023).
>
> We believe that the aspects emphasized in widely recognized understanding datasets and benchmarks effectively capture the most frequent interactions between humans and objects in the real world. This makes them an ideal foundation for learning text-to-image generation tasks involving long and complex instructions. By constructing prompts that span these diverse fields, DIM-4.6B-T2I not only masters long-form instruction processing but also acquires the broad world knowledge necessary to facilitate sophisticated CoT comprehension and precise editing, thereby achieving high GenEval scores and low FID on MJHQ-30K. We have included this clarification in the revision :)
>
> **Revisions**
>
> 1. L176 & L1569-L1586: References and clarifications for the 21 analyzed dimensions in DIM-T2I.
>
> |----------------------------------------|
>
> Q8: The editing dataset, UltraEdit-160K-CoT, is filtered using SSIM, DINOv2, and CLIP similarity metrics before CoT annotation (Line 209). However, later in the pipeline, the data are again classified into misaligned, partially aligned, and aligned categories. **Is the filtering in the first stage necessary? Why? how does it differ from the later alignment classification?**
>
> A: Good question. We would like to clarify the distinct roles of these metrics. SSIM, DINOv2, and CLIP similarity are **image-to-image** metrics used to evaluate the visual similarity between the source and edited images. Higher scores indicate better preservation of the background and non-edited regions. Including a sufficient proportion of pairs with high values in these metrics improves overall visual consistency and stabilizes training.
>
> In contrast, the prompt quality judgment and optimization in the DIM-Edit pipeline focus on **image-text** alignment. The objective is to generate precise editing instructions that align perfectly with the actual editing behavior, thereby preventing noisy supervision signals caused by over-simplified prompts. For example, if an instruction is simply "add dogs" but the ground truth edit contains exactly two dogs, a model generating three dogs might be unfairly penalized despite technically fulfilling the request. By refining the instruction to "add two dogs," we eliminate this ambiguity, ensuring that the training signals are precise and efficient. We have expanded the explanation of these metrics in the revised manuscript to clarify their respective functions.
>
> **Revisions**
>
> 1. L1462-L1466: Clarifications of image-to-image metrics used in data filtering.

---

> ### Author Response · Authors · 2025-11-20
> **Response to Reviewer ui71-Part 6/6**
>
> Q9: It is not explicitly explained why such leakage (in BLIP3-o-60K) might occur. **A short clarification of the specific risk would make this precaution clearer and more accessible to less familiar readers.**
>
> A: We appreciate your suggestion, and would like to clarify our rationale regarding the exclusion of BLIP3-o-60K :)
>
> GenEval is a benchmark designed to evaluate specific capabilities such as spatial reasoning and counting (e.g., "a photo of a dog left of a cat" or "a photo of three dogs"). BLIP3-o-60K is a training dataset explicitly curated to align with these structural patterns. Although the exact test prompts from GenEval are not present in BLIP3-o-60K, the underlying data distributions are nearly identical. For instance, training on "a cat right of a dog" allows a model to easily solve the test case "a dog left of a cat." Previous works have identified this practice as a form of data leakage or "benchmark hacking."
>
> We believe that integrating a specialized corpus solely to artificially inflate "SOTA" scores on specific benchmarks undermines the scientific value of a dataset paper. Consequently, we explicitly excluded such data to ensure a fair and meaningful evaluation. We have revised the manuscript to make this distinction clearer to a broader audience.
>
> **Revisions**
>
> 1. L288-L291: Clarifications for excluding distillation datasets like BLIP3-o-60K.
>
> **If you have any further concerns, please feel free to leave us a message. We are willing to resolve them!**

---

### Author Response · Authors · 2025-11-20
**General Response to All Reviewers**

We sincerely thank the reviewers for their dedicated time and constructive feedback. We are encouraged that all reviewers acknowledge the significance of the DIM data contribution. And we are also glad to see that the just-released Nano Banana Pro adopts the plan-then-edit idea similar to our Draw-In-Mind paradigm to significantly enhance editing performance, which verifies the effectiveness of our paradigm from an industrial perspective to some extent.

The insightful suggestions from the four reviews have significantly helped to improve the quality of our paper. We have addressed all concerns point-by-point below and incorporated the corresponding revisions in blue fonts into the manuscript. **We have explicitly marked the location of the relevant revisions for each question and respectfully invite the reviewers to refer to the revised manuscript for the best reading experience.** Please refer to the reviewer-specific responses below for details :)

---

### Author Response · Authors · 2025-11-29
**Response Summary for Area Chair**

Dear Area Chair,

Thank you for managing the review of our paper. We sincerely appreciate the significant effort put forth by you and the other four reviewers, and we are pleased that all reviewers have now recognized our work.

We treasure the insightful discussion with reviewers, and would like to highlight that **this paper initially received three positive reviews (8, 6, 6) and one negative review (4). Following a constructive two-round discussion with Reviewer RiTv (see "Official Comment by Reviewer RiTv" on Nov 24), the reviewer also acknowledged our work. Consequently, the overall scores are raised from 8/6/6/4 to 8/6/6/6 on Nov 24—three days prior to the API leakage notification.**

We have rigorously addressed all concerns point-by-point, with revisions highlighted in blue in the revised manuscript. All 1st-round responses were submitted by the initial recommended timeline (before Nov 21) and remain unedited for transparency. We provide summarized responses and major revisions below to facilitate your assessment; please refer to specific threads for full details.

---

> ### Author Response · Authors · 2025-11-29
> **Major Revisions (Highlighted by Blue Fonts)**
>
> 1. `L159-L176`: Motivation for DIM-T2I as a foundation of DIM-Edit. *(Reviewer ui71-Q1)*
> 2. `L288-L291`: Clarifications for excluding distillation datasets like BLIP3-o-60K. *(Reviewer ui71-Q9)*
> 3. `L297-L298`: Clarification of initialization. *(Reviewer ui71-Q2)*
> 4. `L417-L424` and Table 5: The performance of automated metrics on MagicBrush test set. *(Reviwer RiTv-W1/Q3)*
> 5. `L454-L462 and Table 7`: Integrated end-to-end evaluation and performance using CoTs from DIM-Edit CoT-finetuned Qwen2.5-VL-3B. *(Reviewer ui71-Q3, RiTv-Summary/Q1, PX1Q-W1, B3js-Q1)*
> 6. `L519-L527 and Table 10`: Necessity of DIM-T2I. *(Reviewer ui71-Q1, RiTv-Q2)*
> 7. `L895-L913 and Table 14`: Impact of DIM CoT for different models. *(Reviewer ui71-Q3)*
> 8. `L914-L921 and Table 15`: Understanding benchmarks' performances. *(Reviwer RiTv-W2/Q3)*
> 9. `L1082-L1124 and Figures 9-13`: Visualizations of how the Draw-In-Mind workflow impacts image editing. *(Reviewer RiTv-W3, PX1Q-Q1, B3js-W1)*
> 10. `L1298-L1348 and Figures 14-16`: Visualizations and analysis of failure cases. *(Reviewer RiTv-W3, PX1Q-Q1, B3js-W1)*
> 11. `L1462-L1466`: Clarifications of image-to-image metrics used in data filtering. *(Reviewer ui71-Q8)*
> 12. `L256/L1509-L1558 and Figure 18`: User study of DIM-Edit's CoT. *(Reviewer ui71-Q5)*
> 13. `L176/L1569-L1586`: References and clarifications for the 21 analyzed dimensions in DIM-T2I. *(Reviewer ui71-Q7)*
> 14. `Figure 1`: Adding legends for a clearer comprehension. *(Reviewer ui71-Q6)*
> 15. `Figure 2`: Revised sample for prompt refinement. *(Reviewer ui71-Q4)*
> 16. `Table 9`: Impact of CoT compositions. *(Reviewer RiTv-W3)*
> 17. `Follow-up discussion on related papers`: will be included in the final revision. *(Reviewer RiTv-2nd Round Discussion)*

---

> ### Author Response · Authors · 2025-11-29
> **Briefly Summarized Response to Reviewer ui71-Part 1/2**
>
> **Q1: Streamline the storyline to emphasize editing as the main task, rather than treating T2I as a co-primary objective.**
>
> A: We clarify that developing DIM-T2I and DIM-Edit are not co-primary objectives; rather, DIM-T2I serves as the necessary cornerstone for the editing task. Our motivation is threefold:
>
> 1. **Curriculum Learning:** We adopt a "T2I-first, Edit-second" strategy. Aligning the MLLM and DiT via T2I generation first is a more robust technical route than learning editing from scratch.
> 2. **Complex Instruction Comprehension:** Our Draw-In-Mind paradigm requires understanding long, complex blueprints. Since existing public datasets lack sufficient prompt complexity, we constructed DIM-T2I (14M images with ultra-long prompts) to bridge the gap between open-source resources and proprietary models.
> 3. **Validation:** Table 2 demonstrates that DIM-T2I enables our model to match the instruction-following capabilities of proprietary models (e.g., BAGEL), confirming it provides the essential foundation required for high-quality editing.
>
> To empirically validate the necessity of DIM-T2I, we conducted an ablation study by training a model exclusively on DIM-Edit (skipping the T2I stage). The results are listed in the following table:
>
> | Initialization | Add | Adjust | Extract | Replace | Remove | Background | Style | Hybrid | Action | Overall |
> | :--- | :---: | :---: | :---: | :---: | :---: | :---: | :---: | :---: | :---: | :---: |
> | Scratch | 2.70 | 2.56 | 1.93 | 2.23 | 2.47 | 2.82 | 4.68 | 2.38 | 2.15 | 2.66 |
> | DIM-4.6B-T2I | 4.09 | 3.47 | 2.30 | 4.00 | 3.43 | 3.87 | 4.92 | 2.85 | 4.08 | 3.67 |
>
> It shows that the model trained from scratch significantly underperforms the version initialized with DIM-4.6B-T2I. This confirms that DIM-T2I is essential for establishing the necessary alignment and instruction comprehension before tackling editing tasks.
>
> **Q2: Clarifying whether DIM-4.6B-Edit is initialized from DIM-4.6B-T2I.**
>
> A: `DIM-4.6B-Edit` is fine-tuned from `DIM-4.6B-T2I`, inheriting the fundamental long-instruction comprehension established during the T2I stage. This foundation enables the sophisticated CoT capabilities required for superior editing. As demonstrated in Q1, training from scratch results in significant degradation, empirically validating the necessity of the T2I initialization.
>
> **Q3: DIM includes an external reasoning interface, which is unavailable to other baselines. Providing the same GPT-4o CoT blueprints to all models, to confirm whether the observed improvements stem from DIM’s superior CoT comprehension or from GPT-4o’s reasoning power itself.**
>
> A: We clarify that `DIM-4.6B-T2I` is trained on long, complex instructions rather than CoT formats; the external designer is strictly for the editing inference phase only, ensuring Table 2 comparisons are fair. We further investigated the impact of DIM-style CoT on other models:
>
> | Model | Params | CoT | Add | Adjust | Extract | Replace | Remove | Background | Style | Hybrid | Action | Overall |
> | :--- | :---: | :---: | :---: | :---: | :---: | :---: | :---: | :---: | :---: | :---: | :---: | :---: |
> | DIM-4.6B-Edit | Und❄️ \| Gen🔥 | ✗ | 3.53 | 3.23 | 2.01 | 3.49 | 1.47 | 3.42 | 4.79 | 2.35 | 3.64 | 3.10 |
> | | | ✓ | 4.09 | 3.47 | 2.30 | 4.00 | 3.43 | 3.87 | 4.92 | 2.85 | 4.08 | 3.67 |
> | Janus-4o | Und🔥 \| Gen🔥 | ✗ | 3.35 | 3.35 | 2.25 | 3.01 | 2.18 | 3.32 | 4.71 | 2.49 | 4.04 | 3.19 |
> | | | ✓ | 3.95 | 2.74 | 2.49 | 3.59 | 2.28 | 3.31 | 4.72 | 2.62 | 4.02 | 3.30 |
> | Step1X-Edit | Und❄️ \| Gen🔥 | ✗ | 3.88 | 3.14 | 1.76 | 3.40 | 2.41 | 3.16 | 4.63 | 2.64 | 2.52 | 3.06 |
> | | | ✓ | 3.56 | 2.47 | 1.81 | 3.13 | 2.02 | 2.84 | 4.18 | 1.80 | 2.46 | 2.70 |
>
> As shown in the new table, `DIM-4.6B-Edit` achieves the highest gains given its explicit training. `Janus-4o` sees mild benefits, while `Step1X-Edit` suffers performance degradation due to sensitivity to input distribution shifts. This confirms that superior CoT comprehension is pivotal for editing performance.
>
> To further validate our data, we fine-tuned `Qwen2.5-VL-3B` on DIM-Edit blueprints. Results are shown below:
>
> | Designer | Add | Adjust | Extract | Replace | Remove | Background | Style | Hybrid | Action | Overall |
> | :--- | :---: | :---: | :---: | :---: | :---: | :---: | :---: | :---: | :---: | :---: |
> | - | 3.53 | 3.23 | 2.01 | 3.49 | 1.47 | 3.42 | 4.79 | 2.35 | 3.64 | 3.10 |
> | Qwen2.5-VL-3B❄️ | 3.80 | 3.24 | 2.03 | 3.89 | 3.21 | 3.52 | 4.92 | 2.71 | 4.05 | 3.49 |
> | Qwen2.5-VL-3B🔥 | 3.96 | 3.36 | 2.25 | 3.98 | 3.31 | 3.81 | 4.95 | 2.83 | 4.02 | 3.61 |
> | GPT-4o | 4.09 | 3.47 | 2.30 | 4.00 | 3.43 | 3.87 | 4.92 | 2.85 | 4.08 | 3.67 |
>
> The resulting performance gains confirm the high quality of the DIM-Edit dataset.
>
> **Q4: The third example on the right in Figure 2 has some inconsistency.**
>
> A: We apologize for the visual ambiguity and have replaced this sample with a clearer, unambiguous example in the revised manuscript.

---

> ### Author Response · Authors · 2025-11-29
> **Briefly Summarized Response to Reviewer ui71-Part 2/2**
>
> **Q5: The paper would be stronger if a small manually verified subset were included to cross-validate the LLM’s automatic scoring results of DIM-Edit.**
>
> A: To rigorously validate data quality, we conducted a human study on 100 randomly sampled instances (25 from each data source) using three annotators. The evaluation focused on two perspectives:
> 1. **Instruction Optimization (CoT Start):** Annotators compared raw vs. optimized instructions. Results (see Table below) show a high win rate for our optimized instructions, aligning with the MLLM assessments in Appendix D. Even "Tied" cases preserved semantic accuracy.
>
> | Instruction Type | DIM-Edit | Raw | Tied |
> | :--- | :---: | :---: | :---: |
> | **Win Rate** | 89% | 5% | 6% |
>
> 2. **Edited Image Imagination Alignment (CoT End):** We assessed the alignment between the Edited Image Imagination and the actual edits. Most samples showed perfect matches. Minor errors were largely negligible VAE reconstruction artifacts (e.g., slight brightness shifts), while major errors were extremely rare and limited to minute edits.
>
> | EII Quality | Perfect Match | Minor Error | Major Error |
> | :--- | :---: | :---: | :---: |
> | **Percentage** | 81% | 15% | 4% |
>
> These results confirm that DIM-Edit produces high-quality, stable CoT-style blueprints that effectively drive superior model performance.
>
> **Q6:  In Figure 1 (lower part), the meanings of the blue and red colors on the left and right sides are not explained.**
>
> A: We apologize for the ambiguity. We have updated Figure 1 with explicit legends to clarify that: (1) on the left, blue and red blocks denote frozen and trainable parameters, respectively; and (2) on the right, blue and red columns represent the performance of baselines versus DIM-4.6B-Edit.
>
> **Q7: Providing the rationale or references for the 21 dimensions in DIM-T2I.**
>
> A: The 21 dimensions were derived from a thorough literature review and an empirical analysis of existing understanding datasets and benchmarks. They are listed as follows:
>
> MME, MMMU, MMMU-Pro, MMLU, MMStar, MMT-Bench, MM-Vet, MM-Vet V2, LLaVA-Bench-Wild, LLaVA-Bench-Wilder, WildVision, COCO, VQAv2, OK-VQA, TextCaps, TextVQA, AI2D, ChartQA, DocVQA, MathVista, MIA-Bench, MegaBench, RWQA, OCRBench, GSM8K, GPQA, IFEval.
>
> We assert that established understanding datasets/benchmarks effectively capture frequent real-world interactions, making them an ideal foundation for complex T2I tasks. By leveraging this diversity, DIM-4.6B-T2I acquires the robust long-instruction processing and world knowledge necessary for CoT comprehension.
>
> **Q8: Is the SSIM-, DINOv2-, and CLIP similarity-based filtering in UltraEdit-160K-CoT necessary? Why? how does it differ from the later alignment classification?**
>
> A: We distinguish the roles of these metrics as follows:
> 1. **Image-to-Image (SSIM, DINOv2, CLIP):** These evaluate visual similarity and background preservation. High scores here ensure visual consistency, thereby improving training stability.
> 2. **Image-Text (Prompt Optimization):** This focuses on alignment. Its objective is to eliminate ambiguity (e.g., refining "add dogs" to "add two dogs") to prevent noisy supervision signals.
>
> **Q9: Explicitly explain why data leakage (in BLIP3-o-60K) might occur.**
>
> A: We excluded BLIP3-o-60K because it is explicitly curated to align with GenEval's structural patterns. While exact prompts differ, the data distributions are nearly identical, which constitutes a form of data leakage or "benchmark hacking." We prioritized fair evaluation over artificially inflating SOTA scores and have clarified this distinction in the revision to ensure scientific integrity.

---

> ### Author Response · Authors · 2025-11-29
> **Briefly Summarized Response to Reviewer RiTv-Part 1/3**
>
> **Summary: The reliance on GPT-4o as the designer limits the external validity of the conclusions.**
>
> A: We address concerns about reliance on GPT-4o through validations from both inference and training perspectives:
>
> 1. **Inference Generalizability (Table 6, attached below):** We demonstrate that various open-source MLLMs can replace GPT-4o as the "designer" without performance loss, proving the paradigm is highly accessible and not tied to proprietary models.
>
> | Designer | Add | Adjust | Extract | Replace | Remove | Background | Style | Hybrid | Action | Overall |
> | :--- | :---: | :---: | :---: | :---: | :---: | :---: | :---: | :---: | :---: | :---: |
> | - | 3.53 | 3.23 | 2.01 | 3.49 | 1.47 | 3.42 | 4.79 | 2.35 | 3.64 | 3.10 |
> | Qwen2.5-VL-7B | 3.95 | 3.35 | 2.25 | 3.85 | 3.31 | 3.57 | 4.88 | 2.81 | 4.02 | 3.55 |
> | MiMo-VL-7B | 3.95 | 3.32 | 2.20 | 3.75 | 2.46 | 3.82 | 4.88 | 2.52 | 3.93 | 3.43 |
> | InternVL3.5-8B | 3.98 | 3.40 | 2.05 | 4.14 | 3.30 | 3.84 | 4.94 | 2.77 | 3.89 | 3.59 |
> | GLM-4.1V-9B | 3.95 | 3.27 | 2.23 | 3.90 | 2.64 | 3.81 | 4.92 | 2.23 | 4.02 | 3.44 |
> | GPT-4o | 4.09 | 3.47 | 2.30 | 4.00 | 3.43 | 3.87 | 4.92 | 2.85 | 4.08 | 3.67 |
>
> 2. **Training Generalizability:** To validate the generalizability of DIM-Edit's CoT blueprints, we fine-tuned `Qwen2.5-VL-3B` on DIM-Edit CoTs. The resulting model generated superior blueprints that improved editing performance (see Table below). This confirms that DIM-Edit effectively bridges the gap between open-source and proprietary models, serving as a dual-utility asset for both editing and reasoning tasks.
>
> | Designer | Add | Adjust | Extract | Replace | Remove | Background | Style | Hybrid | Action | Overall |
> | :--- | :---: | :---: | :---: | :---: | :---: | :---: | :---: | :---: | :---: | :---: |
> | - | 3.53 | 3.23 | 2.01 | 3.49 | 1.47 | 3.42 | 4.79 | 2.35 | 3.64 | 3.10 |
> | Qwen2.5-VL-3B❄️ | 3.80 | 3.24 | 2.03 | 3.89 | 3.21 | 3.52 | 4.92 | 2.71 | 4.05 | 3.49 |
> | Qwen2.5-VL-3B🔥 | 3.96 | 3.36 | 2.25 | 3.98 | 3.31 | 3.81 | 4.95 | 2.83 | 4.02 | 3.61 |
> | GPT-4o | 4.09 | 3.47 | 2.30 | 4.00 | 3.43 | 3.87 | 4.92 | 2.85 | 4.08 | 3.67 |
>
> **W1:  Evaluation relies on LLM-as-Judge; adding quantifiable, automated metrics is recommended.**
>
> A: Following this advice, we evaluated `DIM-4.6B-Edit` on **MagicBrush**, chosen for its reliable human-edited test set and precise pixel-level automated metrics (L1, CLIP-I, DINO):
>
> | Method | Gen Params | L1 $\downarrow$ | CLIP-I $\uparrow$ | DINO $\uparrow$ |
> | :--- | :---: | :---: | :---: | :---: |
> | InstructP2P [CVPR'23] | 0.9B | 0.114 | 0.851 | 0.744 |
> | MagicBrush [NeurIPS'23] | 0.9B | 0.074 | 0.908 | 0.847 |
> | UltraEdit [NeurIPS'24] | 2.0B | 0.066 | 0.904 | 0.852 |
> | FluxEdit [HuggingFace] | 12.0B | 0.114 | 0.779 | 0.663 |
> | FLUX.1 Fill [HuggingFace] | 12.0B | 0.192 | 0.795 | 0.669 |
> | RF-Solver Edit [ICML'25] | 12.0B | 0.112 | 0.766 | 0.675 |
> | ACE++ [arXiv'25] | 12.0B | 0.195 | 0.741 | 0.591 |
> | ICEdit [NeurIPS'25] | 12.0B | 0.060 **(1st)** | 0.928 **(1st)** | 0.853 **(2nd)** |
> | DIM-4.6B-Edit | 1.6B | 0.065 **(2nd)** | 0.928 **(1st)** | 0.882 **(1st)** |
>
>
> As shown in the table, our model achieves superior performance comparable to the previous SOTA `ICEdit` from two perspectives:
>
> 1. **Efficiency:** `ICEdit` relies on a massive 12B backbone (FLUX.1 Fill), whereas `DIM-4.6B-Edit` uses a compact 1.6B backbone.
> 2. **Generalizability:** MagicBrush comprises ~20% of `ICEdit`'s training data but less than 3% of our DIM's. Matching the performance of a 5x larger competitor with significantly less in-domain data strongly validates the efficiency and generalizability of the Draw-In-Mind paradigm and the DIM data.
>
> **W2:  Claims to be a unified understanding–generation model, but lacks evaluation on understanding tasks.**
>
> A: We intentionally froze the MLLM to isolate the contribution of our DIM dataset for the Dataset Track. Achieving SOTA editing performance with this constrained architecture proves that gains stem from data quality, not architectural engineering.
>
> Since the MLLM is frozen, its understanding capabilities remain identical to the original paper. We have added the requested metrics below to prevent confusion:
>
> | Model | Params | MME-P | MMB | SEED | MMMU | MM-Vet |
> | :--- | :---: | :---: | :---: | :---: | :---: | :---: |
> | Janus | 1.3B🔥 | 1338.0 | 69.4 | 63.7 | 30.5 | 34.3 |
> | Emu3-Gen | 8.0B🔥 | - | 58.5 | 68.2 | 31.6 | 37.2 |
> | Show-o | 1.3B🔥 | 1097.2 | - | - | 26.7 | - |
> | Show-o2-7B | 7.0B🔥 | 1620.5 | 79.3 | 69.8 | 48.9 | - |
> | Janus-Pro-7B | 7.0B🔥 | 1567.1 | 79.2 | 72.1 | 41.0 | 50.0 |
> | BAGEL | 14.0B🔥 | 1687.0 | 85.0 | - | 55.3 | 67.2 |
> | MetaQuery-L | 3.0B❄️ \| 3.2B🔥 | 1574.3 | 78.6 | 73.8 | 53.1 | 63.2 |
> | DIM-4.6B-T2I/Edit | 3.0B❄️ \| 1.6B🔥 | 1574.3 | 78.6 | 73.8 | 53.1 | 63.2 |
>
> It is worth noting that our setup demonstrates flexibility; users can seamlessly upgrade to larger MLLMs for enhanced performance without complex parameter tuning.

---

> ### Author Response · Authors · 2025-11-29
> **Briefly Summarized Response to Reviewer RiTv-Part 2/3**
>
> **W3: Insufficient exploration of how the new workflow impacts image generation.**
>
> A: To assess individual component contributions, we conducted an ablation study in our original manuscript (Table 9, attached below):
>
> | CoT Composition | Add | Adjust | Extract | Replace | Remove | Background | Style | Hybrid | Action | Overall |
> | :--- | :---: | :---: | :---: | :---: | :---: | :---: | :---: | :---: | :---: | :---: |
> | DIM-Edit | 4.09 | 3.47 | 2.30 | 4.00 | 3.43 | 3.87 | 4.92 | 2.85 | 4.08 | 3.67 |
> | &nbsp;&nbsp;&nbsp; w/o GLP | 3.85 | 3.29 | 2.06 | 3.91 | 3.24 | 3.55 | 4.80 | 2.79 | 3.92 | 3.49 |
> | &nbsp;&nbsp;&nbsp; w/o LOP | 3.80 | 3.15 | 1.92 | 3.83 | 3.07 | 3.60 | 4.79 | 2.44 | 3.92 | 3.39 |
> | &nbsp;&nbsp;&nbsp; w/o EAL | 3.79 | 3.25 | 1.96 | 3.73 | 2.96 | 3.65 | 4.81 | 2.82 | 3.82 | 3.42 |
> | &nbsp;&nbsp;&nbsp; w/o EII | 3.77 | 3.22 | 1.82 | 3.88 | 2.96 | 3.61 | 4.78 | 2.55 | 3.58 | 3.35 |
>
> Results confirm that all components contribute positively, though impacts vary. GLP shows minimal impact as the task is relatively trivial for the generator; LOP, EAL, and EII demonstrate significant impact because these components require precise regional focus and complex reasoning, which are non-trivial tasks for the generator and thus essential for high performance.
>
> To complement numerical metrics, we added Figures 9-13 to demonstrate advanced scenarios where `DIM-4.6B-Edit` succeeds while baselines (including `Janus-4o`, `Step1X-Edit`, and `DIM w/o CoT`) fail. These cases highlight the practical impact of the Draw-In-Mind paradigm:
>
> 1. Figure 9: **[Instruction Disambiguation]**.
> 2. Figure 10: **[Edit Navigation & Structural Planning]**.
> 3. Figure 11: **[Commonsense-guarded Editing]**.
> 4. Figure 12: **[Advanced Causal Editing]**.
> 5. Figure 13: **[Advanced Temporal Editing]**.
>
> We believe these visualizations provide an intuitive demonstration of how the Draw-In-Mind paradigm significantly enhances image editing capabilities.
>
> Besides, we share the limitations of our work, and provide three failure types with six specific cases in Figures 14-16 to intuitively show the boundaries of DIM-4.6B-Edit:
>
> 1. Figure 14: **[Large-scale All-in-One Editing]**.
> 2. Figure 15: **[Text & Logic Editing]**.
> 3. Figure 16: **[Reference-free Editing (in Pixel Space)]**.
>
> In summary, failures primarily arise when tasks require drastic divergence from the source or the rendering of complex text/geometrical shapes. Even in these scenarios, `DIM-4.6B-Edit` maintains better instruction following than baselines, highlighting persistent challenges among existing open-source models.
>
>
> **Q1: Is there no way for a unified model to unify both thinking/reasoning and generation? If not, what is the significance of unifying understanding and generation? (Just for discussion)**
>
> A: We implemented a unified framework for reasoning and generation by utilizing the inherent `Qwen2.5-VL-3B` model as the designer. Our experiments (see below) demonstrate that fine-tuning this internal module on DIM-Edit CoT data leads to improved performance. While a comprehensive exploration of this paradigm is beyond the current scope, the provided results effectively validate its potential:
>
> | Designer | Add | Adjust | Extract | Replace | Remove | Background | Style | Hybrid | Action | Overall |
> | :--- | :---: | :---: | :---: | :---: | :---: | :---: | :---: | :---: | :---: | :---: |
> | - | 3.53 | 3.23 | 2.01 | 3.49 | 1.47 | 3.42 | 4.79 | 2.35 | 3.64 | 3.10 |
> | Qwen2.5-VL-3B❄️ | 3.80 | 3.24 | 2.03 | 3.89 | 3.21 | 3.52 | 4.92 | 2.71 | 4.05 | 3.49 |
> | Qwen2.5-VL-3B🔥 | 3.96 | 3.36 | 2.25 | 3.98 | 3.31 | 3.81 | 4.95 | 2.83 | 4.02 | 3.61 |
> | GPT-4o | 4.09 | 3.47 | 2.30 | 4.00 | 3.43 | 3.87 | 4.92 | 2.85 | 4.08 | 3.67 |
>
> The second row is the "self-play" mode where internal MLLM-generated CoT embeddings directly guide the painter, bypassing the need for an external designer. We initially minimized this to maintain a streamlined narrative focused on external integration, but have now expanded this discussion in the revision.
>
> For a detailed discussion on the significance of **unifying understanding and editing**, please refer to our response to Q2.

---

> ### Author Response · Authors · 2025-11-29
> **Briefly Summarized Response to Reviewer RiTv-Part 3/3**
>
> **Q2: How does this work differ from prior encoder-side work in models that use an MLLM to unify generation tasks? Is the main difference simply the addition of CoT?**
>
> A: We clarify that unifying *editing* and understanding is distinct from unifying *generation* and understanding. Our work pioneers the former by addressing unique challenges:
>
> 1. **Task Nature:** Unlike T2I, editing is a multimodal-to-visual task. Traditional methods decouple reference images from instructions, leading to semantic loss. We integrate them to ensure instructions are grounded in the visual context.
> 2. **UMM-Powered Architecture:** While `Step1X-Edit` connects models, it lacks sophisticated instruction handling. We pioneer the integration of UMM capabilities with CoT reasoning to achieve instruction-faithful editing. The recent release of `Nano Banana Pro` (which also employs CoT) further validates the timeliness and significance of our Draw-In-Mind paradigm.
>
> From the data perspective (which is paramount for this Dataset Track submission), we address the scarcity of high-quality open-source training data (in contrast to proprietary assets like BAGEL), and provide `DIM-T2I` (14M long-instruction pairs) as a critical contribution to the research community. We further validated the necessity of this foundation by training a model exclusively on `DIM-Edit`, the results are presented below:
>
> | Initialization | Add | Adjust | Extract | Replace | Remove | Background | Style | Hybrid | Action | Overall |
> | :--- | :---: | :---: | :---: | :---: | :---: | :---: | :---: | :---: | :---: | :---: |
> | Scratch | 2.70 | 2.56 | 1.93 | 2.23 | 2.47 | 2.82 | 4.68 | 2.38 | 2.15 | 2.66 |
> | DIM-4.6B-T2I | 4.09 | 3.47 | 2.30 | 4.00 | 3.43 | 3.87 | 4.92 | 2.85 | 4.08 | 3.67 |
>
> The significant performance gap between the scratch model and the `DIM-4.6B-T2I` initialized version confirms that `DIM-T2I` is an indispensable cornerstone for the Draw-In-Mind paradigm and a valuable resource for the broader community.
>
> In the domain of image editing, we pioneer the first high-quality CoT dataset tailored for UMM-powered editing (233K samples). Its value is demonstrated by:
>
> 1. **Extreme Efficiency:** Our DIM-4.6B-Edit outperforms 5x larger competitors. Notably, even our scratch version matches `UltraEdit` (initialized from SD3 and post-trained on 4M pairs), demonstrating a **~20x improvement** in data efficiency.
> 2. **Dual Utility:** The clear performance gains observed when utilizing CoTs from the fine-tuned Qwen2.5-VL-3B (see Q1 Table) demonstrate that DIM-Edit is not only effective for directly training editing behaviors but also serves as a robust corpus for enhancing the planning of accurate "design blueprints" on the understanding side.
>
> In summary, we provide a **full-stack data infrastructure** ranging from foundational long-instruction comprehension (DIM-T2I) to advanced, complex CoT-guided editing (DIM-Edit). This infrastructure democratizes access and encourages broader engagement in advanced image editing research. We believe these contributions are highly meaningful for the academic community.
>
> **Q3: Why claim as a unified model without evaluating understanding tasks and other generation tasks?**
>
> A: We clarify that our primary objective is enhancing **image editing** via the Draw-In-Mind paradigm. General image generation tasks fall outside this scope. For understanding capability evaluations, please refer to our response to **W2**. Consequently, our evaluation focuses on widely recognized editing benchmarks, including the newly incorporated **MagicBrush**.
>
> **[Follow-up Discussion on Nov 24]: Discussion of MGIE [1], Kosmos-G [2], and MIGE [3].**
>
> A: We thank the reviewer for highlighting these relevant modeling-focused works (MGIE [1], Kosmos-G [2], MIGE [3]). We agree they are insightful but note that our work serves as a distinct *data-centric* complement to their *architectural* approaches. We differentiate our contribution in three key ways:
>
> 1. **Native vs. Compressed Tokens:** While referenced works compress visual modalities into limited learnable queries (causing information loss), `DIM-4.6B-Edit` retains full, native MLLM tokens, ensuring minimal information loss.
> 2. **Simplicity via Data:** We demonstrate that high-quality data (`DIM-T2I/Edit`) enables SOTA performance using only a simple 2-layer MLP and a frozen MLLM, eliminating the need for complex connector designs or unstable MLLM tuning required by prior methods.
> 3. **Generalizability:** Unlike the tightly coupled architectures of the cited works, our data-centric approach offers flexibility, allowing seamless upgrades to newer MLLMs (as shown in our generalizability experiments in Table 6) without extensive retraining.
>
> We argue that previous architectural compromises stemmed from a lack of high-quality complex data. Our work fills this gap. We will cite and discuss these papers in the final revision.
>
>
> [1] MGIE, ICLR'24
>
> [2] Kosmos-G, ICLR'24
>
> [3] MIGE,  ACM MM'25

---

> ### Author Response · Authors · 2025-11-29
> **Briefly Summarized Response to Reviewer PX1Q**
>
> **W1: Potentially Unfair Parameter Comparison. Comparing the 4.6B parameter core model against the end-to-end parameters of other models may present an incomplete picture of the total "cost" of the method.**
>
> We clarify that utilizing the inherent `Qwen2.5-VL-3B` as the designer creates a streamlined **one-step designer-painter system**, eliminating the need for external inference rounds. This configuration alone achieves SOTA performance among open-source models, validating the efficiency of the Draw-In-Mind paradigm and the quality of DIM-Edit data. Additionally, we demonstrate that fine-tuning this internal MLLM on DIM-Edit CoT data yields further performance gains (see Table below):
>
> | Designer | Add | Adjust | Extract | Replace | Remove | Background | Style | Hybrid | Action | Overall |
> | :--- | :---: | :---: | :---: | :---: | :---: | :---: | :---: | :---: | :---: | :---: |
> | - | 3.53 | 3.23 | 2.01 | 3.49 | 1.47 | 3.42 | 4.79 | 2.35 | 3.64 | 3.10 |
> | Qwen2.5-VL-3B❄️ | 3.80 | 3.24 | 2.03 | 3.89 | 3.21 | 3.52 | 4.92 | 2.71 | 4.05 | 3.49 |
> | Qwen2.5-VL-3B🔥 | 3.96 | 3.36 | 2.25 | 3.98 | 3.31 | 3.81 | 4.95 | 2.83 | 4.02 | 3.61 |
> | GPT-4o | 4.09 | 3.47 | 2.30 | 4.00 | 3.43 | 3.87 | 4.92 | 2.85 | 4.08 | 3.67 |
>
> We clarify that the "self-play" configuration (Row 2) allows the internal MLLM to drive the painter directly, bypassing external inference. Based on this, we conclude our work offers dual advantages:
>
> 1. **Under Parameter-Constrained Setting:** DIM data efficiently bridges the gap between open- and closed-source designers via lightweight fine-tuning.
> 2. **Under Data-Constrained Setting:** DIM-4.6B-Edit enables immediate performance boosts by seamlessly integrating advanced external designers without retraining, proving the high generalizability of our DIM CoT.
>
> **Q1: Provide some analysis of failure cases. Understanding the limitations and failure modes would help the readers to better understand the boundaries of this approach.**
>
> A: We have included two additional visualization parts to intuitively show the boundaries of this approach as follows.
>
> **[Failure Case]** We provide three failure types with six specific cases in Figures 14-16 to show the boundaries of our DIM-4.6B-Edit:
>
> 1. Figure 14: **[Large-scale All-in-One Editing]**.
> 2. Figure 15: **[Text & Logic Editing]**.
> 3. Figure 16: **[Reference-free Editing (in Pixel Space)]**.
>
> In summary, failures primarily arise when tasks require drastic divergence from the source or the rendering of complex text/geometrical shapes. Even in these scenarios, `DIM-4.6B-Edit` maintains better instruction following than baselines, highlighting persistent challenges among existing open-source models.
>
> **[Advanced Editing]** We provide Figures 9-13 in the revised manuscript, which showcases several advanced usage scenarios. These examples demonstrate complex cases that are successfully handled by DIM-Edit-4.6B, highlighting capabilities that remain beyond the reach of current baseline methods:
>
> 1. Figure 9: **[Instruction Disambiguation]**.
> 2. Figure 10: **[Edit Navigation & Structural Planning]**.
> 3. Figure 11: **[Commonsense-guarded Editing]**.
> 4. Figure 12: **[Advanced Causal Editing]**.
> 5. Figure 13: **[Advanced Temporal Editing]**.
>
> We believe these visualizations provide an intuitive demonstration of how the Draw-In-Mind paradigm significantly enhances image editing capabilities.

---

> ### Author Response · Authors · 2025-11-29
> **Briefly Summarized Response to Reviewer B3js**
>
> **W1: How well can the model handle the inevitably noisier and sometimes incorrect instructions it will receive in a real-world setting?**
>
> A: We have expanded the manuscript with a comprehensive case study analysis. We demonstrate how the Draw-In-Mind workflow effectively handles ambiguous or challenging instructions where baselines fail. Furthermore, we transparently present specific failure cases to openly discuss the current limitations of `DIM-4.6B-Edit`.
>
> Firstly, we explored cases where user instructions are ambiguous or require advanced reasoning in Figures 9-13. These examples demonstrate complex scenarios that are successfully handled by DIM-Edit-4.6B, highlighting capabilities that remain beyond the reach of current baseline methods:
>
> 1. Figure 9: **[Instruction Disambiguation]**.
> 2. Figure 10: **[Edit Navigation & Structural Planning]**.
> 3. Figure 11: **[Commonsense-guarded Editing]**.
> 4. Figure 12: **[Advanced Causal Editing]**.
> 5. Figure 13: **[Advanced Temporal Editing]**.
>
> We believe these visualizations provide an intuitive demonstration of how the Draw-In-Mind paradigm significantly enhances image editing capabilities.
>
> Next, we provide three failure types with six specific cases in Figures 14-16 to intuitively show the boundaries of our DIM-4.6B-Edit:
>
> 1. Figure 14: **[Large-scale All-in-One Editing]**.
> 2. Figure 15: **[Text & Logic Editing]**.
> 3. Figure 16: **[Reference-free Editing (in Pixel Space)]**.
>
> In summary, failures primarily arise when tasks require drastic divergence from the source or the rendering of complex text/geometrical shapes. Even in these scenarios, `DIM-4.6B-Edit` maintains better instruction following than baselines, highlighting persistent challenges among existing open-source models.
>
> **Q1: Whether this "designing" capability could be internalized into the main understanding module itself, thereby streamlining the inference process? Did you explore standard unified training approaches where the model is taught to implicitly "reason" before "acting" (generating the image)?**
>
> A: We clarify that utilizing the inherent `Qwen2.5-VL-3B` as the designer creates a streamlined **one-step designer-painter system**, eliminating the need for external inference rounds. This configuration alone achieves SOTA performance among open-source models, validating the efficiency of the Draw-In-Mind paradigm and the quality of DIM-Edit data. Additionally, we demonstrate that fine-tuning this internal MLLM on DIM-Edit CoT data yields further performance gains (see Table below):
>
> | Designer | Add | Adjust | Extract | Replace | Remove | Background | Style | Hybrid | Action | Overall |
> | :--- | :---: | :---: | :---: | :---: | :---: | :---: | :---: | :---: | :---: | :---: |
> | - | 3.53 | 3.23 | 2.01 | 3.49 | 1.47 | 3.42 | 4.79 | 2.35 | 3.64 | 3.10 |
> | Qwen2.5-VL-3B❄️ | 3.80 | 3.24 | 2.03 | 3.89 | 3.21 | 3.52 | 4.92 | 2.71 | 4.05 | 3.49 |
> | Qwen2.5-VL-3B🔥 | 3.96 | 3.36 | 2.25 | 3.98 | 3.31 | 3.81 | 4.95 | 2.83 | 4.02 | 3.61 |
> | GPT-4o | 4.09 | 3.47 | 2.30 | 4.00 | 3.43 | 3.87 | 4.92 | 2.85 | 4.08 | 3.67 |
>
> We clarify that the "self-play" configuration (Row 2) allows the internal MLLM to drive the painter directly, bypassing external inference. Based on this, we conclude our work offers dual advantages:
>
> 1. **Under Parameter-Constrained Setting:** DIM data efficiently bridges the gap between open- and closed-source designers via lightweight fine-tuning.
> 2. **Under Data-Constrained Setting:** DIM-4.6B-Edit enables immediate performance boosts by seamlessly integrating advanced external designers without retraining, proving the high generalizability of our DIM CoT.
>
> **Q2: What are the key trade-offs that led to favoring the current explicit two-stage design over a more integrated, end-to-end approach? Is it primarily for the flexibility of swapping "designers," or did you find that integrated models inherently struggle to balance these two responsibilities even with high-quality CoT training data?**
>
> A: We clarify that our design prioritizes the flexibility of swapping "designers" to democratize access, rather than solely chasing benchmark scores through massive end-to-end training. While our integrated approach already yields SOTA results and could theoretically be maximized by full system fine-tuning, such an undertaking is outside our scope. Instead, we provide a minimal, efficient training recipe that rigorously verifies the effectiveness of the Draw-In-Mind paradigm and the high quality of the DIM-Edit data.
>
> This modularity significantly lowers the barrier to entry, offering a platform where resource-rich groups can build new systems while others can affordably train the compact `DIM-4.6B-Edit` or integrate existing advanced MLLMs like `InternVL3.5`. By proposing a novel solution focused on "data form" rather than the industrial trend of scaling up model size, we aim to provide a robust, accessible alternative that inspires wider engagement and innovation across the research community.

---

### Meta-Review · Area_Chair_HVk8 · 2026-01-05

**Summary:**

The reviewers initially rated the paper 8/6/6/4 leaning heavily towards acceptance. All the reviewers acknowledged the value in the proposed dataset as well as the addition of Chain of Thought in the visual editing process. The concerns listed by the reviewers mostly centered around the robustness of the evaluation protocol, e.g. whether LLM-as-a-judge is sufficient vs having a manually verified subset and a potential training, inference mismatch.

**Reviewer Concerns:**

The main concern was the thoroughness of the evaluation protocol and the value in having a manually verified subset. In the rebuttal phase, the authors presented a large number of additional experiments which increase the robustness of their evaluation and demonstrate that the gains shown are not dependent on having a specific judge. The authors also presented a thorough rebuttal for other concerns such as the training, inference mismatch.

**Reviewer Scores:**

Based on the weaknesses listed, author rebuttal, and the reviewer engagement, it seems like the the positive reviews would have stayed the same and the negative review would have been raised to be slightly positive. This matches what the authors report as the reviewer scores after discussion.

---

### Decision · Program_Chairs · 2026-01-26

Accept (Poster)